# ESSA: Evolutionary Strategies for Scalable Alignment

## Abstract

Alignment of Large Language Models (LLMs) typically relies on Reinforcement Learning from Human Feedback (RLHF) with gradient-based optimizers such as Proximal Policy Optimization (PPO) or Group Relative Policy Optimization (GRPO). While effective, these methods require complex distributed training, large memory budgets, and careful hyperparameter tuning, all of which become increasingly difficult at billion-parameter scale. We present ESSA, Evolutionary Strategies for Scalable Alignment, a gradient-free framework that aligns LLMs using only forward inference and black-box optimization. ESSA focuses optimization on Low-Rank Adapters (LoRA) and further compresses their parameter space by optimizing only the singular values from an singular value decomposition (SVD) of each adapter matrix. This dimensionality reduction makes evolutionary search practical even for very large models and allows efficient operation in quantized INT4 and INT8 inference mode. Across these benchmarks ESSA improves the test accuracy of Qwen2.5-Math-7B by 12.6% on GSM8K and 14.8% on PRM800K, and raises the accuracy of LLaMA3.1-8B on IFEval by 22.5%, all compared with GRPO. In large-scale settings ESSA shows stronger scaling than gradient-based methods: on Qwen2.5-32B for PRM800K it reaches near-optimal accuracy twice as fast on 16 GPUs and six times as fast on 128 GPUs compared with GRPO. These results position evolutionary strategies as a compelling, hardware-friendly alternative to gradient-based LLM alignment, combining competitive quality with substantially reduced wall-clock time and engineering overhead.

## 1 Introduction

Large Language Models (LLMs) have made significant progress thanks to alignment techniques that guide the model's behavior toward human preferences. Online methods, predominantly Reinforcement Learning from Human Feedback (RLHF) with Group Relative Policy Optimization (GRPO), Proximal Policy Optimization (PPO), or REINFORCE Leave-One-Out (RLOO), remain the de facto standard in practice (Ouyang et al., 2022a; Schulman et al., 2017; Ahmadian et al., 2024; Shao et al., 2024). However, these pipelines are complex to implement: they involve actor/critic training, trajectory generation, backpropagation through long sequences, and distributed synchronization. As models grow larger, practical considerations about how to distribute components across GPUs become critical, with sensitivity to numerous interacting hyperparameters and communication bottlenecks (Zheng et al., 2023; Sheng et al., 2024).

To address these issues, we revisit evolutionary strategies as a scalable, gradient-free alternative. These methods require only forward passes with perturbed parameters and simple aggregation of scalar fitness values, enabling near-parallel training, low memory usage, and robustness to sparse or noisy rewards (Salimans et al., 2017). The classic concern of poor efficiency in very high-dimensional spaces can be addressed through aggressive search space reduction.

We introduce **ESSA (Evolutionary Strategies for Scalable Alignment)**, which pairs Evolutionary Strategies (ES) with parameter-efficient adaptation. We limit optimization to low-rank adapters of attention matrices (Q/K/V/O) and further compress them via singular value decomposition (SVD) parameterization of singular values, making black-box search practical and interpretable (Hu et al. (2021), Vaswani et al. (2017)).

In this work, ESSA is introduced as a gradient-free online alignment procedure applied strictly after a supervised fine-tuning (SFT) warm-start. The method replaces only the post-SFT stage. ESSA can operate fully in quantized INT4 or INT8 inference mode, enabling efficient adaptation of models up to approximately 72B parameters on a single high-memory GPU Dettmers et al. (2022). Across mathematics of varying difficulty, instruction following, and general assistant tasks, ESSA matches baselines trained by GRPO, while offering stronger system scalability and reduced hyperparameter fragility (Shao et al. (2024)).

Taken together, ESSA turns alignment into a simple, highly parallel evaluation loop with minimal synchronization – an attractive fit for modern clusters and continual training settings – while retaining the quality expected from state-of-the-art online methods.

## 2 RELATED WORKS

**Alignment.** Alignment of large language models is commonly based on RLHF (Ouyang et al., 2022a), typically optimized with PPO (Schulman et al., 2017) or REINFORCE (Sutton et al., 1999). Variants such as RLOO (Ahmadian et al., 2024), GRPO (Shao et al., 2024), REINFORCE++ (Hu, 2025) and DAPO (Yu et al., 2025) stabilize training by using relative advantages within groups, yet still inherit gradient estimation variance and substantial memory cost. Offline preference learning (Rafailov et al., 2024; Hong et al., 2024; Meng et al., 2024)) removes online rollouts but is bounded by dataset coverage and preference noise, limiting generalization (Tang et al., 2024; Xu et al., 2024).

**Parameter-efficient training of LLMs.** Parameter-efficient fine-tuning (PEFT) techniques reduce the cost of adapting large language models by updating only a small subset of parameters. Beyond classic approaches such as adapters (Houlsby et al., 2019), prefix-tuning (Li & Liang, 2021), and LoRA (Hu et al., 2021), more recent methods include DoRA (Liu et al., 2024), VeRA (Kopiczko et al., 2024), and tensor-based approaches like LoTR (Bershatsky et al., 2024). Another way to reduce the number of trainable parameters is to optimize only the eigenvalues in the SVD decomposition of transformer matrices, as done in Transformer[2] (Sun et al., 2025), which also inspired our method.

**Evolution strategies.** ES, including Covariance Matrix Adaptation (CMA-ES) (Hansen & Ostermeier, 2001), Natural Evolution Strategies (NES) (Wierstra et al., 2011), Augmented Random Search (ARS) (Mania et al., 2018), and Guided Evolutionary Strategies (GES) (Maheswaranathan et al., 2018), provide powerful gradient-free optimization that is highly parallel and robust to sparse rewards (Salimans et al., 2017). Zero-order optimizers (Zhang et al., 2024) approximate gradients of loss differences but tend to lag in complex reasoning tasks. Applications of ES to large language model alignment remain rare due to high-dimensionality challenges: existing works like GENOME/GENOME+ (Zhang et al., 2025), LoRAHub (Huang et al., 2024), and DFO (Jin et al., 2024) reduce parameter space but are still limited to a variety of experiments.

## 3 EVOLUTIONARY STRATEGIES FOR SCALABLE ALIGNMENT

### 3.1 MOTIVATION

Modern online alignment of LLMs is dominated by gradient-based RLHF variants. In practice, these pipelines are costly: they require long rollouts, backpropagation through large contexts, optimizer-state synchronization across devices, and careful hyperparameter tuning. As model size grows, memory pressure and training fragility increase, and sparse or noisy rewards further destabilize learning.

ESSA offer the opposite trade-off. ESSA restricts learning to low-rank LoRA adapters represented in a compact SVD parameterization, where a few singular values is optimized. In this setting, evolutionary updates require nothing beyond forward evaluations under parameter perturbations together with aggregated scalar rewards, which yields a naturally parallelizable, memory-efficient training loop. The standard drawback of evolutionary methods, namely poor efficiency in extremely high-dimensional search spaces, is addressed here by forcing the search to remain within a compact and task-aligned low-rank subspace.

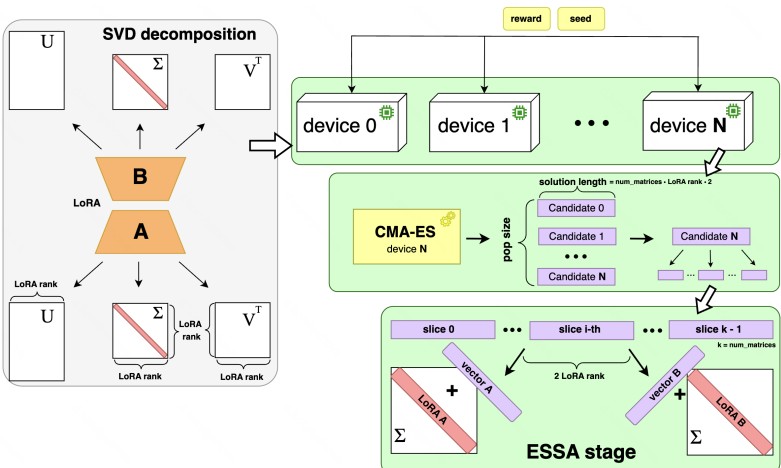

Figure 1: Illustration of the ESSA framework. LoRA adapters are first initialized via SFT and decomposed into fixed SVD bases with trainable singular values. The term device N denotes the GPU worker in distributed evaluation. CMA-ES receives a seed at each device, generates a population of size N+1 locally, evaluates a different candidate, and returns a reward. Each candidate is a vector $\sigma_i \in \mathbb{R}^{\textbf{solution\_length}}, i = 0, 1, \ldots, N$ and is added to SVD vectors of the training matrices. It is partitioned into contiguous slices, each of which corresponds to one LoRA matrix (e.g. $W_Q, W_K, W_V, W_O$ for each transformer layer) and contains $2 \cdot \textbf{LoRA\_rank}$ singular values (for factors $A$ and $B$). The solution length is the dimensionality of the candidate vector, i.e., the concatenation of all perturbation of the trainable LoRA singular values across all matrices and layers.

With the number of layers (**num_layers**), number of matrices per layer (**num_matrices_per_layer**), LoRA_rank: **num_matrices** = **num_layers** · **num_matrices_per_layer** and **solution_length** = **num_matrices** · **LoRA_rank** · 2.

Since the loop is inference-only, ESSA runs efficiently under low-precision (INT4/INT8) inference. It also scales linearly with available hardware by evaluating candidates in parallel with minimal communication (seed + reward) (Salimans et al., 2017). This design not only yields a compact gradient-free search space but also addresses the practical problem of optimizing the alignment stage in large-scale, multi-GPU training pipelines. ESSA integrates established components (LoRA, SVD compression, evolutionary search) into a unified framework that enables inference-only, quantized, distributed alignment with minimal engineering and synchronization overhead.

Crucially, ESSA is not a full training pipeline: it is a gradient-free alignment stage applied on top of a SFT initialization. The SFT warm-start preserves model expressivity and places ES in a task-aware region of parameter space. Without SFT, the LoRA matrices begin as an unstructured Gaussian matrix paired with a zero matrix, whose singular vectors are effectively identity-based and uninformed. We demonstrate the importance of SFT stage using a toy example on MNIST (LeCun & Cortes, 2005) in Appendix E. SFT is also inexpensive compared to online RL alignment. Prior work reports that SFT requires only a small fraction of the compute used for RL-based alignment (Ouyang et al., 2022b). The expensive component of alignment is the online RL loop, and ESSA eliminates this phase while preserving the lightweight warm-start. Experiments without SFT are provided in Appendix F.1 and demonstrate that ESSA can operate without SFT, while being primarily intended as a gradient-free replacement for the online alignment phase.

### 3.2 METHOD

**Initialization.** First, we reduce optimization problem to low-rank updates of each attention projection matrix $W_0 \in \mathbb{R}^{m \times n}$:

$$\Delta W = BA, \qquad B \in \mathbb{R}^{m \times r}, \ A \in \mathbb{R}^{r \times n}, \ r \ll \min(m, n).$$

Then, we run a short SFT stage to initialize the LoRA adapters with task-aware parameters. For every backbone-task pair in the paper, we train exactly one SFT LoRA adapter.

**SVD.** To shrink the trainable space even further, we decompose each SFT-LoRA factor separately:

$$A = U_A \Sigma_A V_A^\top, \qquad B = U_B \Sigma_B V_B^\top.$$

The orthogonal matrices $U_A, V_A, U_B, V_B$ are kept fixed after the initial SFT step, while only the top singular values in $\Sigma_A$ and $\Sigma_B$ remain trainable. This SVD-LoRA representation preserves the expressive power of LoRA while reducing the number of variables that ES must explore.

**Evolutionary Optimization.** We use CMA-ES (Hansen & Ostermeier, 2001) as the optimizer, maintaining and updating a multivariate normal search distribution over the selected singular values. Each ESSA iteration proceeds as follows (more detailed algorithm is provided in Figure 1 and in Appendix A):

1. CMA-ES samples $\lambda \geq 2$ (population size) candidate singular-value vectors.
   $$x_{i+1}^{(k)} \sim m_i + \sigma_i \mathcal{N}(0, C_i) \quad \text{for} \quad k = 1, ..., \lambda, \quad \text{so that} \quad x_{i+1}^{(k)} \sim \mathcal{N}(m_i, \sigma_i^2 C_i),$$
   where $m_i \in \mathcal{R}^n$ is a current mean of the search distribution; $\sigma_i \in \mathcal{R} > 0$ is an "overall" standard deviation; $C_i \in \mathcal{R}^{n \times n}$ is a covariance matrix encoding anisotropic search directions; $n = \text{solution\_length}$. Up to the scalar factor $\sigma_i^2$, $C_i$ is the covariance matrix of the search distribution

2. For each candidate, we reconstruct $A$ and $B$ by adding the candidate's singular-value offsets to the fixed SVD decomposition, forming updated low-rank factors and computing $\Delta W = BA$, and evaluate the model on the alignment objective to obtain a scalar reward.

3. After all candidates are evaluated in parallel, CMA-ES updates $m_i$, $\sigma_i$, and $C_i$. This allows the search distribution to gradually align itself with beneficial directions in the objective landscape. Communication between workers is limited to random seeds and scalar rewards, allowing near-linear scaling across many GPUs.

### 3.3 THEORETICAL ANALYSIS

We compare the per-iteration latency of gradient-based online alignment (e.g., RLHF/GRPO/PPO/RLOO) with a single ESSA update, taking into account both computation and inter-GPU communication. The key observation is that gradient methods require expensive all-reduce of model-size gradients, whereas ESSA communicates only a random seed and scalar rewards. Consequently, the communication cost of ESSA scales essentially independently of model size.

A simple latency model shows that for any realistic cluster bandwidth, there exists a conservative population size threshold such that when the ESSA population size multiplied by the batch size processed by the single population instance $N_{\text{pop}} B^{\text{essa}}$ is below this bound, ESSA achieves a strictly lower per-iteration time than the idealized gradient pipeline. Moreover, because real clusters seldom achieve perfect device splitting or peak network bandwidth, the practical speed advantage of ESSA is typically even larger.

Full mathematical notation, the precise expressions for computation and communication time, and the formal proof of the population size bound are provided in Appendix B.

## 4 EXPERIMENTS

**Tasks and Models.** We evaluate ESSA on three categories of alignment workloads: (i) School-level math reasoning. We train and evaluate on GSM8K with accuracy as the primary metric. Backbones: Qwen2.5-7B, Qwen2.5-Math-7B. (ii) Advanced math reasoning. We train on PRM800K and evaluate on MATH500, AIME'24/'25, MinervaMath, and OlympiadBench using pass@k / avg@16. Backbones: Qwen2.5-Math-7B, Qwen2.5-32B, Qwen2.5-72B. (iii) Instruction following. We train on an if-eval-like dataset and evaluate on IFEval. Backbones: Llama-3.1-8B. (iiii) General-Purpose Assistant. We train and evaluate on HelpSteer2. Backbones: Llama-3.1-8B.

**Baseline.** We use GRPO as a baseline. GRPO is the standard and most robust online RLHF method widely used in both industry and open-source practice (Xi et al., 2025; DeepSeek-AI et al., 2025;

Yang et al., 2025), making it the natural baseline for comparison. Our goal with ESSA is not to outperform specialized gradient optimizers, but to evaluate whether a gradient-free approach can match the quality of GRPO-based RLHF while operating purely in inference mode.

**SFT.** We start from SFT checkpoints appropriate for each dataset. Experiments without SFT are provided in Appendix F.1. We do not include the time spent on the SFT in all the presented results. It is presented separately in Appendix H.2. Since ESSA and GRPO use the same SFT checkpoint within each backbone-task pair, the SFT cost contributes an identical constant to both methods and does not affect their relative performance.

**Other Details.** We use standard BFLOAT16 precision unless otherwise specified. Qwen2.5-72B is also trained under INT4 for per-device evaluation in ESSA. For training the models with the GRPO algorithm, we use the verl library (Sheng et al., 2024), which the authors describe as the most efficient in terms of model allocation and interaction speed. All experiments, unless otherwise noted, are conducted on 8 GPUs. For larger models Qwen2.5-32B and Qwen2.5-72B we use 16 and 32 GPUs, respectively. The sizes of the training and validation datasets, as well as some other details of the experiments, are given in Appendix H.

### 4.1 SENSITIVITY TO HYPERPARAMETERS

**ESSA hyperparameter sensitivity.** We investigate how ESSA accuracy depends on its key hyperparameters – LoRA rank, population size, and the fraction $\alpha$ of singular values optimized in each SVD factor. A full grid search is performed on five settings Qwen2.5-7B and Qwen2.5-Math-7B on GSM8K and PRM800K, and LLaMA-3.1-8B on IFEval. Figure 2 shows the results for Qwen2.5-Math-7B on GSM8K as a representative example.

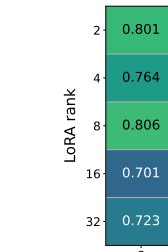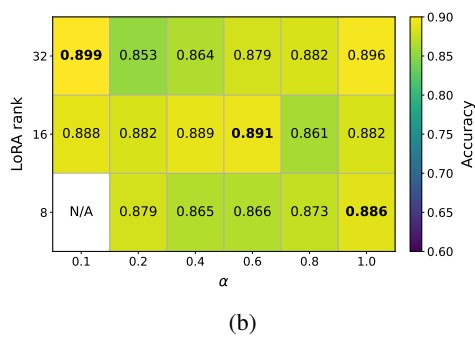

(a)                         (b)

Figure 2: Hyperparameter sensitivity of ESSA on **Qwen2.5-Math-7B** for **GSM8K**. Batch size 100. **(a)** Accuracy when varying LoRA rank and population size. **(b)** For each LoRA rank, the population size is fixed to the best value found in (a), while the percentage $\alpha$ of trainable singular values is varied. This illustrates how ESSA performance depends jointly on adapter rank and the fraction of singular values optimized. The single white cell occurs because for LoRA rank 8 and $\alpha = 0.1$, rounding down yields zero trainable singular values, so no valid accuracy is reported.

This example highlights three consistent trends observed across all tasks: (i) increasing LoRA rank beyond moderate values does not necessarily improve accuracy, and in some PRM800K setups the best results occur with rank as low as 2; (ii) very small populations ($\approx 8$) underperform, while the benefit of larger populations levels off between about 24 and 96; (iii) for the fraction $\alpha$ of trainable singular values, accuracy remains stable once $\alpha$ reaches moderate levels ($\geq 0.4$), showing that ESSA can achieve its best quality without updating all singular values. The complete set of sensitivity plots for the remaining four tasks, which confirm these conclusions and show the cases where rank 2 is optimal, are provided in Appendix C.

**Effect of SFT initialization.** We also examine how the maximum ESSA accuracy depends on the quality of the initial LoRA matrices $A$ and $B$, which are obtained from the SFT stage.

To vary the initialization quality, we traine the SFT model on different fractions of the GSM8K dataset and then run ESSA with identical hyperparameters. The setup is as follows: Qwen2.5-Math-7B, LoRA rank 16, population size 192, and $\alpha = 1.0$. Table 1 reports the final ESSA accuracy as a function of the percentage of SFT data used.

| SFT dataset fraction | 5% | 25% | 50% | 75% | 100% |
|---|---|---|---|---|---|
| Max ESSA accuracy | 0.713 | 0.731 | 0.807 | 0.863 | 0.872 |

Table 1: ESSA validation maximum accuracy as a function of **GSM8k** dataset fraction used to initialize **Qwen2.5-Math-7B** LoRA. Settings: LoRA rank 16, pop. 192, batch size 100, $\alpha = 1.0$).

The results show that reducing the SFT dataset from 100% to only 5% lowers the final ESSA accuracy by more than 15 percentage points. Thus, the quality of the initial supervised fine-tuning plays a key role in the ultimate performance of the aligned model. We also conduct an experiment, running the SFT on one domain and training ESSA on another. The results demonstrate ESSA's robustness to the OOD shift in Appendix G.

## 4.2 PARALLELIZATION

Figure 3 compares the wall-clock time required by GRPO and ESSA to reach a fixed test accuracy of 0.835 on Qwen2.5-32B for the PRM800K benchmark, when training is distributed across an increasing number of GPUs. For ESSA we use a LoRA rank 16 and an population size 128. For GRPO we match the LoRA rank (16) and use a learning rate $1 \times 10^{-5}$, global batch size 512, and mini batch size 64.

As the number of GPUs grows from 16 to 128, both methods benefit from additional parallelism, but the gains differ substantially: GRPO decreases from nearly 400 minutes at 16 GPUs to roughly 150 minutes at 128 GPUs ($\approx 2.6\times$ speed-up); ESSA drops from about 200 minutes to under 20 minutes over the same range ($\approx 10\times$ speed-up). On the smallest cluster (16 GPUs) ESSA is already about twice as fast as GRPO. This gap grows with increasing parallelism and reaches a factor of roughly six on 128 GPUs.

ESSA's scaling advantage follows directly from its inference-only optimization loop: evaluation of population members is embarrassingly parallel and requires communication of only random seeds and scalar rewards. In contrast, GRPO must synchronize large gradient tensors each step, leading to communication bottlenecks that limit scaling efficiency.

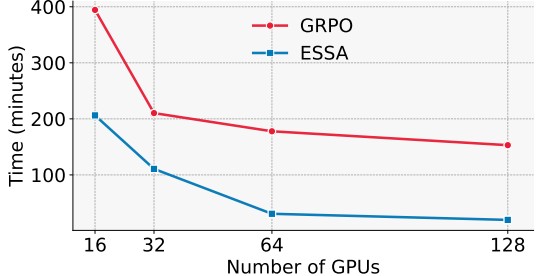

Figure 3: GRPO and ESSA scaling on **PRM800K** with **Qwen2.5-32B**: time to reach 0.835 accuracy vs. GPU count. ESSA: LoRA rank 16, pop. 128, batch size 256, $\alpha = 1.0$. GRPO: LoRA rank 16, lr $1 \times 10^{-5}$, global batch 512, mini batch 64.

We also conduct an experiment on a single GPU using the smaller Qwen2.5-3B model, where no parallelism is available. The result is presented in Appendix F.1. This experiment demonstrate that despite the limited hardware ESSA converges faster than GRPO in this setting as well, while the primary goal of this work is to introduce a method that utilizes multi-GPU resources more efficiently than gradient-based training.

## 4.3 PRECISION ANALYSIS

Because ESSA uses the model purely in inference mode, it can be trained even when the underlying model weights are quantized. We evaluate this capability on Qwen2.5-32B trained on PRM800K with LoRA rank 8, population size 64 and $\alpha = 1.0$. Table 2 reports the best validation accuracy achieved during training for three numerical precisions. Full training curves are provided in Appendix D.

| Precision | BFLOAT16 | INT8 | INT4 |
|---|---|---|---|
| Max ESSA accuracy | 0.847 | 0.844 | 0.838 |

Table 2: ESSA validation maximum accuracy as a function of **Qwen2.5-32B** weight precision on PRM800K. Settings: LoRA rank 8, pop. 64, $\alpha = 1.0$, batch size 256.

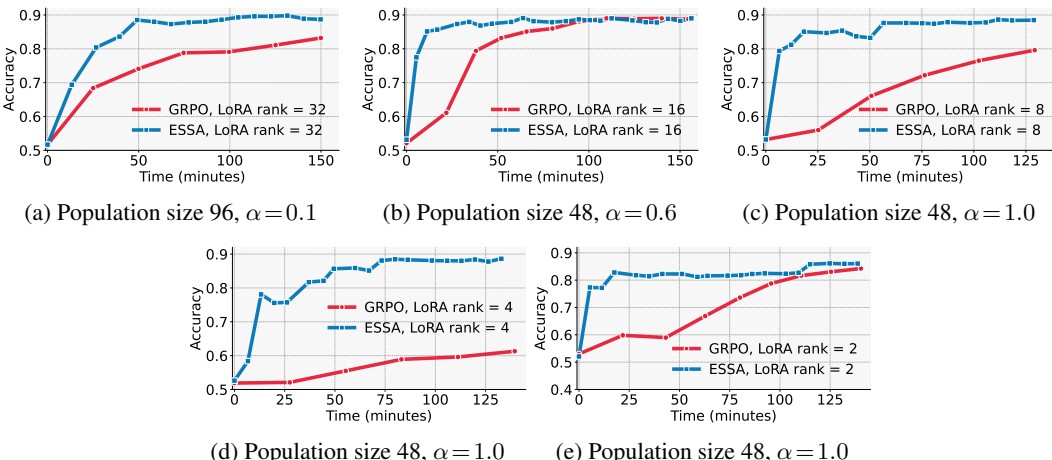

(a) Population size 96, $\alpha=0.1$     (b) Population size 48, $\alpha=0.6$     (c) Population size 48, $\alpha=1.0$

(d) Population size 48, $\alpha=1.0$     (e) Population size 48, $\alpha=1.0$

Figure 4: Validation accuracy over time on **GSM8K** with **Qwen2.5-Math-7B**. Panels (a)-(e) correspond to LoRA ranks 32, 16, 8, 4, and 2, respectively. ESSA (blue): batch size 100. GRPO (red): lr $1\times10^{-5}$, global batch 512, mini batch 64. ESSA rises rapidly and plateaus early across all ranks, while GRPO improves more gradually.

Although the maximum accuracy decreases
slightly as precision is reduced, the drop is minor (less than one percentage point from BFLOAT16 to INT4). This enables substantial savings in compute and memory: with INT4 quantization, a model as large as 72B parameters can fit on a single GPU for processing each population member, as demonstrated in our large-scale experiments later in the paper.

## 4.4 COMPARISON TO BASELINE (GRPO)

### 4.4.1 SCHOOL MATH

Figure 4 shows the validation accuracy versus wall-clock time for Qwen2.5-Math-7B on the GSM8K benchmark, comparing ESSA with GRPO across different LoRA ranks. Across all ranks, ESSA rises sharply during the first 10-20 minutes and reaches accuracies near $0.85$-$0.90$ significantly earlier than GRPO, which typically requires 60-100 minutes to approach the same level.

For moderate and large ranks (32, 16) both methods eventually converge to a similar final accuracy ($\approx 0.88$-$0.90$), but ESSA attains this plateau far sooner and with less fluctuation. At smaller ranks ESSA maintains accuracy close to its high-rank plateau, while GRPO lags behind for most of training. The trajectories demonstrate that ESSA is considerably less sensitive to LoRA rank: lowering the rank from 32 to 2 has only a mild effect on both early-time growth and final accuracy, whereas GRPO's convergence speed degrades markedly as rank decreases.

On school-level math reasoning tasks, ESSA consistently delivers a faster time-to-quality than GRPO across all LoRA ranks, making it well suited for rapid iteration or training under tight time budgets. For completeness, the same comparison performed with the Qwen2.5-7B model on GSM8K is reported in Appendix F.1. Those curves confirm the same pattern: ESSA consistently converges faster than GRPO while matching final accuracy.

### 4.4.2 BEYOND SCHOOL MATH

Advanced-math benchmarks require multi-step symbolic reasoning, long derivations, and careful numeric manipulation. To faithfully measure the benefit of alignment strategies we therefore employ large backbones, Qwen2.5-32B and Qwen2.5-72B, which (i) possess enough raw capacity to tackle these high-difficulty problems and (ii) let us observe how training methods scale with model size.

Figure 5 illustrate the validation accuracy over time for Qwen2.5-32B and Qwen2.5-72B, respectively. Across both models the curves show the same pattern as on school-math: ESSA rises sharply early, reaches its plateau considerably sooner than GRPO, and maintains that level with low variance.

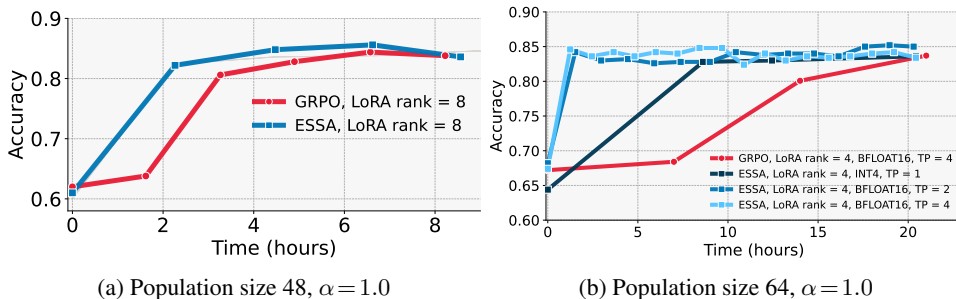

(a) Population size 48, $\alpha = 1.0$      (b) Population size 64, $\alpha = 1.0$

Figure 5: Validation accuracy over time on **PRM800K**. **Qwen2.5-32B** with LoRA rank 8 for both methods **(a)** and **Qwen2.5-72B** with LoRA rank 4 for both methods **(b)**. For Qwen2.5-72B we run ESSA under BFLOAT16 with tensor parallelism ($TP$): $TP = 2$ and $TP = 4$, and under INT4 with $TP = 1$, keeping the total GPU budget at 32 for both methods. ESSA (blue): batch size 256. GRPO (red): lr $1 \times 10^{-5}$, global batch 512, mini batch 64. Across both scales, ESSA reaches strong validation accuracy earlier and matches or exceeds GRPO throughout.

| Method | MATH500 | MinervaMath | OlympiadBench | AIME'24 | AIME'25 | AMC'23 | Avg |
|--------|---------|-------------|---------------|---------|---------|--------|-----|
| | | | **avg@k** | | | | |
| GRPO | 81.8 | 41.2 | 45.7 | 14.6 | 10.0 | 61.7 | 42.5 |
| ESSA | **82.1** | **41.8** | **47.6** | **17.3** | **12.1** | **63.3** | **44.0** |
| | | | **pass@8** | | | | |
| GRPO | **93.6** | **53.3** | 65.3 | 32.9 | **29.2** | **87.3** | 60.2 |
| ESSA | 92.6 | 52.2 | **66.4** | **35.3** | 28.0 | 86.9 | 60.2 |

Table 3: Results on advanced-math benchmarks with **Qwen2.5-32B**. Rows are grouped by metric: avg@k and pass@8. For **AIME'24**, **AIME'25**, and **AMC'23**, avg@k is reported at $k = 16$; for other benchmarks, $k = 8$. The final **Avg** averages across available entries.

| Method | MATH500 | MinervaMath | OlympiadBench | AIME'24 | AIME'25 | AMC'23 | Avg |
|--------|---------|-------------|---------------|---------|---------|--------|-----|
| | | | **avg@k** | | | | |
| GRPO | **83.0** | 43.0 | **49.1** | **18.8** | 11.0 | 62.2 | 44.5 |
| ESSA | 82.8 | **43.5** | 48.4 | 17.9 | **13.3** | **62.5** | **44.7** |
| | | | **pass@8** | | | | |
| GRPO | **94.2** | 51.8 | 66.7 | **37.5** | 23.5 | 85.6 | 59.9 |
| ESSA | 93.0 | **52.9** | **66.8** | 37.4 | **26.8** | **85.8** | **60.5** |

Table 4: Results on advanced-math benchmarks with **Qwen2.5-72B**. Rows are grouped by metric: avg@k and pass@8. For **AIME'24**, **AIME'25**, and **AMC'23**, avg@k is reported at $k = 16$; for other benchmarks, $k = 8$. The final **Avg** averages across available entries.

In total, we allocate 16 and 32 GPUs to the 32B and 72B models, respectively, across all configurations. In the BFLOAT16 precision regime the 72B model no longer fits on a single GPU, so we use tensor parallelism ($TP$) with size of 2 and 4. GRPO requires at least 4 GPUs for this configuration because both forward and backward passes must be distributed. In contrast, ESSA can operate with INT4 weights, allowing a full 72B model instance to reside on a single GPU and enabling one-candidate-per-GPU evaluation.

We observe that the BFLOAT16 model converges faster than the INT4 version. This happens because placing a 72B model in an INT4 representation leaves less space for KV-cache and INT4 matrix multiplications are slower on standard accelerators. Crucially, increasing tensor parallelism shows minimal impact on the accuracy growth trajectory in this experiment. This indicates that, even when the inference worker does not fit on a single device, that is, when the per-device evaluation rule is broken, ESSA remains efficient.

We also evaluate the trained models on the advanced-math benchmarks using both ESSA and GRPO. The detailed scores are reported in Table 3 for Qwen2.5-32B and in Table 4 for Qwen2.5-72B. For Qwen2.5-32B, ESSA improves or matches GRPO on avg@k across all benchmarks and shows comparable pass@8 and avg@k performance. For Qwen2.5-72B, where the tasks are especially challenging, ESSA again achieves competitive or better results: it delivers slightly higher pass@8 and avg@k.

### 4.4.3 INSTRUCTION FOLLOWING

We next evaluate ESSA in a domain that is qualitatively different from mathematics: instruction following.

Here the goal is to align the model to follow natural language instructions rather than to perform structured reasoning. We use the IFEval benchmark with LLaMA3.1-8B and fix the LoRA rank to 8, comparing ESSA and GRPO under identical data and initialization.

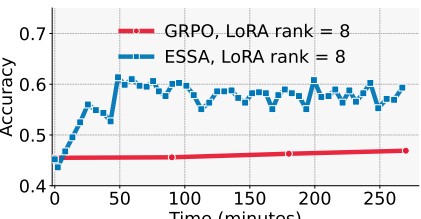

Figure 6 shows that ESSA reaches roughly $0.6$ accuracy within the first 60 minutes and maintains that level for the remainder of training. GRPO, in contrast, exhibits a much slower and steadier increase, saturating around $0.45$ even after more than four hours. Although the ESSA trajectory displays higher short-term variance, its early and sustained advantage demonstrates that gradient-free evolutionary optimization is effective even in open-ended, non-mathematical instruction-following tasks.

Figure 6: Validation accuracy over time on **IFEval** with **LLaMA-3.1-8B.** ESSA (blue): LoRA rank 8, pop. 24, batch size 500, $\alpha = 1.0$. GRPO (red): LoRA rank 8, lr $1 \times 10^{-5}$, global batch 512, mini batch 64. ESSA improves around $0.6$-$0.65$, while GRPO remains nearly flat near $0.45$ throughout training.

### 4.4.4 GENERAL-PURPOSE ASSISTANT SETUP

To evaluate alignment in a general-purpose assistant scenario, we employ a more sophisticated reward model rather than simple verifiable rewards. Specifically, we use the RLHFlow/ArmoRM-Llama3-8B-v0.1(Wang et al., 2024a)[1] reward model to provide nuanced, preference-based feedback signals.

The instruction prompts for this setting were drawn from the HelpSteer2 (Wang et al., 2024b) dataset, which contains diverse user instructions and serves as a strong benchmark for open-domain assistant alignment. ESSA and GRPO are both trained with this preference-based reward signal and evaluated under the same protocol to assess their ability to align large language models to general-purpose assistant behavior.

Figure 7 reports the validation reward as a function of wall-clock time for LoRA ranks 8, 16, and 32. Across all ranks, ESSA and GRPO achieve similar final reward levels, with their learning curves showing comparable overall trends.

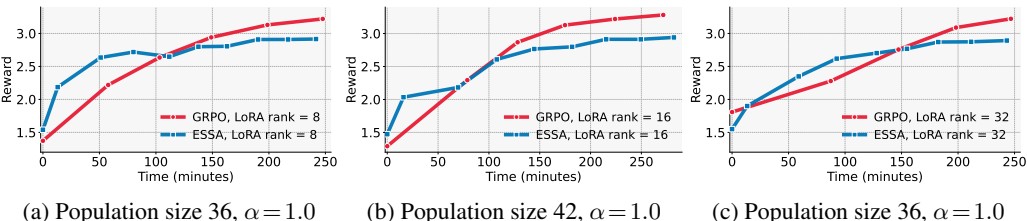

(a) Population size 36, $\alpha = 1.0$     (b) Population size 42, $\alpha = 1.0$     (c) Population size 36, $\alpha = 1.0$

Figure 7: Validation reward over time on **HelpSteer2** with **LLaMA-3.1-8B.** Panels (a)-(c) correspond to LoRA ranks 8, 16, 32 respectively. ESSA (blue): batch size 100. GRPO (red): lr $1 \times 10^{-5}$, global batch 512, mini batch 64. Both methods improve steadily, with ESSA showing faster early gains and GRPO slightly higher final reward. Overall, the plots indicate that ESSA achieves comparable alignment quality to GRPO with lower training complexity and similar convergence behavior.

---

[1]https://huggingface.co/RLHFlow/ArmoRM-Llama3-8B-v0.1

### 4.4.5 SVD-LoRA IN GRPO

In all previous experiments the LoRA factors $A$ and $B$ were fully trainable in GRPO, whereas ESSA optimized only the singular values after an SVD decomposition. To make the comparison strictly fair we repeat the GRPO baseline with the same restriction: only the singular values of the SVD of $A$ and $B$ are updated while their singular vectors were kept fixed, so that both methods have exactly the same number of trainable parameters. This SVD-GRPO variant is evaluated on Qwen2.5-7B with PRM800K, sweeping LoRA ranks 16,8,4,2 and keeping all other GRPO hyperparameters and the SFT initialization identical to the main baseline.

Figure 8 shows that SVD-GRPO struggles to learn effectively: even at rank 16 it plateaus around 0.5 accuracy and degrades further as the rank decreases. By contrast, ESSA with only rank 2 rapidly reaches about 0.72 accuracy and remains stable, outperforming SVD-GRPO by more than twenty percentage points despite operating in an equally low-dimensional parameter space.

This phenomenon can be explained. Deep networks typically exhibit dense, rotated curvature of the objective function, but in high-dimensional parameterizations such as full LoRA, these interactions are spread across many weakly correlated coordinates. Each parameter sees only a small fraction of the curvature. After SVD compression, the same curvature is concentrated into a very small subspace, and diagonal first-order optimizers like Adam fail to make progress. ES, by contrast, never computes gradients, its updates depend only on scalar rewards. Moreover CMA-ES adapts its covariance matrix in a way that aligns, in expectation, with the inverse Hessian of objective function.

Thus, for ESSA, moving from direct LoRA parameters to SVD-LoRA simply reduces dimensionality without making the search problem harder. For GRPO, however, the same reparameterization changes both the curvature and the gradient statistics in a way that standard first-order updates are not well adapted to.

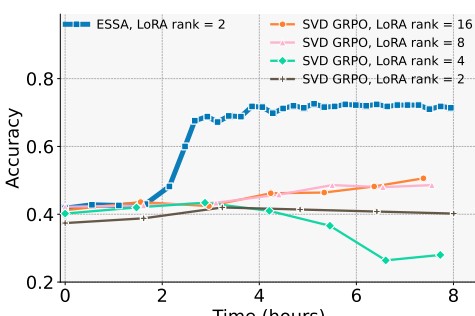

Figure 8: Validation accuracy over time on **PRM800K** with **Qwen2.5-7B**. ESSA (blue): LoRA rank 2, pop. 24, batch size 300, $\alpha = 1.0$. SVD-GRPO (only singular values are updated): LoRA ranks 16/8/4/2, lr $1 \times 10^{-2}$, global batch 512, mini batch 64. ESSA achieves $\approx 0.72$ accuracy while SVD-GRPO saturates at or below 0.5 even at LoRA rank 16.

## 5 DISCUSSION

ESSA shows that scalable and efficient LLM alignment is possible without gradients or backward passes, relying instead on inference-only evolutionary search in a compact, hardware-friendly parameter space. Across advanced math and instruction-following benchmarks, ESSA consistently matches or outperforms state-of-the-art gradient-based approaches such as GRPO, while delivering faster time-to-quality, greater robustness to hyperparameters, and dramatically reduced engineering complexity.

Our theoretical analysis further supports these empirical findings, demonstrating that ESSA's iteration time and parallel efficiency scale substantially better with model and cluster size, thanks to minimal synchronization and communication overhead. The ability to operate natively in low-precision (INT8/INT4) mode enables alignment of very large models – up to 72B parameters – using only a single GPU per candidate, with negligible accuracy loss. These results position evolutionary strategies, when paired with parameter-efficient adaptation, as a compelling alternative to classic RLHF pipelines – offering a simple, scalable, and broadly applicable framework for LLM alignment.

**Limitations & Future Work.** ESSA still depends on a decent SFT warm-start and is ultimately bounded by the expressivity of fixed-rank LoRA; with weak seeds or tiny ranks accuracy can plateau early. Very large populations also raise total FLOPs even though communication stays cheap. Future work will explore hybrid ES-gradient phases, adaptive rank expansion, and fully on-device / federated evolution in which edge GPUs or phones evaluate candidates and return only scalar rewards.

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

---

**Algorithm 1** ESSA: Distributed Evolutionary Search over LoRA Singular Values

---

**Require:** Training set $\mathcal{D}$, LoRA factors $A_i \in \mathbb{R}^{r \times n}$, $B_i \in \mathbb{R}^{m \times r}$ for all attention matrices, fraction $\alpha$ of top singular values to optimize, population size $P$, number of GPUs $N$, number of ES generations $E$.

1: **SVD initialization:**
2: **for** each LoRA factor $A_i, B_i$ **do**
3:     $A_i = U_{A_i} \Sigma_{A_i} V_{A_i}^\top$, $B_i = U_{B_i} \Sigma_{B_i} V_{B_i}^\top$
4:     Keep $U_{A_i}, V_{A_i}, U_{B_i}, V_{B_i}$ fixed; collect singular values $\sigma_{A_i} = \mathrm{diag}(\Sigma_{A_i})$, $\sigma_{B_i} = \mathrm{diag}(\Sigma_{B_i})$
5: **end for**
6: Concatenate the top $\alpha \times 100\%$ singular values of each $\sigma_{A_i}, \sigma_{B_i}$ into a single parameter vector $\sigma$.
7: Initialize $N$ workers with identical random seeds and a common CMA-ES state.
8: **for** $t = 0, 1, \ldots, E - 1$ **do**
9:     **for** each worker $k = 1, \ldots, N$ **in parallel do**
10:       Sample candidate perturbations of $\sigma$ using the shared random seeds.
11:       For each candidate:
        1.    Reconstruct $A_i', B_i'$ by adding perturbation vector to $\sigma$ and computing

$$A_i' = U_{A_i} \, \mathrm{diag}(\tilde{\sigma}_{A_i}) \, V_{A_i}^\top, \;\; B_i' = U_{B_i} \, \mathrm{diag}(\tilde{\sigma}_{B_i}) \, V_{B_i}^\top.$$

        2.    Update the model weights $W_i' = W_i + B_i' A_i'$.
        3.    Draw a random mini-batch $\mathcal{D}_{\mathrm{mini}} \subset \mathcal{D}$ of fixed size and compute the reward $F = F(W'; \mathcal{D}_{\mathrm{mini}})$.
12:     **end for**
13:     All workers exchange seeds and the corresponding rewards so that each worker knows the full set of evaluations.
14:     Each worker reconstructs the same perturbations from the shared seeds and performs one CMA-ES update of the search distribution.
15: **end for**
16: Reconstruct the final LoRA factors from the evolved singular values and return the aligned model.

---

# A   ESSA ALGORITHM

The overall procedure of ESSA is summarized in Algorithm 1, which outlines the main steps of the distributed evolutionary search over LoRA singular values.

# B   DETAILED THEORETICAL ANALYSIS

This appendix provides the complete derivation of the latency model summarized in Section 3.3. We first introduce the notation and expressions for per-iteration computation and communication time for both gradient-based methods and ESSA, then establish the optimal split of devices between training and generation, and finally prove a conservative bound on the ESSA population size under which ESSA is guaranteed to be faster.

**Notation.** Let $B^{\mathrm{grad}}$ be the global batch size used in gradient methods, $B^{\mathrm{essa}}$ the batch size processed by a single population instance, $b_{\mathrm{fb}}, b_{\mathrm{gen}}$ the microbatch sizes, $m_{\mathrm{fb}}, m_{\mathrm{gen}}$ GPUs per model instance for training and generation, $G$ total GPUs with $G_{\mathrm{fb}}, G_{\mathrm{gen}}$ allocated to each, $\tau_{\mathrm{fb}}(b), \tau_{\mathrm{gen}}(b)$ the forward-backward and generation microbatch times, $\eta_{\mathrm{fb}} = \tau_{\mathrm{fb}}(b_{\mathrm{fb}})/b_{\mathrm{fb}}$ and $\eta_{\mathrm{gen}} = \tau_{\mathrm{gen}}(b_{\mathrm{gen}})/b_{\mathrm{gen}}$ the per-sample times, and $k_{\mathrm{fb}}^{\mathrm{parallel}} = G_{\mathrm{fb}}/m_{\mathrm{fb}}$, $k_{\mathrm{gen}}^{\mathrm{parallel}} = G_{\mathrm{gen}}/m_{\mathrm{gen}}$ the numbers of microbatches processed in parallel. $T^{\mathrm{grad}} = T_{\mathrm{fb\text{-}gen}}^{\mathrm{grad}} + T_{\mathrm{comm}}^{\mathrm{grad}}$, $T^{\mathrm{essa}} = T_{\mathrm{gen}}^{\mathrm{essa}} + T_{\mathrm{comm}}^{\mathrm{essa}}$, with asynchronous training and generation so that $T_{\mathrm{fb\text{-}gen}}^{\mathrm{grad}} = \max\left(T_{\mathrm{fb}}^{\mathrm{grad}}, T_{\mathrm{gen}}^{\mathrm{grad}}\right)$.

**Computation.** Processing $B^{\text{grad}}$ samples by gradient training requires $B^{\text{grad}}/b_{\text{fb}}$ microbatches; $k_{\text{fb}}^{\text{parallel}}$ run in parallel:

$$T_{\text{fb}}^{\text{grad}} = \frac{B^{\text{grad}}}{b_{\text{fb}}\, k_{\text{fb}}^{\text{parallel}}}\, \tau_{\text{fb}}(b_{\text{fb}}) = \frac{B^{\text{grad}}\, m_{\text{fb}}}{G_{\text{fb}}}\, \eta_{\text{fb}}. \qquad T_{\text{gen}}^{\text{grad}} = \frac{B^{\text{grad}}\, m_{\text{gen}}}{G_{\text{gen}}}\, \eta_{\text{gen}}.$$

For ESSA, a population of $N_{\text{pop}}$ candidates is evaluated purely by generation:

$$T_{\text{gen}}^{\text{essa}} = \frac{N_{\text{pop}} B^{\text{essa}}\, m_{\text{gen}}}{G}\, \eta_{\text{gen}}.$$

**Communication.** Let $M_{\text{params}}$ be the model-parameter size (bytes). Gradient methods communicate gradients using all-reduce which consists of two collective operations: reduce-scatter and all-gather. Each moves a block of size $M_{\text{params}}/G$ across the $G - 1$ other devices. With effective interconnect peak bandwidth peak_bw (bytes/s) – the sustained per-GPU bandwidth for large collective messages:

$$T_{\text{comm}}^{\text{grad}} = 2 \cdot \frac{M_{\text{params}}(G-1)}{G\, \text{peak\_bw}}.$$

ESSA communicates only a random seed and the resulting reward, requiring a single all-gather of size $M_{\text{essa}} = 2 \times 4$ bytes:

$$T_{\text{comm}}^{\text{essa}} = \frac{M_{\text{essa}}(G-1)}{G\, \text{peak\_bw}}.$$

**Optimal device split.** Let $\theta \in (0,1)$ be the fraction of devices used for training ($G_{\text{fb}} = \theta G$, $G_{\text{gen}} = (1-\theta)G$). Then

$$T_{\text{fb-gen}}^{\text{grad}}(\theta) = \frac{B^{\text{grad}}}{G} \max\left( \frac{m_{\text{fb}}\eta_{\text{fb}}}{\theta}, \frac{m_{\text{gen}}\eta_{\text{gen}}}{1-\theta} \right).$$

**Lemma B.1** (Optimal split). *The minimum of $T_{\text{fb-gen}}^{\text{grad}}(\theta)$ over $\theta \in (0,1)$ is attained at*

$$\theta^\star = \frac{m_{\text{fb}}\eta_{\text{fb}}}{m_{\text{fb}}\eta_{\text{fb}} + m_{\text{gen}}\eta_{\text{gen}}}, \qquad T_{\text{fb-gen}}^{\text{grad}}(\theta^\star) = \frac{B^{\text{grad}}}{G}\big(m_{\text{fb}}\eta_{\text{fb}} + m_{\text{gen}}\eta_{\text{gen}}\big).$$

*Proof.* $\max(a/\theta, b/(1-\theta))$ is minimized when the two arguments are equal: $a/\theta = b/(1-\theta)$, with $a = m_{\text{fb}}\eta_{\text{fb}}$, $b = m_{\text{gen}}\eta_{\text{gen}}$. □

Define the ideal gradient iteration time (perfect scheduling):

$$T_\star^{\text{grad}} = T_{\text{fb-gen}}^{\text{grad}}(\theta^\star) + T_{\text{comm}}^{\text{grad}} = \frac{B^{\text{grad}}}{G}\big(m_{\text{fb}}\eta_{\text{fb}} + m_{\text{gen}}\eta_{\text{gen}}\big) + 2 \cdot \frac{M_{\text{params}}(G-1)}{G\, \text{peak\_bw}}.$$

Clearly $T^{\text{grad}} \geq T_\star^{\text{grad}}$.

**Theorem B.2** (ESSA iteration is faster under a conservative bound). *Suppose we only assume $m_{\text{fb}} \geq 1$, $m_{\text{gen}} \leq G$, and $2M_{\text{params}} - M_{\text{essa}} \geq M_{\text{params}}$. If the population size satisfies*

$$N_{\text{pop}} B^{\text{essa}} < B^{\text{grad}}\left( \frac{\eta_{\text{fb}}}{G\eta_{\text{gen}}} + 1 \right) + \frac{(G-1)M_{\text{params}}}{\text{peak\_bw}\, G\eta_{\text{gen}}}, \tag{1}$$

*then $T^{\text{essa}} < T_\star^{\text{grad}} \leq T^{\text{grad}}$.*

*Proof.* Starting from $T^{\text{essa}} < T_\star^{\text{grad}}$ and applying the bounds $m_{\text{fb}} \geq 1$, $m_{\text{gen}} \leq G$, $2M_{\text{params}} - M_{\text{essa}} \geq M_{\text{params}}$ to the exact inequality yields equation 1. □

**Discussion.** Theorem B.2 provides a conservative population-size threshold below which ESSA is guaranteed to be faster than the idealized gradient pipeline. Because real clusters rarely achieve the perfect split $\theta^\star$ and typically operate below the nominal peak_bw, the practical advantage of ESSA is often even larger than predicted by equation 1.

## C  SENSITIVITY TO HYPERPARAMETERS

### C.1  HEATMAPS

For completeness, Figures 9-12 present the full hyperparameter sensitivity for Section 4.1. These include Qwen2.5-7B on both GSM8K and PRM800K, Qwen2.5-Math-7B on PRM800K, as well as LLaMA-3.1-8B on IFEval.

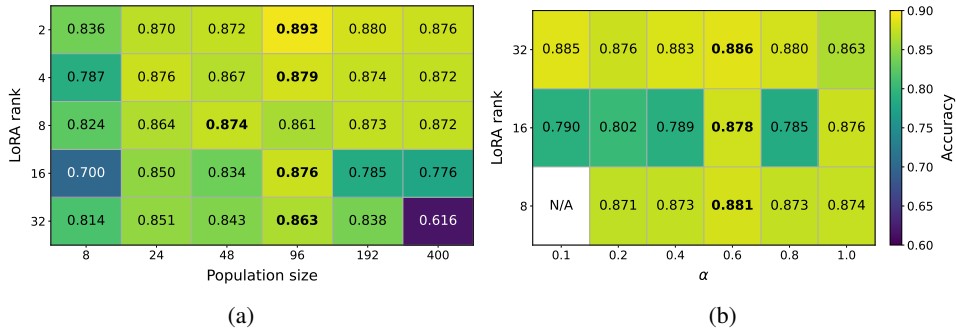

(a)                              (b)

Figure 9: Hyperparameter sensitivity of ESSA on **Qwen2.5-7B** for **GSM8K**. Batch size 100. **(a)** Accuracy when varying LoRA rank and population size. **(b)** For each LoRA rank, the population size is fixed to the best value found in (a), while the percentage $\alpha$ of trainable singular values is varied. This illustrates how ESSA performance depends jointly on adapter rank and the fraction of singular values optimized. The single white cell occurs because for LoRA rank 8 and $\alpha = 0.1$, rounding down yields zero trainable singular values, so no valid accuracy is reported.

Figure 9 examines the hyperparameter sensitivity of ESSA on Qwen2.5-7B for GSM8K. Varying the LoRA rank and population size shows that performance remains stable across a broad range, with accuracy peaking at 0.893 for LoRA rank 2 and population size 96. Notably, even very low-ranks (2-4) achieve top performance, while larger ranks yield diminishing or slightly degraded results. When fixing the population size and varying the fraction $\alpha$ of trainable singular values, ESSA maintains consistently high accuracy ($\approx$ 0.87-0.89) across all $\alpha$ values. This indicates strong robustness to the degree of SVD sparsification.

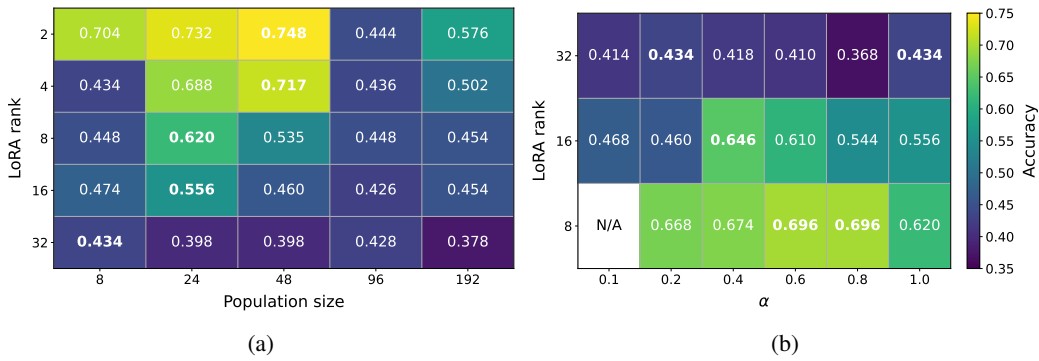

(a)                              (b)

Figure 10: Hyperparameter sensitivity of ESSA on **Qwen2.5-7B** for **PRM800K**. Batch size 300. **(a)** Accuracy when varying LoRA rank and population size. **(b)** For each LoRA rank, the population size is fixed to the best value found in (a), while the percentage $\alpha$ of trainable singular values is varied. This illustrates how ESSA performance depends jointly on adapter rank and the fraction of singular values optimized. The single white cell occurs because for LoRA rank 8 and $\alpha = 0.1$, rounding down yields zero trainable singular values, so no valid accuracy is reported.

Figure 10 analyzes the hyperparameter sensitivity of ESSA on Qwen2.5-7B for PRM800K. When varying LoRA rank and population size, accuracy peaks at 0.748 for LoRA rank 2 and population size 48, showing that small adapter ranks with moderate population sizes are most effective. Performance drops notably for larger ranks (16) or oversized populations (96), indicating diminishing

returns beyond a compact search space. With population size fixed, varying the fraction $\alpha$ of trainable singular values shows that moderate values ($\alpha \approx 0.4$-$0.8$) yield the best results – up to 0.696 accuracy for LoRA rank 8 – while very low or full updates slightly reduce performance. These results confirm that ESSA performs best with low-rank adapters, moderate population sizes, and partial singular value optimization.

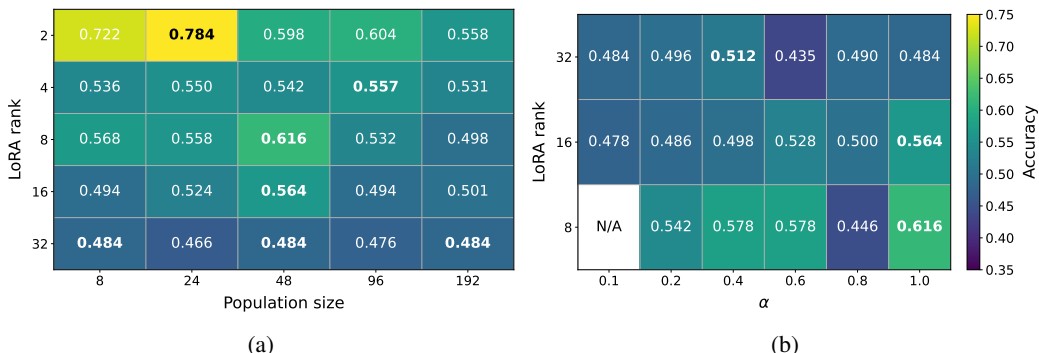

(a)                                                      (b)

Figure 11: Hyperparameter sensitivity of ESSA on **Qwen2.5-Math-7B** for **PRM800K**. Batch size 300. **(a)** Accuracy when varying LoRA rank and population size. **(b)** For each LoRA rank, the population size is fixed to the best value found in (a), while the percentage $\alpha$ of trainable singular values is varied. This illustrates how ESSA performance depends jointly on adapter rank and the fraction of singular values optimized. The single white cell occurs because for LoRA rank 8 and $\alpha = 0.1$, rounding down yields zero trainable singular values, so no valid accuracy is reported.

Figure 11 presents the hyperparameter sensitivity of ESSA on Qwen2.5-Math-7B for PRM800K. When varying LoRA rank and population size, accuracy peaks at 0.784 for LoRA rank 2 and population size 24, showing that compact low-rank adapters with small populations perform best. Higher ranks or very large populations lead to gradual accuracy degradation, indicating that excessive search dimensionality does not improve performance. Fixing population size and varying the fraction $\alpha$ of trainable singular values, ESSA achieves its highest score (0.616) at LoRA rank 8 and full update ($\alpha = 1.0$), while intermediate $\alpha$ values yield competitive but slightly lower results. The results highlight that ESSA performs optimally with lightweight low-rank configurations and remains robust to SVD sparsification, even on complex reasoning tasks.

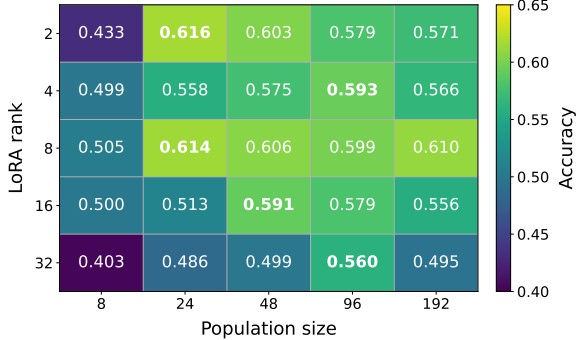

Figure 12: Hyperparameter sensitivity of ESSA on **LLaMA-3.1-8B** for **IFeval**. Accuracy when varying LoRA rank and population size.

Figure 12 shows the hyperparameter sensitivity of ESSA on LLaMA-3.1-8B for IFeval. Varying LoRA rank and population size reveals stable performance across configurations, with the highest accuracy (0.614) achieved at LoRA rank 8 and population size 24. Low ranks (4-16) yield comparable results, while very high-rank (32) adapters slightly underperform, suggesting a sweet spot around low-range ranks and moderate population sizes.

## C.2    TRAINING DYNAMICS

### C.2.1    QWEN2.5-7B ON GSM8K

Appendix Figures 13, 14 and 15 provide the complete training-dynamic study of ESSA on Qwen2.5-7B for GSM8K. This analysis shows how validation accuracy evolves over time as we vary the three key ESSA hyperparameters (Figure 9).

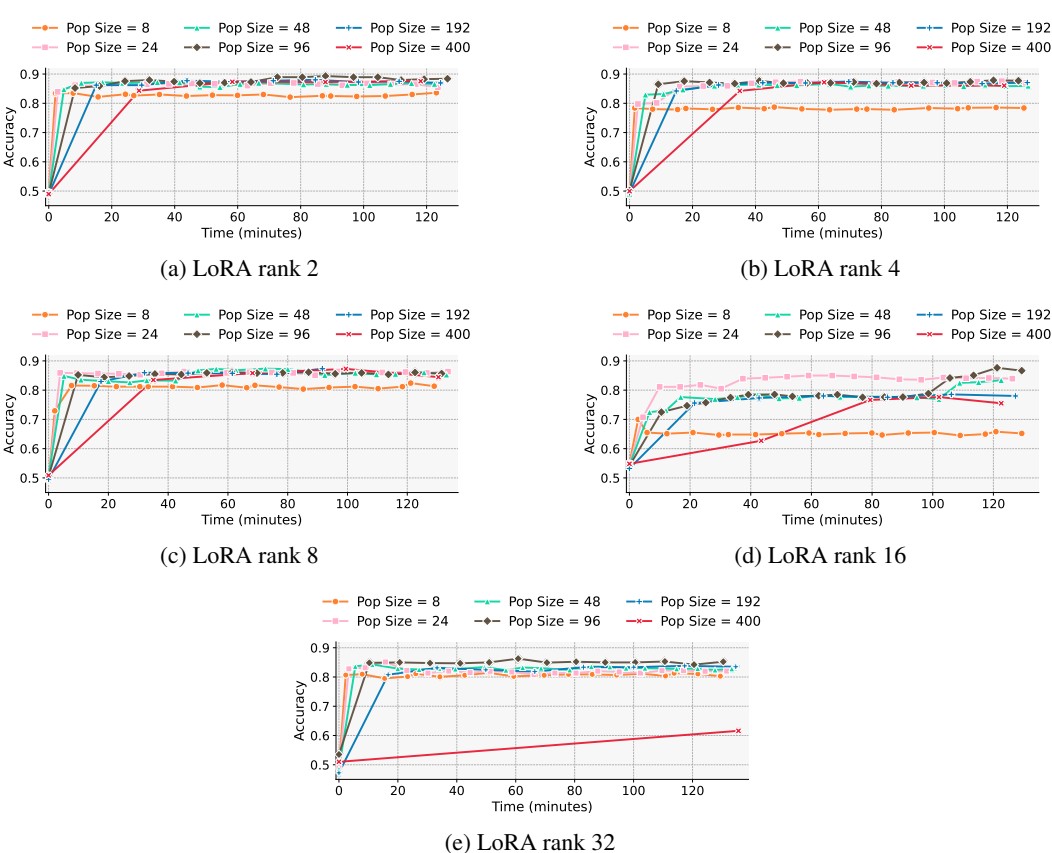

(a) LoRA rank 2

(b) LoRA rank 4

(c) LoRA rank 8

(d) LoRA rank 16

(e) LoRA rank 32

Figure 13: Validation accuracy over time on **GSM8K** with **Qwen2.5-7B** when varying the ESSA population size (8, 24, 48, 96, 192, 400) at fixed LoRA ranks (2, 4, 8, 16, 32). Batch size 100.

Figure 13 shows the full training dynamics of ESSA on GSM8K with Qwen2.5-7B when varying the ESSA population size.

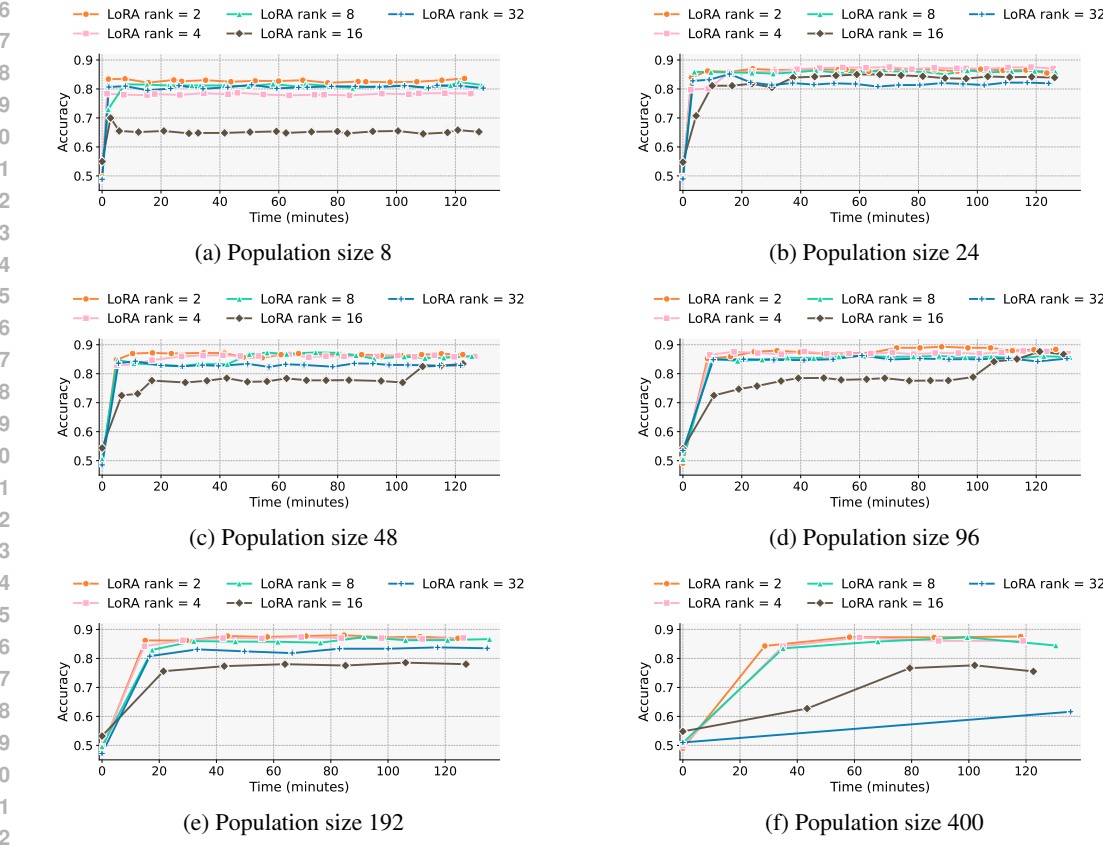

Figure 14: Validation accuracy over time on **GSM8K** with **Qwen2.5-7B** when varying the LoRA rank (2, 4, 8, 16, 32) at fixed population sizes (8, 24, 48, 96, 192, 400). Batch size 100.

Figure 14 shows the full training dynamics of ESSA on GSM8K with Qwen2.5-7B when varying the LoRA rank.

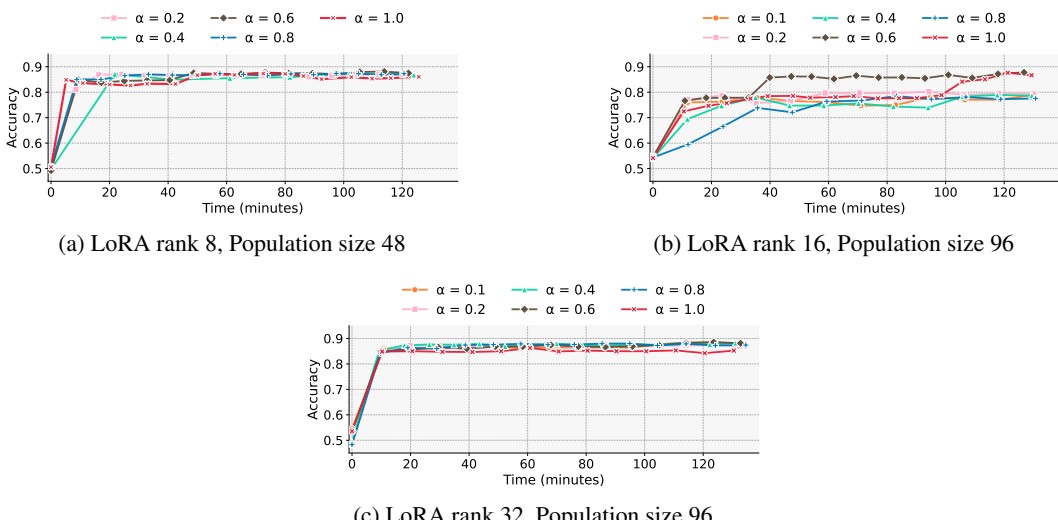

Figure 15: Validation accuracy on **GSM8K** with **Qwen2.5-7B** while varying $\alpha$. LoRA rank and population size are fixed to the optimal choices from Figure 9. Batch size 100.

Figure 15 shows the full training dynamics of ESSA on GSM8K with Qwen2.5-7B when varying the fraction $\alpha$ of trainable singular values.

### C.2.2 QWEN2.5-MATH-7B ON GSM8K

Appendix Figures 16, 17 and 18 provide the complete training-dynamic study of ESSA on Qwen2.5-Math-7B for GSM8K. These graphs complement the main text sensitivity analysis by showing how validation accuracy evolves over time as we vary the three key ESSA hyperparameters (Figure 2).

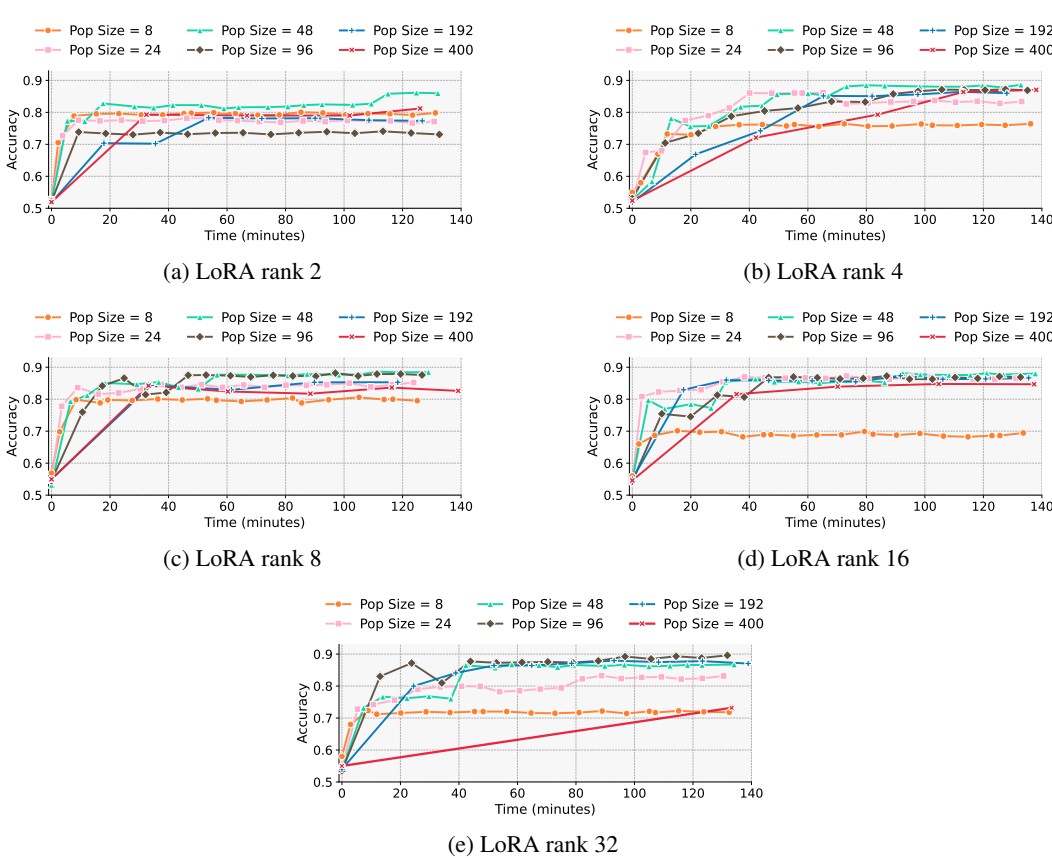

Figure 16: Validation accuracy over time on **GSM8K** with **Qwen2.5-Math-7B** when varying the ESSA population size (8, 24, 48, 96, 192, 400) at fixed LoRA ranks (2, 4, 8, 16, 32). Batch size 100.

Figure 16 shows the full training dynamics of ESSA on GSM8K with Qwen2.5-Math-7B when varying the ESSA population size.

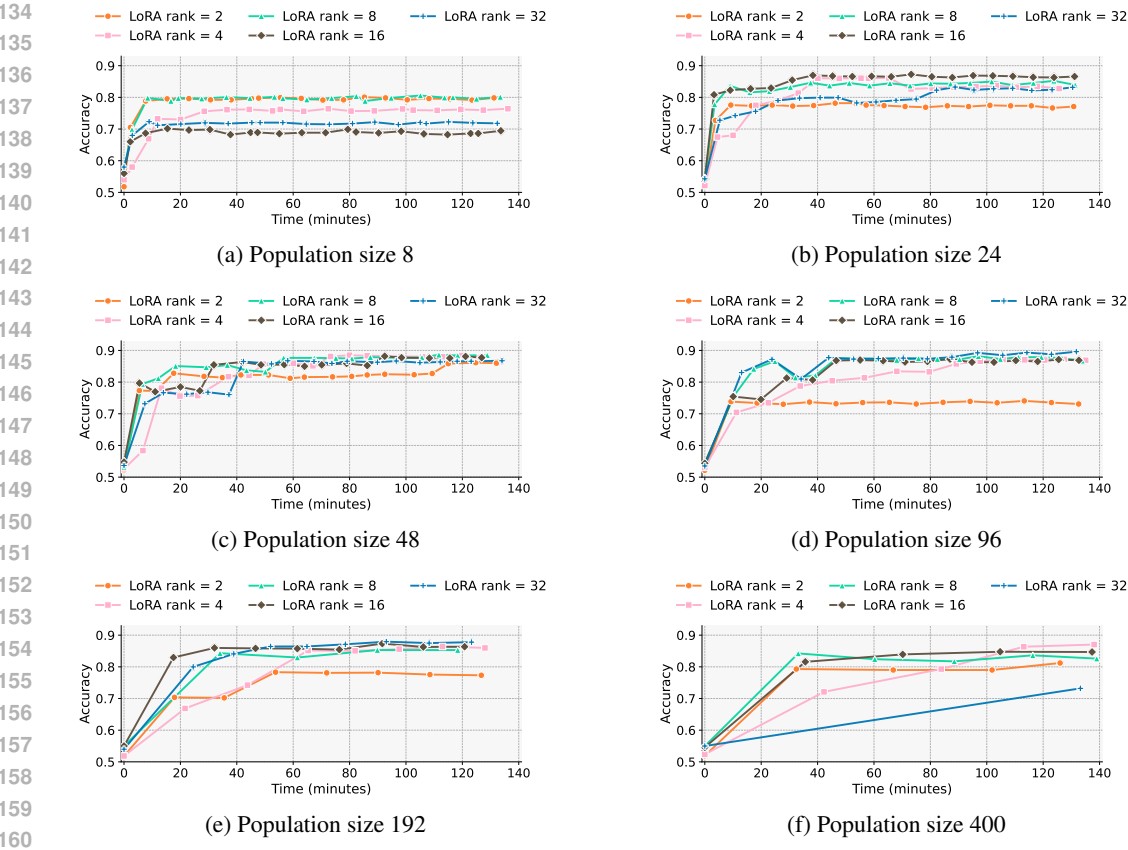

Figure 17: Validation accuracy over time on **GSM8K** with **Qwen2.5-Math-7B** when varying the LoRA rank (2, 4, 8, 16, 32) at fixed population sizes (8, 24, 48, 96, 192, 400). Batch size 100.

Figure 17 shows the full training dynamics of ESSA on GSM8K with Qwen2.5-Math-7B when varying the the LoRA rank.

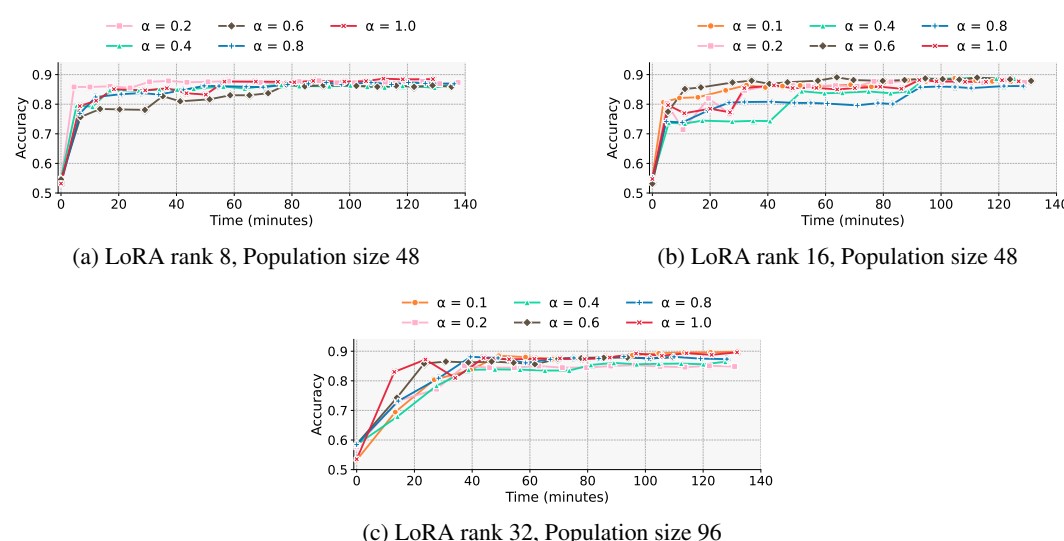

Figure 18: Validation accuracy on **GSM8K** with **Qwen2.5-Math-7B** while varying $\alpha$. LoRA rank and population size are fixed to the optimal choices from Figure 2. Batch size 100.

Figure 18 shows the full training dynamics of ESSA on GSM8K with Qwen2.5-Math-7B when varying the fraction $\alpha$ of trainable singular values.

### C.2.3 QWEN2.5-7B ON PRM800K

Appendix Figures 19, 20 and 21 provide the complete training-dynamic study of ESSA on Qwen2.5-7B for PRM800K. This analysis shows how validation accuracy evolves over time as we vary the three key ESSA hyperparameters (Figure 10)

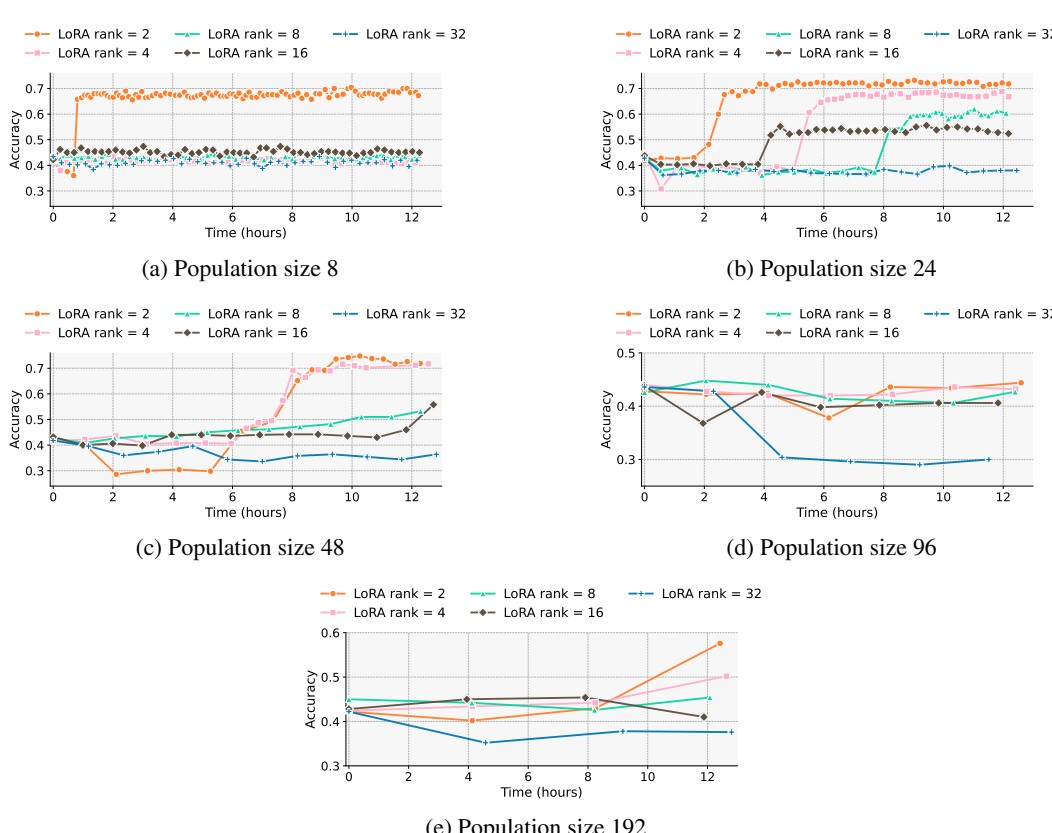

Figure 19: Validation accuracy over time on **PRM800K** with **Qwen2.5-7B** when varying the ESSA population size (8, 24, 48, 96, 192) at fixed LoRA ranks (2, 4, 8, 16, 32). Batch size 300.

Figure 19 shows the full training dynamics of ESSA on PRM800K with Qwen2.5-7B when varying the ESSA population size.

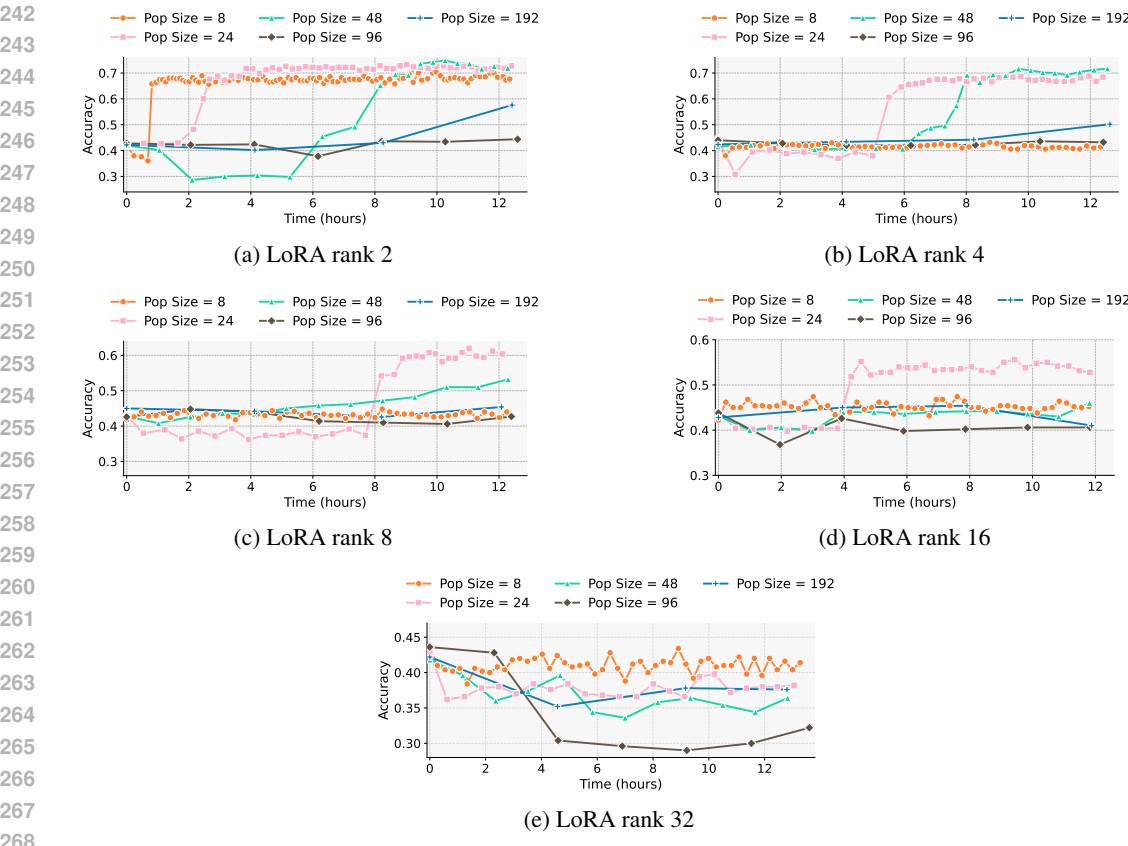

Figure 20: Validation accuracy over time on **PRM800K** with **Qwen2.5-7B** when varying the LoRA rank (2, 4, 8, 16, 32) at fixed population sizes (8, 24, 48, 96, 192). Batch size 300.

Figure 20 shows the full training dynamics of ESSA on PRM800K with Qwen2.5-7B when varying the LoRA rank.

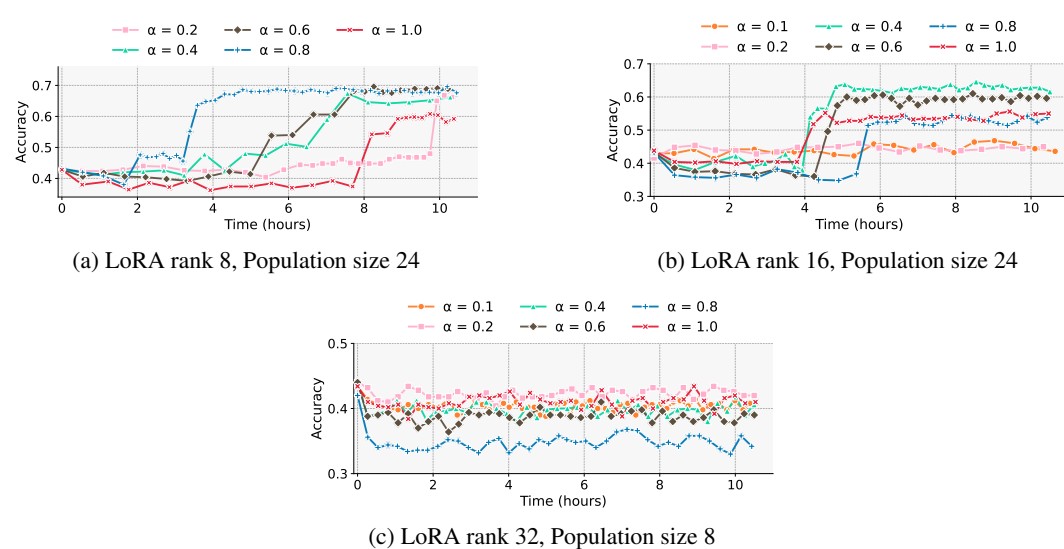

Figure 21: Validation accuracy over time on **PRM800K** with **Qwen2.5-7B** while varying $\alpha$. LoRA rank and population size are fixed to the optimal choices from Figure 10. Batch size 300.

Figure 21 shows the full training dynamics of ESSA on PRM800K with Qwen2.5-7B when varying the fraction $\alpha$ of trainable singular values.

### C.2.4 QWEN2.5-MATH-7B ON PRM800K

Appendix Figures 22, 23 and 24 provide the complete training-dynamic study of ESSA on Qwen2.5-Math-7B for PRM800K. These graphs complement the main text sensitivity analysis by showing how validation accuracy evolves over time as we vary the three key ESSA hyperparameters (Figure 11).

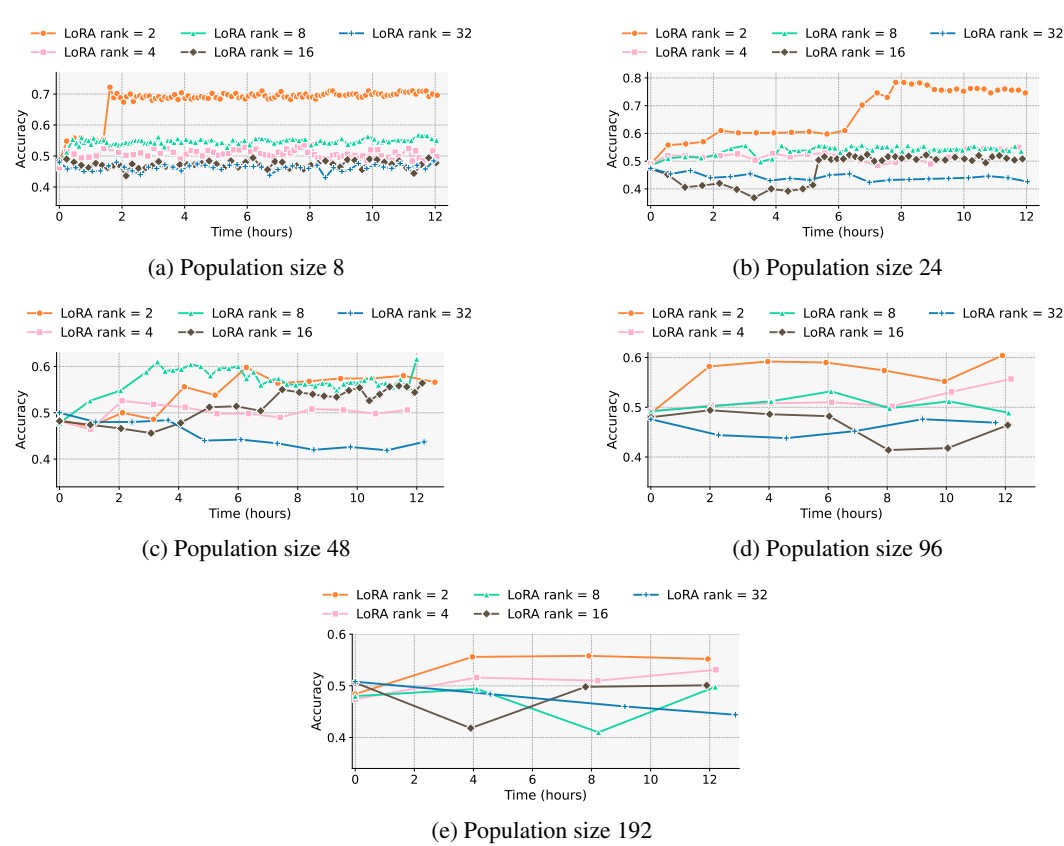

Figure 22: Validation accuracy over time on **PRM800K** with **Qwen2.5-Math-7B** when varying the ESSA population size (8, 24, 48, 96, 192) at fixed LoRA ranks (2, 4, 8, 16, 32). Batch size 300.

Figure 22 shows the full training dynamics of ESSA on PRM800K with Qwen2.5-Math-7B when varying the ESSA population size.

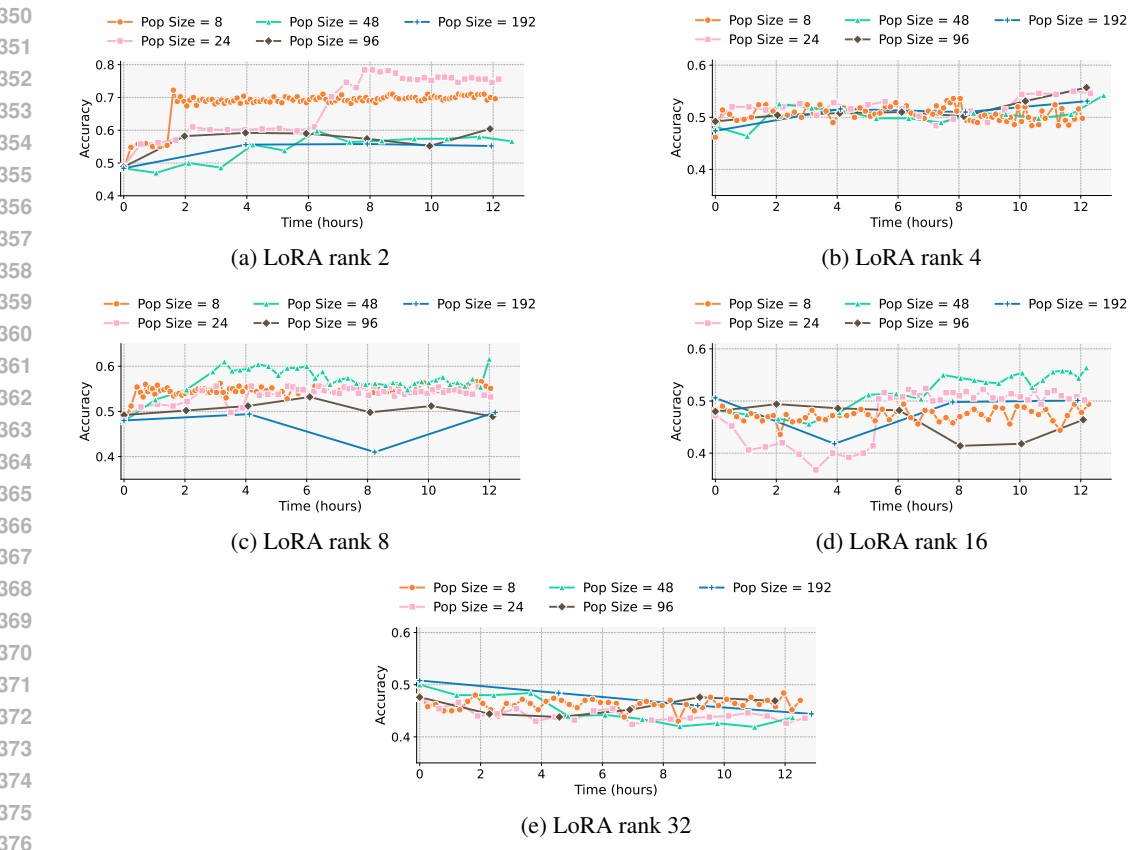

(a) LoRA rank 2

(b) LoRA rank 4

(c) LoRA rank 8

(d) LoRA rank 16

(e) LoRA rank 32

Figure 23: Validation accuracy over time on **PRM800K** with **Qwen2.5-Math-7B** when varying the LoRA rank (2, 4, 8, 16, 32) at fixed population sizes (8, 24, 48, 96, 192). Batch size 300.

Figure 23 shows the full training dynamics of ESSA on PRM800K with Qwen2.5-Math-7B when varying the LoRA rank.

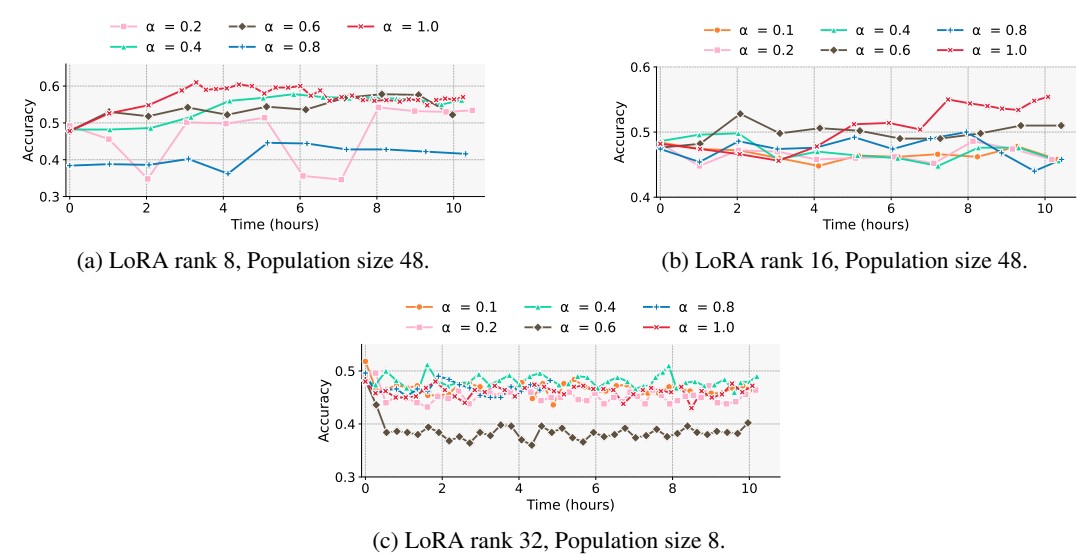

(a) LoRA rank 8, Population size 48.

(b) LoRA rank 16, Population size 48.

(c) LoRA rank 32, Population size 8.

Figure 24: Validation accuracy over time on **PRM800K** with **Qwen2.5-Math-7B** when varying $\alpha$. LoRA rank and population size are fixed to the optimal choices from Figure 11. Batch size 300.

Figure 24 shows the full training dynamics of ESSA on PRM800K with Qwen2.5-Math-7B when varying the fraction $\alpha$ of trainable singular values.

### C.2.5 LLaMA3.1-8B ON IFEval

Appendix Figures 25 and 26 provide the complete training-dynamic study of ESSA on LLaMA3.1-8B for IFEval. Two graphs complement the main text sensitivity analysis by showing how validation accuracy evolves over time as we vary the two key ESSA hyperparameters (Figure 12)

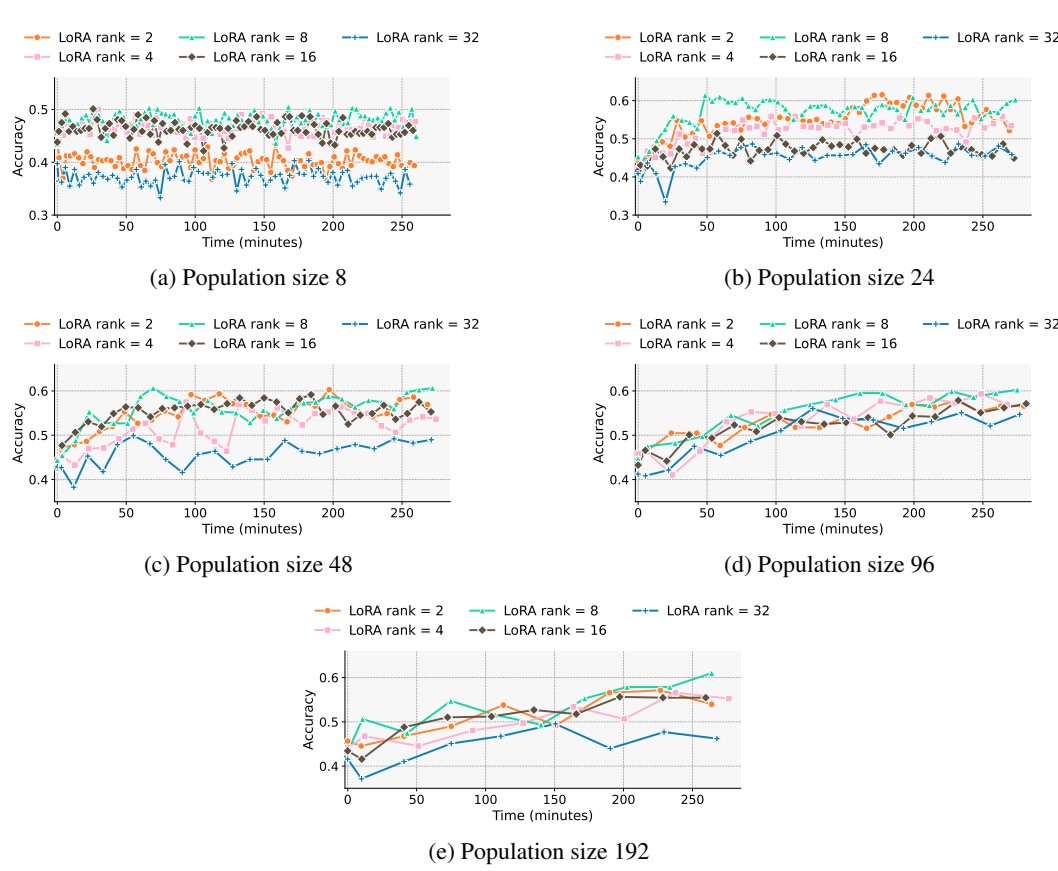

(a) Population size 8        (b) Population size 24

(c) Population size 48        (d) Population size 96

(e) Population size 192

Figure 25: Validation accuracy over time on **IFEval** with **LLaMA3.1-8B** when varying the ESSA population size (8, 24, 48, 96, 192) at fixed LoRA ranks (2, 4, 8, 16, 32). Batch size 500, $\alpha = 1.0$.

Figure 25 shows the full training dynamics of ESSA on IFEval with LLaMA3.1-8B when varying the ESSA population size.

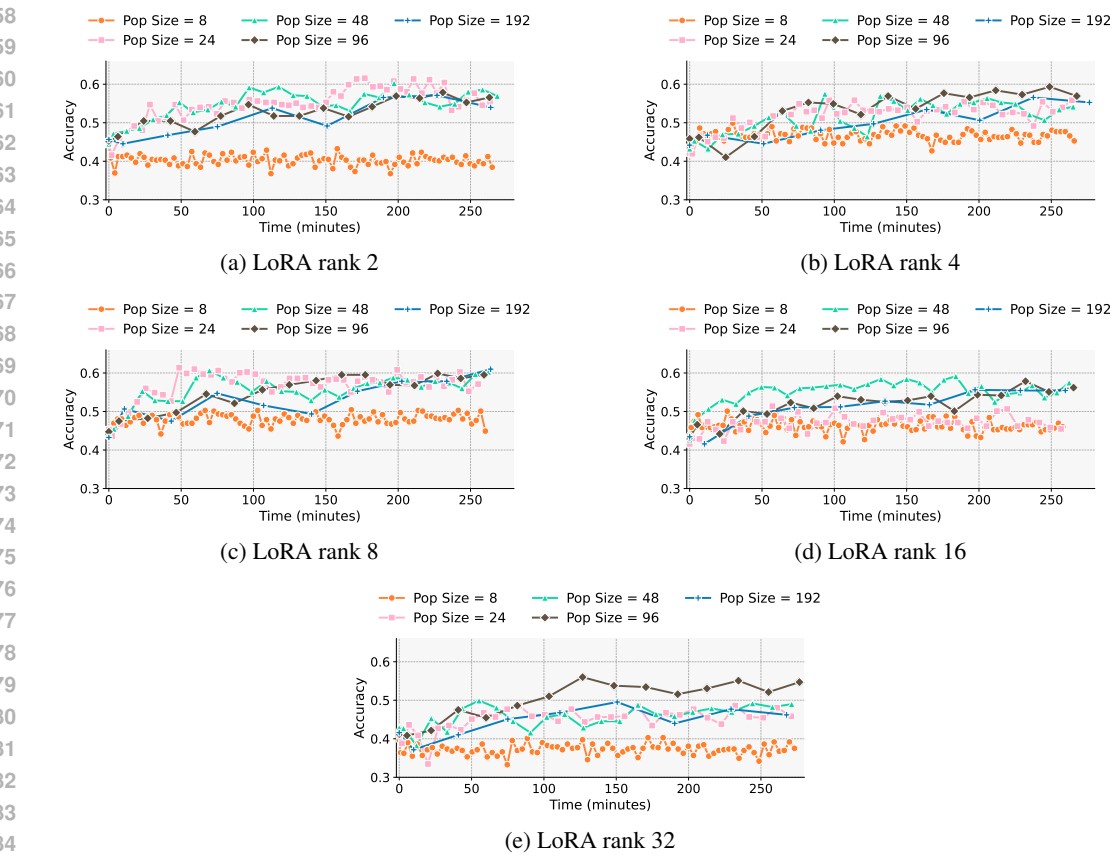

(a) LoRA rank 2

(b) LoRA rank 4

(c) LoRA rank 8

(d) LoRA rank 16

(e) LoRA rank 32

Figure 26: Validation accuracy over time on **IFEval** with **LLaMA3.1-8B** when varying the LoRA rank (2, 4, 8, 16, 32) at fixed population sizes (8, 24, 48, 96, 192). Batch size 500, $\alpha = 1.0$.

Figure 26 shows the full training dynamics of ESSA on IFEval with LLaMA3.1-8B when varying the LoRA rank.

## D   PRECISION ANALYSIS

Figure 27 shows the full training curves of ESSA when running Qwen2.5-32B on PRM800K with LoRA rank 8, population size 64, $\alpha = 1.0$, and per-candidate batch size 256, for three different numerical precisions: BFLOAT16, INT8, and INT4. Shaded areas indicate one standard deviation across runs.

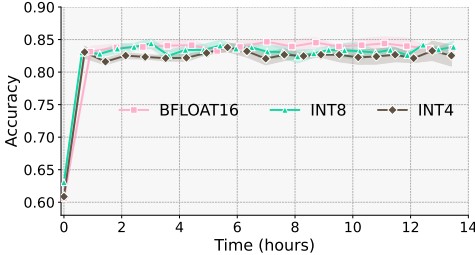

Figure 27: Validation accuracy over training time for Qwen2.5-32B on PRM800K under different weight precisions (BFLOAT16, INT8, INT4). Settings: LoRA rank 8, pop. 64, batch size 256, $\alpha = 1.0$.

Across all precisions, accuracy rises quickly within the first hour and then remains stable. Reducing precision from BFLOAT16 to INT8 and even to INT4 produces only a slight reduction in final accu-

racy, consistent with the summary in Table 2 of the main text. These results confirm that ESSA can safely exploit aggressive quantization to cut memory usage and computation cost while maintaining near-identical performance.

# E   TOY EXAMPLE ON MNIST

A natural concern arises when considering a singular-value-only parameterization. If we restrict the problem to simple linear or logistic regression, scaling the rows of a fixed low-rank update may seem too limited to support meaningful adaptation. In this setting only the singular magnitudes change while all directions remain fixed, which can give the impression of insufficient flexibility.

Motivated by this concern, we emphasize that ESSA's effectiveness in LLMs relies on the structured representations already present in the pretrained backbone and inherited through the downstream SFT initialization. To make this distinction concrete, we designed an MNIST experiment that replicates the LLM training pipeline as closely as possible within a minimal linear model.

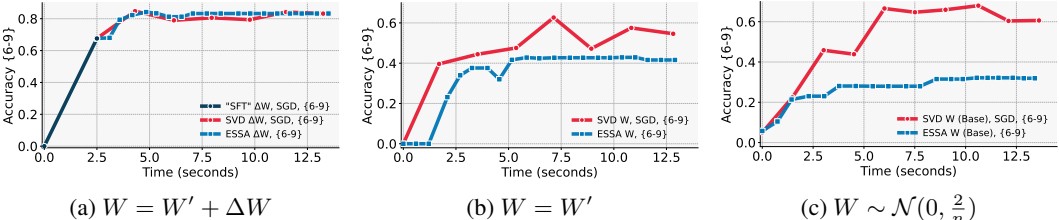

(a) $W = W' + \Delta W$          (b) $W = W'$          (c) $W \sim \mathcal{N}(0, \frac{2}{n})$

Figure 28: Validation accuracy over time on **MNIST** (on the test set of digits 6-9) with a single matrix $\mathbf{W} \in \mathcal{R}^{n \times m}$, $n = 784$, $m = 10$. Panels (a)-(c) correspond to $W = W' + \Delta W$, $W = W'$ and $W \sim \mathcal{N}(0, \frac{2}{n})$ setups respectively, where $W'$ is pretrained base weights. Train size 400. ESSA (blue): solution length 10, population size 24, batch size 64. SGD (red): lr 0.1, batch size 64.

Toy example on MNIST proceeds as follows:

1. At the pretrain stage (analog of LLM pretraining) we train a logistic regression model on MNIST digits 0-5 (train size 5000) using SGD optimizer. We obtain a "base model" $W = W'$, where $W'$ is the single matrix of size 784 by 10, that knows something about the data distribution, similar to LLM pretraining.

2. At the "SFT" stage (analogue of LoRA SFT) we take 400 samples of digits 6-9, initialize a matrix update $\Delta W$ with Kaiming init, and train only $\Delta W$ using SGD: $W = W' + \Delta W$, where $\Delta W$ is the single matrix of size 784 by 10.

3. At the last stage we decomposed $\Delta W = BSA$ and train only the singular values $S$ using ES or SGD optimizer. ESSA optimize singular values directly using train accuracy, while the SGD baseline optimized them through the standard cross-entropy loss.

In Figure 28a, ESSA follows the behavior of SGD on the MNIST task with high fidelity. This demonstrates that once the singular vectors encodes useful directions, modulating only the associated singular amplitudes is already sufficient to achieve effective adaptation. In Figure 28b, we remove the additive update entirely. We decompose the pretrained weights $W = W' = B'S'A'$, and train only the singular values $S'$ using ESSA and SGD. Even in this more constrained setup, ESSA achieved performance comparable to SGD. The pretrained matrix contained enough structure that rebalancing its singular directions remained a viable strategy for improving accuracy. Finally, in Figure 28c, we examine the case without pretraining or SFT, where the model is initialized with and the singular vectors carry no semantic information. In this setup ESSA improves only slowly, while SGD makes substantially faster progress. Without a structured subspace, changing only singular values is insufficient to emulate full learning dynamics.

These experiments demonstrate that the singular-value parameterization is effective precisely in the settings where LLMs operate: pretrained models with downstream SFT produce meaningful low-rank directions, and ESSA leverages this structure by adjusting their relative scaling. This behavior is consistent with the results observed in our large-scale LLM experiments.

## F    COMPARISON TO GRPO

### F.1    SCHOOL MATH

Appendix Figure 29 presents the results of Qwen2.5-7B on GSM8K in addition to Qwen2.5-Math-7B (Figure 4).

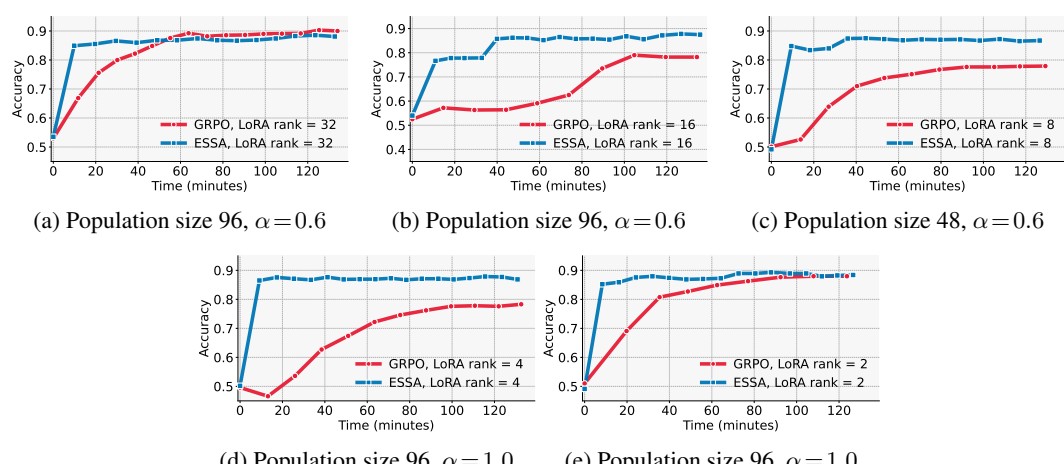

Figure 29: Validation accuracy over time on **GSM8K** with **Qwen2.5-7B**. Panels (a)-(e) correspond to LoRA ranks 32, 16, 8, 4, and 2, respectively. ESSA (blue): batch size 100. GRPO (red): lr $1 \times 10^{-5}$, global batch 512, mini batch 64.

Appendix Figure 34 presents the results of Qwen2.5-7B and Qwen2.5-Math-7B on GSM8K without SFT warm-start in addition to School Math experiments (Section 4.4.1). LoRA matrix $A$ is initialized using Kaiming initialization, matrix $B$ is zero.

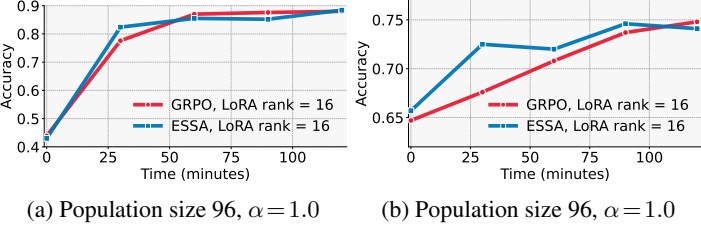

Figure 30: Validation accuracy over time on **GSM8K** with **Qwen2.5-7B** (a) and **Qwen2.5-Math-7B** (b) without SFT warm-start. ESSA (blue): train size 100. GRPO (red): lr $1 \times 10^{-5}$, global batch 512, mini batch 64.

Appendix Figure 31 presents the results of Qwen2.5-3B on one GPU in addition to School Math experiments (Section 4.2).

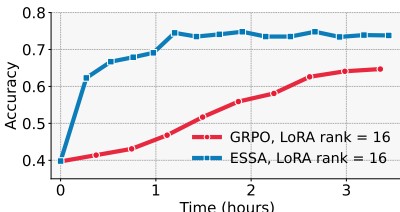

Figure 31: Validation accuracy over time on **GSM8K** with **Qwen2.5-3B** on one GPU. ESSA (blue): population size 48, $\alpha = 1.0$, batch size 100. GRPO (red): lr $1 \times 10^{-5}$, global batch 512, mini batch 64.

## F.2 BEYOND SCHOOL MATH

Appendix Figures 32 and 33 present the full results of our beyond school math experiments on PRM800K.

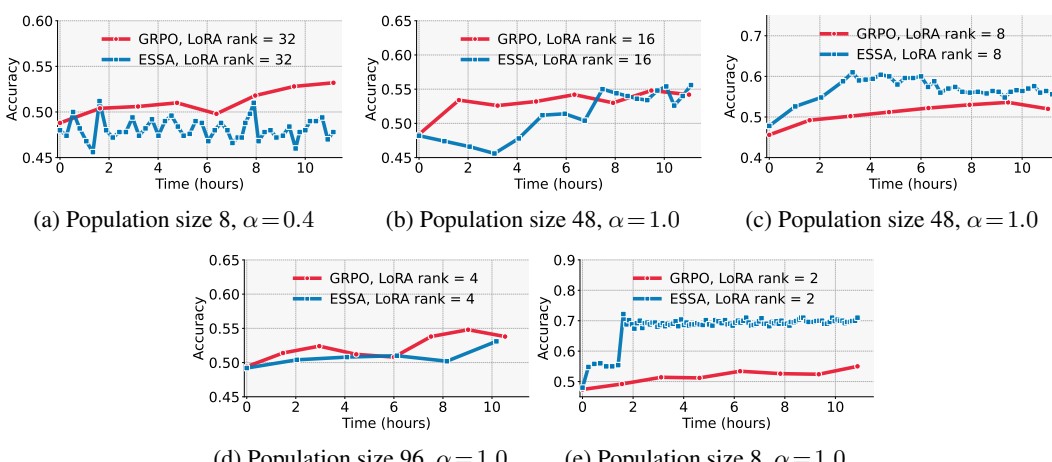

Figure 32: Validation accuracy over time on **PRM800K** with **Qwen2.5-Math-7B**. Panels (a)-(e) correspond to LoRA ranks 32, 16, 8, 4, and 2, respectively. ESSA (blue): batch size 300. GRPO (red): lr $1 \times 10^{-5}$, global batch 512, mini batch 64.

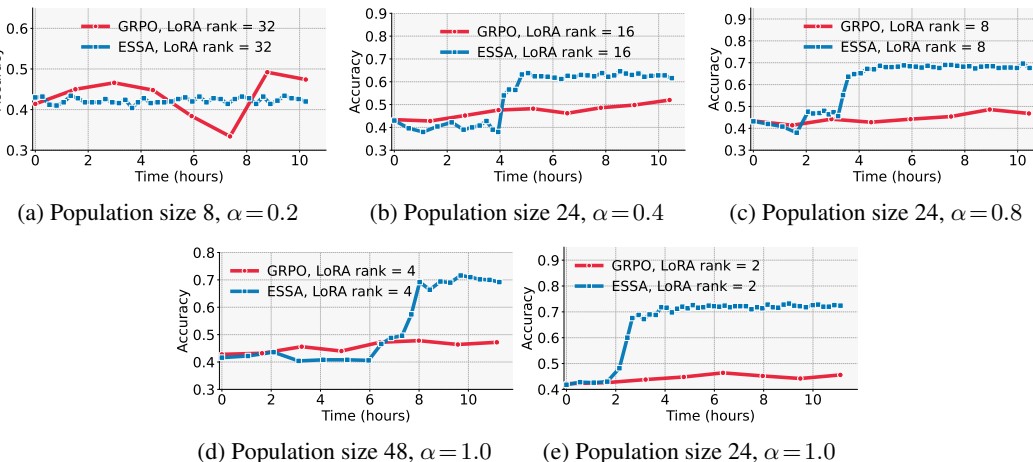

Figure 33: Validation accuracy over time on **PRM800K** with **Qwen2.5-7B**. Panels (a)-(e) correspond to LoRA ranks 32, 16, 8, 4, and 2, respectively. ESSA (blue): batch size 300. GRPO (red): lr $1 \times 10^{-5}$, global batch 512, mini batch 64.

We evaluate both Qwen2.5-Math-7B and Qwen2.5-7B, sweeping LoRA ranks and comparing ESSA with GRPO. Across both model variants on GSM8K (Section 4.4.1, Figure 29) and PRM800K (Section 4.4.2, Section F.2) ESSA climbs to high accuracy much more rapidly than GRPO and often reaches a higher plateau, particularly when the LoRA rank is small or moderate. In experiments without SFT (Figure 34) runs starting from a random LoRA achieve lower absolute accuracy. The relative performance between ESSA and GRPO remains consistent. Experiment with Qwen2.5-3B (Figure 31) on one GPU confirms that ESSA's benefits are not restricted to multi-GPU setups, but persist even when only one device is available.

## G  OOD Drift

To assess the robustness of ESSA under distribution shift, we conduct an additional set of experiments that instantiate exactly the cross-domain setting proposed in the review. We select two datasets with markedly different reward structures: HelpSteer2 and IFEval. HelpSteer2 encourages general-purpose response quality, while IFEval enforces rigid constraint satisfaction. Moving between them therefore offers a direct test of how well ESSA adapts when the SFT warm-start and the target alignment domain differ. All HelpSteer2 experiments use the same reward model as in the main paper (RLHFlow/ArmoRM-Llama3-8B-v0.1). All IFEval experiments use if-eval-like dataset as a train dataset.

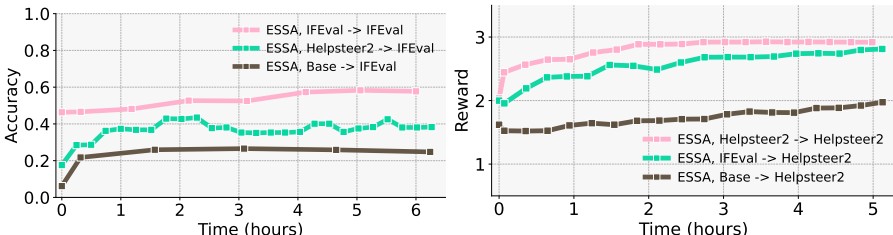

(a) LoRA rank 8, population size 48, $\alpha = 1.0$   (b) LoRA rank 8, population size 36, $\alpha = 1.0$

Figure 34: Validation metrics over time on **IFEval** (a) and **HelpSteer2** (b) with **Llama3.1-8B**. (a) IFEval → IFEval (pink), HelpSteer2 → IFEval (green), base model → IFEval (brown) experiments, batch size 500. (b): HelpSteer2 → HelpSteer2 (pink), IFEval → HelpSteer2 (green), base model → HelpSteer2 (brown) experiments, batch size 100.

For each target dataset B we evaluate three configurations. The first performs SFT on dataset A and ESSA on B (A → B). The second performs both SFT and ESSA on B (B → B). The third applies ESSA directly to B without any SFT warm-start (Base model → B), initialized with default LoRA parameters (Kaiming-uniform for weight A and zeros for weight B).

Across both directions, ESSA reliably improves performance on the target dataset, even when the SFT checkpoint originates from a mismatched domain. In the IFEval to HelpSteer2 condition the gap between cross-domain and in-domain SFT is modest, indicating that the structured initialization learned from if-eval-like dataset already provides useful axes for ESSA to exploit when optimizing for HelpSteer2. In the opposite direction the gap is larger, which is expected given that HelpSteer2 does not teach the model the strict formatting and constraint-following behavior required by IFEval. Nevertheless ESSA exhibits stable monotonic improvement and substantially surpasses the Base → IFEval configuration that starts from random LoRA parameters. In both directions, SFT warm-starts clearly outperform training from scratch, confirming that even an out-of-distribution SFT checkpoint supplies meaningful structure for ESSA's evolutionary search.

These results demonstrate that ESSA maintains its effectiveness under distribution shift and that the method can leverage any reasonable SFT initialization, even when it originates from a domain with incompatible reward structure.

## H  Implementation Details

### H.1  Datasets

| Dataset | SFT Train Size | ESSA/GRPO Train Size | Validation |
|---------|---------------|----------------------|------------|
| GSM8K | 1495 | 5978 | 1319 |
| PRM800K | 3600 | 7900 | 500 |
| IFEval | 1700 | 3414 | 541 |
| HelpSteer | 19000 | 1324 | 1024 |

Table 5: Dataset sizes across all experiments.

**System Prompts**

- **Math-Reasoning:** `Please reason step by step and place your final answer in \boxed{}.`
- **Instruction-Following:** `You are a helpful, honest, and concise assistant.`

## H.2 SFT HYPERPARAMETERS

| Model and Dataset | SFT Batch Size | Learning Rate | Epochs | SFT GPU Hours |
|---|---|---|---|---|
| Qwen2.5-7B (GSM8K) | 16 | $1 \times 10^{-4}$ | 1 | 0.18 |
| Qwen2.5-Math-7B (GSM8K) | 16 | $1 \times 10^{-4}$ | 1 | 0.16 |
| Qwen2.5-7B (PRM800K) | 16 | $5 \times 10^{-4}$ | 1 | 0.43 |
| Qwen2.5-Math-7B (PRM800K) | 16 | $5 \times 10^{-4}$ | 1 | 0.40 |
| Qwen2.5-32B (PRM800K) | 128 | $1 \times 10^{-6}$ | 3 | 9.80 |
| Qwen2.5-72B (PRM800K) | 128 | $1 \times 10^{-6}$ | 3 | 51.97 |
| Llama3.1-8B (IF-Eval-like) | 16 | $5 \times 10^{-4}$ | 1 | 0.50 |
| Llama3.1-8B (HelpSteer2) | 128 | $1 \times 10^{-6}$ | 3 | 3.74 |

Table 6: Summary of hyperparameters used for SFT across all experiments.

## H.3 ESSA HYPERPARAMETERS

For ESSA, we performed a hyperparameter grid search covering various values of LoRA rank, population size, fraction of singular values $\alpha$, batch size, and precision (Table 7). For each experiment, the best ESSA hyperparameters were selected based on validation performance (Table 8). We use up to 128 NVIDIA H100 80GB GPUs per experiment.

| Parameter | Values Tested | Default |
|---|---|---|
| LoRA Rank | 2, 4, 8, 16, 32 | 8 |
| Population Size | 8, 24, 36, 48, 64, 96, 192, 400 | 96 |
| Fraction of Singular Values ($\alpha$) | 0.1, 0.2, 0.4, 0.6, 0.8, 1.0 | 1.0 |
| Batch Size | 100, 256, 300, 500 | 100 |
| Precision | INT4, INT8, BFLOAT16 | BFLOAT16 |

Table 7: ESSA hyperparameter grid and default values for ablation studies.

| Experiment | LoRA Rank | Population Size | Batch Size | Fraction $\alpha$ |
|---|---|---|---|---|
| GSM8K (Qwen2.5-Math-7B) | 32 | 96 | 100 | 0.1 |
| GSM8K (Qwen2.5-7B) | 2 | 96 | 100 | 1.0 |
| PRM800K (Qwen2.5-Math-7B) | 2 | 24 | 300 | 1.0 |
| PRM800K (Qwen2.5-7B) | 2 | 48 | 300 | 1.0 |
| PRM800K (Qwen2.5-32B) | 8 | 48 | 256 | 1.0 |
| PRM800K (Qwen2.5-72B, INT4) | 4 | 64 | 256 | 1.0 |
| IFEval (LLaMA3.1-8B) | 8 | 24 | 500 | 1.0 |
| HelpSteer (LLaMA3.1-8B) | 8 | 36 | 100 | 1.0 |

Table 8: Best ESSA hyperparameters for each experiment.

