# OpenReview forum: "ESSA: Evolutionary Strategies for Scalable Alignment"
_ICLR.cc/2026/Conference — Submitted to ICLR 2026_

### Official Review · Reviewer_ojT5 · 2025-10-18

**Soundness:** 2
**Presentation:** 1
**Contribution:** 2
**Rating:** 4
**Confidence:** 3

**Summary:**

The paper proposed a gradient-free method ESSA for training LLMs. The main idea is to update the singular values of LoRA adapters via evolutionary optimization, which only requires forward pass and avoids gradient computation. The method is shown to be time-efficient and achieves better downstream performance compared to GRPO in the task of LLM finetuning on math and instruction following datasets.

**Strengths:**

* The method is time-efficient and computational-efficient, making it a potential approach for optimizing LLMs under the limited computational resource scenarios, e.g. edge devices.
* The algorithm is relatively easy to implement and does not require gradient computation.

**Weaknesses:**

* **The applicable scope of the method is limited.**
  * ESSA is strongly coupled with the setting of LoRA-based RLHF on a well-initialized SFT model.
  * If I understand the algorithm correctly, when the model cannot be fit into a single GPU, ESSA's speed will be severely affected since each GPU cannot run the evaluation independently.
  * The acceleration strongly depends on the number of available GPUs. When the computational resource is limited, e.g. only a single GPU is available, the acceleration seems to be diminishing.

* **The comparison between ESSA and GRPO seems unfair.** As mentioned in line 200-203, the ESSA is trained using INT4 while GRPO is trained using BFloat16; the later naturally requires more computational FLOPs and induces additional time. Have the authors compared two methods under the same computational data type?

* **The paper is poorly written; key definitions are missing.** There is no mathematical formulation of the algorithm, making it hard to interpret the Figure 1. For instance, it is unclear what is "layer n", "rank N" and "solution length". The explicit statement of the algorithmic update (i.e. CMA-ES algorithm) is missing as well. I suggest the authors to present a formal algorithmic description for a clearer presentation.

**Questions:**

* It seems that the evaluation data type is not aligned for ESSA and GRPO. According to Figure 7, the initial accuracy of both methods exhibit evident difference.

---

> ### Author Response · Authors · 2025-11-19
>
> Thank you for the thoughtful and detailed review, as well as for highlighting the strengths of our work. We appreciate the reviewer’s recognition of the method’s computational efficiency and practical relevance under constrained-resource settings. Your comments have helped us refine the clarity and positioning of the paper.
>
> ### **W1.1 LoRA and SFT**
>
> We would like to clarify that the use of LoRA in ESSA is not a limiting assumption but rather a practical and widely adopted training strategy in modern RLHF pipelines. Recent research has shown that LoRA can match its performance in many RL and reasoning-focused settings. For example, “LoRA Without Regret” (Thinking Machines) demonstrates that LoRA combined with RL attains quality comparable to fully trainable models [1], and Li et al., “LLMs Can Easily Learn to Reason from Demonstrations: Structure, Not Content, Is What Matters!” show that LoRA achieves reasoning performance on par with full tuning even in challenging tasks [2].
>
> To further support that ESSA is not fundamentally tied to a LoRA + SFT initialization, we conducted additional experiments where ESSA is applied without any SFT stage, starting directly from the base model on GSM8K. For both Qwen2.5-Math-7B https://8upload.com/display/cdc1600f9e8d63e1/16_grpo_essa_page-0001.jpg.php (anonymous link) and Qwen2.5-7B https://8upload.com/display/c1b047a50aec8b8d/16_grpo_essa_2_page-0001.jpg.php (anonymous link), ESSA reaches performance comparable to GRPO under this no-SFT setup. Nevertheless, the SFT-initialized configuration consistently performs better, which is expected, since without SFT the algorithm essentially optimizes the eigenvalues of a zero and randomly initialized LoRA matrices, making learning  unstable. Importantly, the SFT stage is lightweight: it requires minimal resources relative to RL and significantly stabilizes subsequent optimization [3, 4].
>
> ### **W1.2/W2 Clarifying the role of precision**
>
> Regarding the concern about ESSA using INT4 while GRPO is trained in BFLOAT16: in lines 200-202 we specifically use INT4 to highlight an important property of ESSA - its ability to operate reliably under constrained precision. This enables ESSA to train large models such as 72B parameters without additional communication or memory overhead, something gradient-based methods cannot achieve because their optimizer states must still be stored in BFLOAT16 or FP32. As shown in Figure 5(b), ESSA remains comparable to GRPO even at lower numerical precision.
>
> It is also important to emphasize that all main comparisons between ESSA and GRPO (Figures 4-8) use BFLOAT16 for both methods. INT4 appears only in two specific scenarios:
>
> 1. Section 4.3 (Precision Analysis), where we compare BF16 vs INT8 vs INT4 for ESSA only, to study ESSA’s robustness to low-precision inference.
> 2. The 72B-parameter backbone experiment (Figure 5(b)), where INT4 is required to fit the model on a single device; BFLOAT16 would not fit without tensor parallelism (TP).
>
> To directly address the fairness of comparisons and the applicability of ESSA in aligned-precision settings, we additionally trained Qwen2.5-72B on PRM800K in BFLOAT16. This setup also addresses your concern about scenarios where an inference worker does not fit on a single GPU. In this case the worker necessarily must be split across multiple devices, so we ran two variants using tensor parallelism over 2 (TP $= 2$) and 4 (TP $=4$) GPUs per worker. All other hyperparameters match Figure 5(b): $\mathrm{LoRA\\\_rank}=4$, $\mathrm{population\\\_size}= 64$, $\alpha = 1.0$, and 32 GPUs in total. The results are shown in https://8upload.com/display/e8d805f5204663c6/prm_72b__2__page-0001__2_.jpg.php (anonymous link).
>
> We observe that the BFLOAT16 model converges faster than the INT4 version. This is expected: fitting a 70B-scale model into INT4 leaves less space for KV-cache and INT4 matrix multiplications are slower on standard accelerators. Crucially, increasing tensor parallelism (2 vs. 4 GPUs per worker) shows minimal impact on the accuracy growth trajectory in this experiment. This indicates that, even when the inference worker does not fit on a single device, ESSA remains efficient.

---

> > ### Comment · Reviewer_ojT5 · 2025-11-23
> > **Follow-up Concerns and Questions**
> >
> > Thanks for the response, which clarifies some of my questions. I still have the following concerns and questions after reading the response.
> >
> > * ESSA with randomly initialized LoRA weight converges surprisingly fast and even aligns with GRPO. If I understand this correctly, what ESSA essentially does is to adjust the conditioning of a *random matrix* of rank 16. Could authors comment more on the effectiveness of ESSA under this setup?
> > * Following my previous concern, it would be beneficial to provide the convergence (reward) of the method with the singular values updated by gradient-based methods, which should serve as an upper bound of ESSA. Conducting this experiment helps justifying the effectiveness of updating singular values only and enhance the understanding of ESSA.
> > * The authors mentioned "all main comparisons between ESSA and GRPO (Figures 4-8) use BFLOAT16 for both methods." in the response, which is not aligned with what is stated in line 200-202 in the manuscript.

---

> > > ### Author Response · Authors · 2025-11-26
> > >
> > > As the rebuttal period is approaching its final week, we wanted to kindly follow up regarding our clarifications and the additional experiments we have already provided in response to your concerns.
> > >
> > > If there are any remaining questions or further experiments that would help solidify your assessment, we would be happy to run them within the remaining time. Please feel free to indicate anything that would be useful, we are fully committed to addressing all points thoroughly.
> > >
> > > If our current replies resolve your concerns, we would appreciate it if you could consider updating your score accordingly. Thank you for your time and consideration.

---

> ### Author Response · Authors · 2025-11-19
>
> ### **W1.3 Acceleration with limited resources**
>
> In the modern machine learning landscape, model sizes frequently exceed 7 billion parameters. For models of this scale and larger, the memory required for model training quickly surpasses the capacity of a single high-end GPU. As noted by Tazi et al., "As we can see, as soon as we reach 7B(!), the memory requirements for the weights, gradients, and optimizer states already start to add up significantly and exceed the size of a typical GPU's memory (e.g., 80 GB for an H100 GPU)" [5]. Our work focuses specifically on this common scenario, where gradient-based methods necessitate multiple GPUs by default for training. We demonstrate how the same number of GPUs can be utilized more efficiently with our evolutionary strategy-based approach.
>
> Nevertheless, to address the concern about performance under limited resources, we conducted an experiment on a single GPU, training Qwen2.5-3B on GSM8k. The results, available at https://8upload.com/display/d326eaff217b9ddf/essa_grpo_3b_page-0001.jpg.php (anonymous link), illustrate the difference in efficiency between methods in a resource-constrained setting and highlight ESSA's advantages.
>
> ### **W3 On Algorithmic Presentation**
>
> A complete pseudocode description of ESSA is provided in Appendix A, and the revised paper will present a fully formalized specification of all components that appear in Figure 1. To avoid ambiguity, several terms have been renamed and redefined.
>
> **ESSA.** To make Figure 1 self-contained, we introduce an updated version of the diagram (see anonymous link https://8upload.com/display/5a21b2d4714d0c4a/essa_device_page-0001.jpg.php), accompanied by a step-by-step explanation.
>
> - The previous term **“layer n”** is replaced with **“slice n”**, clarifying that these slices do **not** correspond to transformer blocks. Each slice  corresponds to a single LoRA attention matrix (e.g., $W_Q$, $W_K$, $W_V$, $W_O$) across all layers. The full CMA-ES candidate vector is partitioned into contiguous slices, each containing the  $2 \cdot \mathrm{LoRA\\\_rank}$ singular values for the LoRA factors  $A$ and $B$.
> - The previous **“rank N”** is renamed to **“device N”**, indicating the **GPU worker rank** in the distributed evaluation setup, not the LoRA rank or SVD rank. CMA-ES receives a seed at each rank, generates a population of size N+1 locally, evaluates a different candidate, and returns a reward.
>
>     For a worker index $i$, CMA-ES sampled candidate is:
>
>     $σ_i∈R^\mathrm{solution\\\_length},$ $i = 0, 1, … ,$ **$N,$**
>
>     where $N$ is the number of evaluation devices.
>
> - The term **“solution length”** denotes the dimensionality of the CMA-ES search vector - the concatenation of all SVD-LoRA singular values across all matrices and all layers. With number of layers ($\mathrm{num\\\_layers}$), number of training matrices per layer ($\mathrm{num\\\_matrices\\\_per\\\_layer}$):
>
>   $\mathrm{solution\\\_length} = \mathrm{num\\\_layers} \cdot \mathrm{num\\\_matrices\\\_per\\\_layer} \cdot \mathrm{LoRA\\\_rank} \cdot 2.$
>
>     Since $\mathrm{num\\\_matrices} = \mathrm{num\\\_layers} \cdot \mathrm{num\\\_matrices\\\_per\\\_layer},$
>
>     $\mathrm{solution\\\_length} = \mathrm{num\\\_matrices} \cdot \mathrm{LoRA\\\_rank} \cdot 2$
>
>
> **CMA-ES.** For clarity, we expand the definition of the underlying optimizer. CMA-ES maintains a multivariate normal search distribution updated at every generation. At iteration $i$, it samples $λ≥2$ (population size) candidate vectors:
>
> $x^{(k)}\_{i+1} ∼ m\_i + σ_i\mathcal{N}(0, C\_i)$ for  $k = 1, . . . , λ,$ so that $x\_{i+1}^{(k)} \sim \mathcal{N}(m\_i,\sigma\_i^2C\_i).$
>
> Here:
>
> - $m_i \in \mathcal{R}^{n}$  — current mean of the search distribution,
> - $σ_i \in \mathcal{R}>0$ — “overall” standard deviation,
> - $C_i \in \mathcal{R}^{n×n}$ — covariance matrix encoding anisotropic search directions; up to the scalar factor $σ_i^{2}$, $C_i$ is the covariance matrix of the search distribution,
> - $n=\mathrm{solution\\\_length}$.
>
> After evaluating all offspring, CMA-ES ranks them and updates $m\_i$, $\sigma\_i$, and $C\_i$ according to the standard adaptation rules described in Hansen’s tutorial [6]. This allows the search distribution to gradually align itself with beneficial directions in the objective landscape.
>
> ### **Q1 On the Initial Accuracy Difference**
>
> In Figure 7 ESSA and GRPO are initialized from the identical SFT LoRA checkpoint and are evaluated under the same computational data type. Their initial evaluation points differ due to sampling nondeterminism and inference-level variability [7].

---

> ### Author Response · Authors · 2025-11-19
>
> [1] Schulman et al, LoRA Without Regret
>
> [2] Li et al, LLMs Can Easily Learn to Reason from Demonstrations Structure, not content, is what matters!
>
> [3] Allal et al, The Smol Training Playbook: The Secrets to Building World-Class LLMs
>
> [4] Ouyang et al, Training language models to follow instructions with human feedback
>
> [5] Tazi et al., The Ultra-Scale Playbook: Training LLMs on GPU Clusters
>
> [6] Hansen, The CMA Evolution Strategy: A Tutorial
>
> [7] He et al, Defeating Nondeterminism in LLM Inference

---

> ### Author Response · Authors · 2025-11-19
>
> Dear Reviewer, thank you again for your thoughtful and detailed feedback; in our response we have provided substantial clarifications, additional experiments, and deeper explanations that directly address your concerns regarding applicability, fairness of comparisons, algorithmic clarity, and evaluation setup, since these clarifications significantly strengthen the understanding of our contribution and resolve points that may have influenced the initial score, we kindly ask you to consider raising your rating to reflect this improved understanding of the work.

---

> ### Author Response · Authors · 2025-11-24
>
> We thank the reviewer for the follow-up questions and for carefully examining our additional experiments. We appreciate the opportunity to further clarify these points.
>
> ### **Q1. Why does ESSA converge so quickly from a randomly initialized LoRA?**
>
> In the “BASE → ESSA’’ setup, the LoRA factors follow the standard initialization scheme: matrix $A$ is initialized using Kaiming initialization, matrix $B$ is zero. ESSA does not optimize the LoRA matrices directly; instead, it optimizes only the singular values of their SVD factorization. Geometrically, this restricts ESSA to rescaling a fixed set of directions without being able to rotate or change them. This is a very limited search space.
>
> Despite this limitation, ESSA still improves the base model because Qwen2.5 already contain extensive general mathematical prior. GSM8K requires only school-level arithmetic and shallow reasoning patterns, which Qwen-7B already partially possesses. ESSA, even with its restricted parameterization, can reinforce these latent capabilities and align the output format. This explains why the method still converges even though the underlying LoRA basis is random and not tailored to the task.
>
> To show that this effect depends on the inherent structure of the backbone rather than on ESSA itself, we performed an additional experiment on **Llama-3.1-8B** using **IFEval**, a task where the base model has much weaker alignment with the target behavior. Using LoRA rank 8, population 96, and 6 hours of ESSA training, the results were:
>
> | Method | Score (mean ± std) |
> | --- | --- |
> | **Base only** | 0.062 ± 0.001 |
> | **Base → ESSA** | 0.237 ± 0.011 |
> | **SFT only** | 0.455 ± 0.001 |
> | **SFT → ESSA** | 0.589 ± 0.008 |
>
> The accuracy–time plot (https://8upload.com/display/2c3bf563568c6b04/essa_sft_base.jpg.php anonymous link) illustrates the same dynamic: ESSA improves a raw base model, but only modestly, and performance quickly plateaus. In contrast, SFT → ESSA continues to climb steadily and reaches much higher accuracy. This confirms our interpretation: SFT provides a meaningful low-rank structure for the LoRA subspace, and ESSA is effective precisely because it refines this structured space. Starting from a random LoRA basis offers only limited gains, as shown by the BASE → ESSA curve. This is exactly why, throughout our main experiments, ESSA is always applied on top of an SFT-initialized model: the SFT stage supplies the meaningful low-rank geometry that ESSA builds upon.
>
> ### **Q2. Gradient-based optimization of SVD parameters**
>
> We want to emphasize that the experiment you suggests has already been included in the paper. Section 4.4.5 presents a GRPO variant in which the optimization is restricted to the singular values of the SVD decomposition of the LoRA factors, exactly matching the ESSA parameterization.
>
> Figure 8 reports the corresponding convergence curves. Despite having the same number of trainable parameters as ESSA, SVD-GRPO fails to make meaningful progress: even at LoRA rank 16 it saturates near 0.5 accuracy, and performance degrades for lower ranks. ESSA, using only rank 2, reaches approximately 0.72 accuracy and remains stable.
>
> We provide a theoretical explanation of this effect in our response to Reviewer 5kYU, and we added a concise summary of the reasoning directly into Section 4.4.5 of the revised paper. After SVD compression, curvature becomes concentrated in a very low-dimensional subspace. First-order optimizers such as Adam operate with diagonal preconditioning and cannot adapt to the rotated curvature geometry in this parameterization. Evolutionary strategies, and CMA-ES in particular, do not rely on gradients and their covariance adaptation tracks the inverse Hessian statistically, making the SVD parameterization natural rather than detrimental.
>
> ### **Q3. Precision setup**
>
> The statement in our response does not contradict the text in lines 200–202. In the original version, lines 200–202 specify that the 72B ESSA run uses INT4, while all GRPO baselines and all other model sizes use BF16 unless otherwise stated. This is fully consistent with our response, which stated that all main ESSA-GRPO comparisons (Figures 4–8) use BFLOAT16, and that the only case where ESSA operated in INT4 while GRPO used BFLOAT16 was the 72B PRM800K experiment (Figure 5). In the revised version we supplemented this setting with additional runs (see https://8upload.com/display/e8d805f5204663c6/prm_72b__2__page-0001__2_.jpg.php , anonymous link) where ESSA is also evaluated in BFLOAT16, ensuring full consistency in the precision configuration.
>
> To remove any possible ambiguity, we revised the corresponding paper sentence to make the default-precision rule explicit:
>
> > “We use standard BFLOAT16 precision unless otherwise specified. Qwen2.5-72B is also trained under INT4 for per-device evaluation in ESSA.”

---

> > ### Comment · Reviewer_ojT5 · 2025-11-26
> >
> > Thanks for the response.
> >
> > I would like to challenge the following argument:
> > > ESSA, even with its restricted parameterization, can reinforce these latent capabilities and align the output format. This explains why the method still converges even though the underlying LoRA basis is random and not tailored to the task.
> >
> > Consider the simple logistic regression problem: $\min - \sum_i \log \text{softmax}(Wx_i)_{y_i}$. Let $W=W' + BSA$, where $W'$ is a frozen weight, $BSA$ is the factorization of a LoRA block with $S$ being trainable, which matches the setup of ESSA. Let $B=I$ without loss of generality, we have $WX=W'X + S(AX)$. Evidently, the row-wise scaling applies to all samples $X$ *equivalently*. Given that $W'$ and $A$ are frozen, the learning is ineffective. If the problem modeling cannot even effectively solve the simple LR problem, I cannot be convinced that it can boost the performance of LLMs and even surpasses GRPO.
> >
> > I also expect additional elaborations on the following statement:
> > > First-order optimizers such as Adam operate with diagonal preconditioning and cannot adapt to the rotated curvature geometry in this parameterization. Evolutionary strategies, and CMA-ES in particular, do not rely on gradients and their covariance adaptation tracks the inverse Hessian statistically, making the SVD parameterization natural rather than detrimental.
> >
> > In particular, could authors comment more on why the evolution strategy is more effective than first-order methods in solving the problem from the perspective of optimization? Is it because it uses second information? If so, how exactly?

---

> > > ### Author Response · Authors · 2025-11-28
> > >
> > > Thank you for the follow-up questions, they are very helpful, and we appreciate the chance to clear up the remaining confusion. We also ran the experiments you suggested.
> > >
> > > ### **Q1 Toy experiment**
> > >
> > > **Why ESSA can work when it only adjusts singular values.**
> > >
> > > If the model has no meaningful initialization, then changing only the singular values is basically just a row-wise rescaling, and this won’t solve the problem.
> > >
> > > But this is not the situation ESSA operates in when training LLMs.
> > >
> > > Large language models are:
> > >
> > > 1. Massively pretrained, so their latent space already contains rich structure.
> > > 2. SFT-initialized with LoRA adapters on the downstream task.
> > > 3. ESSA then only needs to rescale the directions learned during SFT, not discover them.
> > >
> > > Geometrically, SFT gives us “good axes” (eigenvectors) and ESSA only adjusts their lengths.
> > >
> > > This is why the method works in practice, even though the parameterization seems restrictive in theory.
> > >
> > > To check this intuition directly, we did exactly the experiment you proposed, but in a form closer to how LLMs are actually trained.
> > >
> > > **MNIST logistic-regression experiments.**
> > >
> > > We used MNIST and set up a pipeline that mirrors the LLM workflow as closely as possible, despite the simplicity of the model.
> > >
> > > - Pretrain stage (analog of LLM pretraining). We trained a logistic regression model on MNIST digits 0-5 (SGD, learning rate 0.1, batch size 128, train size 5000). This gives a “base model” $W'$ (the single matrix of size 784 by 10) that knows something about the data distribution, similar to LLM pretraining.
> > > - “SFT” stage (analog of downstream supervised tuning). We took 400 samples of digits 6-9, initialized a matrix update $\Delta W$ with Kaiming init, and trained only $\Delta W$ using SGD: $W = W' + \Delta W$, where $W$ is the single matrix of size 784 by 10. This is the analogue of LoRA SFT.
> > > - ESSA (solution length 10, population size 24, batch size 64) vs SGD (learning rate 0.1, batch size 64) on singular values
> > >
> > > We decomposed $\Delta W = B S A$ (the single matrix of size 784 by 10) and trained only the singular values $S$, either with ESSA or with SGD.
> > >
> > > Result: ESSA tracks SGD remarkably well on this small problem (anonymous link https://8upload.com/display/dd30460a639d0e49/essa_mnist_sft.jpg.php). On the plot, accuracy is measured on the test set for the “downstream” task (digits 6-9).
> > >
> > > This matches exactly what we see on LLMs: once the low-rank directions are meaningful, scaling them is enough to improve performance substantially.
> > >
> > > **What happens with no the additive matrix and no “SFT”?**
> > >
> > > Here we skip the “SFT” stage entirely. We decompose the pretrained weight (the single matrix of size 784 by 10):
> > >
> > > $W = W' = B' S' A',$
> > >
> > > and train only the singular values $S′$ using ESSA and SGD (SGD: learning rate 0.1, batch size 64, train size 400; ESSA: solution length 10, population size 24, batch size 64, train size 400).
> > >
> > > Result: ESSA still optimizes the downstream task surprisingly well, achieving performance comparable to SGD, despite having no access to a learned low-rank subspace (anonymous link https://8upload.com/display/cab944da46d06bbe/essa_mnist.jpg.php).
> > >
> > > Although the singular-value parameterization is restrictive, the pretrained matrix $W’$ already contains structure that ESSA can exploit by rebalancing its singular directions.
> > >
> > > **Random-initialization setup.**
> > >
> > > For completeness, we also tested the case without pretraining and without “SFT”:
> > >
> > > - we initialized $W$ (the single matrix of size 784 by 10) with Kaiming initialization (same as typical neural nets),
> > > - then performed SVD and optimized only the singular values (SGD: learning rate 0.1, batch size 64, train size 400; ESSA: solution length 10, population size 24, batch size 64, train size 400).
> > >
> > > Here SGD clearly outperforms ESSA, while ESSA improves slowly (anonymous link https://8upload.com/display/7c52d713e607b3ed/essa_base.jpg.php). A network with only Kaiming init has no meaningful geometric structure yet, so scaling its singular values cannot replace proper learning.
> > >
> > > ESSA is capable of optimizing even without any meaningful initialization, although in that case it performs worse than gradient-based methods. When an initialization is present, which is the realistic situation for LLMs after pretraining and SFT, adjusting only the singular values becomes surprisingly effective and ESSA reaches performance comparable to gradient-based training. We demonstrated this both in a controlled toy example on MNIST and, in the main paper, on real tasks and LLMs of different sizes.

---

> > > ### Author Response · Authors · 2025-11-28
> > >
> > > ### **Q2 CMA-ES and Hessian**
> > >
> > > Let’s consider the convex quadratic objective $f_{H}(x)=\frac{1}{2}x^{⊤}Hx$, where $H>0$ - Hessian, centered at the optimum $x^{⋆}=0$. Its level sets are ellipsoids {${x: x^{⊤}Hx=\text{const}}$}.  Let's remember that CMA-ES samples new candidate points as  $x=m+σy$, $y∼\mathcal{N}(0,C)$. The CMA tutorial states explicitly that: “Given a search distribution $\mathcal{N} (m, C)$, there is a close relation between $H$ and  $C$: Setting $C = H^{−1}$  on $f_H$ is equivalent to optimizing the isotropic function $f_{\text{sphere}}(x) = \frac{1}{2}x^{T}x$ (where $H = I$) with $C = I$.” [1]
> > >
> > > Formally, under the change of variables  $z=H^{1/2}x$, the function becomes spherical. If $C=H^{−1}$, then the induced distribution in the z-space is:
> > >
> > > $z=H^{1/2}x=H^{1/2}m+H^{1/2}(x−m)$.
> > >
> > > Since $x−m∼\mathcal{N}(0,C)$,
> > >
> > > $H^{1/2}(x−m)∼\mathcal{N}(0,H^{1/2}CH^{1/2})$.
> > >
> > > If $C=H^{−1}$,
> > >
> > >  $H^{1/2}CH^{1/2}=H^{1/2}H^{−1}H^{1/2}=I$.
> > >
> > > Thus $z∼N(H^{1/2}m,I)$.
> > >
> > > This means that in the new coordinates, the search distribution becomes isotropic (spherical). “The final objective of covariance matrix adaptation is to closely approximate the contour lines of the objective function f. On convex-quadratic functions this amounts to approximating the inverse Hessian matrix, similar to a quasi-Newton method.” [1]
> > >
> > > Moreover CMA-ES works on arbitrary landscapes. For non-quadratic functions there is no single Hessian that describes the whole region. CMA-ES learns the effective curvature of the landscape as experienced through the distribution it currently samples, which is not the analytical Hessian but a statistically meaningful surrogate. Whenever the algorithm samples a population, some points achieve lower objective values, and the directions these points come from are systematically biased toward regions where the landscape locally permits improvement. This bias is enough to extract geometric information.
> > >
> > > The covariance update is built from normalized steps
> > >
> > > $y_{i:\lambda}=\frac{x_{i:\lambda}-m}{\sigma}$,
> > >
> > > where $m\in R^{n}$ - the distribution mean,  $σ>0$ - the global step-size (overall standard deviation), $x_{i:λ}∈R^{n}$ - the i-th best offspring among $λ$ samples, sorted by objective value, n - solution length.
> > >
> > > Normalized steps represent the directions from which successful offspring arrive. These $y_{i:\lambda}$ contain geometric information: if certain directions repeatedly produce successful solutions, their components become larger in the selected set, and the covariance update amplifies variance along those directions.
> > >
> > > The rank-µ update adds information from selected steps in the current generation:
> > >
> > > $C \leftarrow (1 - c_\mu \sum_i w_i)C + c_\mu \sum_{i=1}^{\lambda} w_i, y_{i:\lambda} y_{i:\lambda}^{\top}$,
> > >
> > > where $w_i∈\mathcal{R}$ - the recombination weights ($\sum_{i=1}^{\mu} w_i=1$, $w_1≥w_2≥...≥w_μ>0$ - positive weights, some weights $w_{μ+1},...,w_λ$ may be negative ),  $c_µ ≤ 1$ - learning rate for updating the covariance matrix, $µ ≤ λ$ - number of (positively) selected search points in the population, number of strictly positive recombination weights.
> > >
> > > The outer products $y_{i:\lambda}y_{i:\lambda}^{\top}$ are empirical estimates of second-order structure. Positive weights increase variance along directions where selected steps point; negative weights reduce variance in directions dominated by unsuccessful steps. The scalar $(1 - c_\mu\sum_i w_i)$ fades out old information, giving the update an exponential time window.
> > >
> > > To capture long-range directional trends, CMA-ES uses the evolution path:
> > >
> > > $p_c \leftarrow (1-c_c)p_c+h_σ\sqrt{c_c(2-c_c),\mu_{\mathrm{eff}}} y_w$,
> > >
> > > where  $y_w=\sum_{i=1}^{\mu} w_i y_{i:\lambda}$, $h_σ$ --- Heaviside function, $c_c≤1$ - learning rate, $μ\_{\text{eff}}=\frac{(\sum\_{i=1}^{μ}w\_i)^2}{\sum\_{i=1}^μw\_i^2}$.
> > >
> > > This cumulative vector is a memory of consecutive successful steps. Even if the landscape is non-quadratic or curved, consistent improvement along a direction causes $p_c$ to align with that direction. Because it aggregates information across generations rather than a single batch, $p_c$ provides a low-variance estimate of persistent curvature.
> > >
> > > The rank-1 update injects this long-horizon information into the covariance:
> > >
> > > $C \leftarrow C + c_1p_c p_c^{\top}$,
> > >
> > > where $c_1≈ \frac{2}{n^2}$.
> > >
> > > This increases variance along the dominant direction encoded in $p_c$, improving exploration efficiency in narrow valleys or curved basins. Combining both components yields the full update rule:
> > >
> > > $C \leftarrow (1 - c_1 - c_\mu \sum_i w_i)C$
> > >
> > > $+c_1 p_c p_c^{\top}$
> > >
> > > $+ c_\mu \sum_{i=1}^{\lambda} w_i, y_{i:\lambda} y_{i:\lambda}^{\top}.$
> > >
> > > The resulting covariance is the second-moment structure of successful steps, an “effective inverse curvature” that reflects how the landscape locally permits progress under the current distribution.
> > >
> > > A more detailed explanation can be found in the CMA-ES tutorial [1].

---

> > > ### Author Response · Authors · 2025-11-28
> > >
> > > We hope these additional results clarify the behavior of ESSA. If they satisfactorily resolve the remaining concerns, we would greatly appreciate your consideration in raising the evaluation score.
> > >
> > > [1] Hansen, The CMA Evolution Strategy: A Tutorial

---

### Official Review · Reviewer_DR9j · 2025-10-27

**Soundness:** 2
**Presentation:** 1
**Contribution:** 2
**Rating:** 2
**Confidence:** 3

**Summary:**

This paper proposes ESSA, a gpu-light post-training method that evolves LoRA adapters instead of doing gradient-based RLHF. LoRA adapts large models by adding small low-rank matrices to selected layers and training only those adapters. Evolutionary algorithms (e.g., CMA-ES) are gradient-free searches that sample candidate parameters, score them with a reward, and update a population toward better candidates. ESSA SVD-decomposes the LoRA adapters, freezes the singular vectors, and evolves only the singular values using inference-only evaluations, enabling easy parallelism and INT8/INT4 training. ESSA shows strong scaling with more GPUs and often reaches target accuracy faster than GRPO; quantized training incurs small accuracy loss; tasks include GSM8K/advanced math, IFeval, and a HelpSteer setup.

**Strengths:**

**Originality**: T*he method has no original component, but manage to blend together known ideas in an interesting way.*  The core move is casting alignment as inference-only evolutionary search in a *compressed* SVD-LoRA subspace (optimizing only singular values). This  is a creative recombination of known ideas (LoRA, CMA-ES/ES) that manage to make post-training faster, as it removes the need to computate gradient, allowing the search to be done in INT8/INT4. This is a practical, underexplored angle, that allows consumer grade gpu to align bigger models.

**Empirical Quality:** *TL;DR: the presented result are convincing, and this is good. But they are far to be enough*.
Empirically, the work delivers credible *systems* evidence: strong wall-clock scaling with GPU count, and clear time-to-quality wins over GRPO at high scale. Nevertheless, it would require more experiments to assert this is a definitive result *(see weakness section)*

**Clarity:** part of the work are clear, such as the main ideas of the algorithm.

**Significance:** If the goal is to make post-training alignment scalable and hardware-friendly, this approach is consequential: (i) it materially reduces inter-GPU communication, (ii) it tolerates training-time quantization with small accuracy loss, and (iii) it works across several tasks (math reasoning, instruction following, reward-model tuning) with broadly competitive accuracy.

**Weaknesses:**

1. **Presentation quality.** The manuscript quality is low. It has numerous concrete issues that impede review and reproducibility. In particular, the paragraph “Task and Models” contains major grammar mistakes and punctuation; periods used instead of commas, fragments like “. GSM8K with accuracy as the primary metric.” where sentences do not have verbs. Other problems (but there are many other):
    1. Figure problems: Figure 2 right-panel lower-left square has no numbers; Figure 2 left-panel has inconsistent digit colors; Figure 3 description contains a typo “!x!” artifact.
    2. Table 1 lacks headers/column names/model identifiers, making it impossible to parse.
    3. *Communication of clarity and novelty.* The paper does not crisply articulate *what is new* beyond “apply ES to SVD-LoRA after SFT.” As written, it reads like applying well-known algorithms (LoRA + CMA-ES/ES) to select singular directions post-SFT. The contribution needs to be isolated: specify the novel elements (e.g., singular-values-only parameterization, quantized-training recipe, communication scheme) and contrast against close alternatives.
    4. *Method section organization and writing.* The “Method” subsection reads like an extension of Related Work: it re-introduces LoRA and CMA-ES, breaks narration into many short paragraphs, and uses inconsistent/broken formatting (examples include “1.Samples vs 2.  For each”). This hurts readability and makes the contribution hard to identify.
2. **Theory section is unclear and contextless***.* The manuscript jumps into “latency of gradient-based online alignment” and a latency model without prior grounding. Key terms are not defined where needed (e.g., what an “all-reduce of model-size gradients” specifically entails in this setup? what is a strong base? what is a latency model?). The theoretical *contribution* is unclear and hard to grasp.
3. **Baseline breadth and fairness.** Comparisons lean heavily on GRPO with thin hyperparameter tuning, while ESSA receives broader sweeps and is sometimes run with INT8/INT4 where GRPO remains BF16, an apples-to-oranges wall-clock comparison. Can GRPO be run in INT8? Stronger non-RL baselines (DPO https://arxiv.org/abs/2305.18290 /ORPO https://arxiv.org/abs/2403.07691/ SIMPO https://arxiv.org/abs/2405.14734) are missing; these are common industry references and could materially close gaps.
4. **Evaluation design**. The study emphasizes time-to-accuracy but *does not account for tokens or total FLOPs;* without accurate budget accounting (with different budgets), “scalability” is only half-substantiated.
5. **Missing Ablations:** Critical ablations are absent that isolate the dependence on SFT and show use how the method perform: (i) base model with no finetuning evaluation, (ii) ESSA without the extra SFT (e.g., start from a pretrained model), and (iii) SFT-only without ESSA on identical splits. Furthermore, it is not clear whether the cost of SFT is taken into account in the cost of ESSA as it is part of it
6. **Datasets, scope, and external validity.** Most experiments are on mathematical-reasoning corpora (GSM8K, MATH variants), which are *not* alignment datasets; labeling outcomes as “alignment” overstates generality. One of the main 7B backbone is Qwen2.5-math, already math-specialized, so measured gains are entangled with the backbone’s pre-training; it remains unclear how well the method helps on non-specialized, less-aligned bases. The paper needs breadth: add practitioner-standard families (Gemma, Mistral) and non-math datasets to support claims of generality.

------
In conclusion, I suggest rejecting ESSA because, despite the interesting idea, the paper’s presentation and novelty are unclear (broken writing, malformed figures/tables, theory without concrete claims) and the evaluation is insufficient and biased (narrow baselines, missing ablations/cost accounting, and limited, non-alignment datasets).

**Questions:**

- What, precisely, is the paper’s **novel contribution** beyond applying a standard ES (e.g., CMA-ES) to search singular values after as SFT?
- can you please provide a **single end-to-end algorithm box** (inputs, outputs, steps, hyperparameters) that makes the narration linear and self-contained?
- What is the formal **contribution** of the theory section? Is there a theorem/lemma that yields testable predictions (e.g., iteration time, scaling law, or sample complexity)?
- what is meant by  **“all-reduce of model-size gradients”** explicitly?
- can you provide a **closed-form expression** for ESSA’s per-iteration **communication cost**: bytes/GPU = f(population size, seeds, rewards, ranks, layers).
- What is model latency?
- What does “**strong base**” mean in “Unless stated otherwise, we start from strong base and SFT checkpoints”?  How does it differe from SFT? Do you fine tune additionally models like Qwen-Math for Essa?
- You use mainly **math-reasoning datasets** (GSM8K, MATH500). In what sense are these **alignment** tasks? If the claim is “alignment,” where are safety/harms/behavioral alignment metrics?
- Do you take into account SFT  in ESSA resource count?

---

> ### Author Response · Authors · 2025-11-23
>
> Thank you for the thoughtful and detailed review. We appreciate the time and care you invested in evaluating our submission, as well as the recognition of its strengths, particularly the originality in combining evolutionary search with a compressed SVD-LoRA subspace, the practical implications for scalable and hardware-efficient post-training, and the promising empirical results. Your comments highlight both the potential of the approach and the areas where the paper can be substantially improved, and we are grateful for the constructive guidance.
>
> ### **S1. Clarifying the originality**
>
> We would like to clarify that while our method indeed builds on established components (LoRA, SVD factorization, evolutionary strategies), its originality lies precisely in the novel integration and application of these elements to a previously unexplored problem setting. It is not accurate to characterize the method as “not original” solely because it leverages known ideas: virtually all impactful advances in LLM training, such as RLHF, DPO, or instruction tuning, are themselves recombinations of existing techniques into new, effective pipelines.
>
> To the best of our knowledge, no prior work has applied evolutionary algorithms to the post-training stage of large language models. Our contribution is the first to demonstrate that inference-only evolutionary search can function as a viable alternative to gradient-based alignment. Moreover, we go beyond introducing the idea: we systematically analyze its behavior across diverse scenarios, evaluating both quality and efficiency, and showing when and why this approach works.
>
> In doing so, we open a new training paradigm for LLMs, one that avoids gradients, operates in quantized regimes, reduces communication overhead, and enables post-training on consumer-grade hardware. This direction has been largely absent from prior research, and we believe our work highlights its importance and potential.
>
> ### **W1. Response to comments on presentation quality**
>
> We sincerely appreciate your careful attention to presentation details and your identification of typos, formatting issues, and clarity problems in the initial submission. These comments were extremely helpful. However, we would like to emphasize that such issues, while important to fix, should not be grounds for rejecting the paper, as they do not diminish the scientific contribution or the novelty of the method. All of the identified issues are fully correctable, and we have already conducted a thorough proofreading pass. The current revised version no longer contains these deficiencies.
>
> **(a) Figures.** The white square in Figure 2 (right panel) is intentional: the vertical axis varies the LoRA rank, while the horizontal axis varies the percentage of trainable singular values. Because we round the number of trainable singular values down to an integer, 0.1 of rank 8 yields 0, which is nonsensical for our setting. This is explained explicitly in the caption. In the new revision, we have added an “N/A” label to make this immediately clear.
>
> The inconsistent colors in the left panel of Figure 2 have been fixed. Note that Figures 9–11 in the original submission were already consistent; now all related figures have no problems with color. The typo in Figure 3 has also been corrected.
>
> **(b) Table 1.** We originally avoided adding column labels to prevent clutter, as the meaning was already described clearly in the caption and Section 4.1. The top row indicates the fraction of the original SFT dataset used during the SFT stage, and the bottom row reports the maximum accuracy reached during ESSA training after SFT on that data fraction. For additional clarity, we have now added explicit row labels: **“SFT dataset fraction”** and **“Max ESSA accuracy.”**
>
> **(c) Clarity of novelty.** You suggest that the novelty section should highlight the singular-values-only parameterization, quantized training, and the communication scheme. In fact, two of these points are already stated prominently in the introduction (first page, in bold). The third point you mention (the communication scheme) is *not* claimed as a novelty of our work; this aspect is largely adapted from [1], as we describe. Our novelty claim instead includes demonstrating that ESSA reliably recovers gradient-base methods level quality across diverse tasks and conditions.
>
> **(d) Method section structure.** In these subsections we do not re-explain LoRA or CMA-ES; we introduce only the mathematical notation needed to clearly define ESSA. Since ESSA is built on top of LoRA and CMA-ES, some overlap in notation is unavoidable, but this helps us precisely specify how the method operates and which parameters it manipulates.
>
> Beyond the issues you mentioned, we have corrected a number of other minor errors throughout the text. We invite you to review the revised paper, and we would be grateful for any additional feedback you may have on the clarity or presentation of the material.

---

> ### Author Response · Authors · 2025-11-23
>
> ### **W2/Q4/Q6/Q7. Theoretical terminology**
>
> In writing the manuscript, we relied on terminology that is standard in the fields of Distributed Deep Learning and High-Performance Computing applied to LLMs (e.g., widely used in literature discussing ZeRO [2], Megatron-LM [3], PyTorch DDP [4], Large Scale Training [5] without introductory definitions). Below, we provide the requested clarifications for the terminology used in our theoretical analysis and our initialization procedure.
>
> **Q4 Definition of "All-Reduce of Model-Size Gradients".** By “all-reduce of model-size gradients,” we refer to the standard pattern used in data-parallel training:
>
> - We have $N$ identical replicas of the model on $N$ GPUs (a data-parallel setup).
> - Each replica processes a different mini-batch and computes local gradients with respect to all model parameters.
> - To keep the replicas synchronized, we then average these gradients across all GPUs and apply the same parameter update to every replica
>
> This averaging is implemented as an all-reduce collective (e.g., as described in the [NCCL Collectives User Guide](https://www.google.com/url?sa=E&q=https%3A%2F%2Fdocs.nvidia.com%2Fdeeplearning%2Fnccl%2Fuser-guide%2Fdocs%2Fusage%2Fcollectives.html)), applied to a tensor whose size matches the entire parameter vector of the model, hence “model-size gradients.”
>
> In practice, this all-reduce is often realized as a Ring All-Reduce, which can be decomposed into a reduce-scatter followed by an all-gather. Modern large-scale training systems typically use such implementations [5, 6]. This is precisely what we mean by an “all-reduce of model-size gradients.”
>
> **W2/Q6 Definition of Latency Model and Theoretical Contribution.** In the context of systems optimization, "latency" refers to the total wall-clock time required to complete one full optimization step (iteration).
>
> By a "simple latency model," we refer to an idealized theoretical framework that calculates the fundamental computational and communication costs based on hardware peak bandwidth and FLOPs. This model intentionally abstracts away real-world variability, such as network jitter, system noise, or stragglers, to provide a clear, first-principles comparison between the two methods.
>
> The formal contribution of the theoretical section is the derivation of a closed-form inequality (Theorem 3.1) based on this model. It proves that ESSA’s communication cost scales as $O(1)$ (transmitting only seeds and scalar rewards), whereas gradient-based methods scale as $O(M)$ (where $M$ is the model size). This allows us to predict the "break-even" population size below which ESSA is mathematically guaranteed to be faster than gradient methods.
>
> We have clarified this distinction in the revised manuscript: *"We start from strong SFT checkpoints appropriate for each dataset. Experiments without SFT are provided in Appendix E.1."*
>
> **Q7 Clarification of "Strong Base" and SFT.** By "strong base," we explicitly mean that we initialize our alignment phase from a model that has already undergone Supervised Fine-Tuning (SFT) on the target domain, rather than starting from a raw pre-trained base model.
>
> Regarding your question on additional fine-tuning: Yes, for models like Qwen2.5-Math-7B, we perform an additional, short SFT stage on the specific SFT dataset to initialize the LoRA adapter. This ensures that the parameters start in the optimal task-specific region before we decompose them via SVD and begin the ESSA evolutionary search. This SFT step is identical for both the baseline (GRPO) and our method.

---

> ### Author Response · Authors · 2025-11-23
>
> ### **W3. Baselines, hyperparameters, and fairness of comparison**
>
> Below we clarify the hyperparameter choices, quantization considerations, and the rationale behind the selected baselines.
>
> **Hyperparameter sweeps for GRPO.** For each LoRA rank $({2,4,8,16,32})$ we performed targeted GRPO sweeps over the commonly used  learning rates ${(1×10^{-5},5×10^{-4},6×10^{-6}})$. The sweep space is narrower than for ESSA because each GRPO configuration is very expensive: every run requires full online rollouts, policy updates over long trajectories, and gradient synchronization. A broader grid would make GRPO prohibitively costly at the scales we target. By contrast, ESSA’s inference-only pipeline makes broader exploration essentially free. Importantly, we kept the computation budget equal in terms of GPU-hours across methods, ensuring a fair head-to-head comparison under the same total compute budget.
>
> **Why GRPO remains in BF16.** Regarding INT8/INT4: gradient-based training cannot be fully quantized without severe instability. Existing low-precision training approaches rely on mixed precision, where several tensors must stay in BF16 or FP32 for stable updates including master weights, accumulated gradients, optimizer states, model weights during update steps [5]. Fully quantizing all components simultaneously causes the training process to become unstable and fail.
>
> Even with mixed precision, the memory footprint remains dominated by all these components. ESSA, in contrast, stores only a single quantized model instance. This allows ESSA to keep a full 70B model on one GPU, while GRPO must shard such models regardless of quantization. Thus, ESSA retains a structural hardware advantage that gradient-based methods cannot match.
>
> Moreover, aggressively quantizing gradient-based RL methods often reduces final quality. We therefore train GRPO in BF16 to provide a strong, not artificially weakened, baseline.
>
> **Why DPO, ORPO, and SIMPO are not the correct comparison class.** Our work specifically addresses the bottlenecks of online, on-policy RL alignment, where sampling and evaluation happen continuously. Methods like DPO, ORPO, and SIMPO are offline direct-alignment techniques: they operate on a static preference dataset and use pure supervised backpropagation. These methods have fundamentally different computational regimes, scalability limits, and use cases.
>
> Prior work also shows that when online sampling is available, RL-based alignment consistently outperforms offline direct-alignment [7, 8], especially on tasks with verifiable rewards such as math reasoning and instruction following. Consequently, the standard practice in such domains is to use online RL (e.g., GRPO, RLVR), not offline approaches [9, 10].
>
> Thus, these offline algorithms target a complementary setting. Since our contribution aims to improve scalability within the RL-based, online alignment family, GRPO is the most informative and relevant baseline for evaluating ESSA.
>
> ### **W4. On FLOPs, token budgets, and the choice of a fair comparison metric**
>
> In our setting making a fair comparison in terms of FLOPs or token budgets is quite challenging because the methods have very different computational profiles. ESSA relies almost entirely on autoregressive sampling and generates many candidate trajectories, whereas GRPO-style baselines use far fewer generated tokens per update but require expensive backward passes.
>
> As a result, a *token* budget would tend to favor GRPO (since it uses fewer generated tokens per step), while a *FLOPs* budget would tend to favor ESSA (since modern GPUs are highly optimized for low-precision inference, e.g., FP8/FP4, making pure decoding extremely efficient). This asymmetry makes it difficult to define a single “fair” budget that is not implicitly biased toward one family of methods.
>
> Similarly, metrics such as MFU (Model FLOPs Utilization) are not ideal for our comparison. MFU depends on time per step and also affected by communication overheads.
>
> Given these constraints, we chose wall-clock *time-to-accuracy* on a fixed hardware configuration as the most practical and comparable primary metric. It directly reflects the end-to-end cost of reaching a given performance level under realistic conditions, without implicitly tilting the comparison toward either gradient-based methods or ESSA.

---

> > ### Author Response · Authors · 2025-11-23
> >
> > ### **Q3. Contribution of the theory section**
> >
> > In the theory section, our main formal contributions are:
> >
> > **Tight iteration-time characterization for gradient-based pipelines.** We derive the minimum possible per-iteration latency for a broad class of gradient-based methods under our hardware/latency model, and we prove what the optimal device split is between training and inference stages in a pipelined setup.
> >
> > **Provable advantage of ESSA over any idealized gradient pipeline.** Using this result, we show that there exist ESSA hyperparameters (population size and batch size) such that any idealized gradient-based pipeline, even optimally configured under our model, has strictly larger per-step wall-clock time than the ESSA setup. This yields a clear, testable prediction about iteration time that we verify empirically.
> >
> > **Communication complexity independent of model size for ESSA.** We prove that in ESSA, the communication time between GPUs is independent of the number of model parameters, in contrast to gradient-based methods where communication scales with parameter size. This leads to the prediction (again testable) that ESSA remains communication-efficient as model size grows.
> >
> > **Hardware-scaling implications (bandwidth and federated/on-device).** Within our latency model, ESSA’s scaling behavior does not depend on parameter size and depends on inter-GPU / intra-node peak memory bandwidth by a factor of 2× less than gradient-based methods. This formally supports the claim that ESSA is easier to scale to settings such as federated learning and on-device training, which we highlight in the future work section.
> >
> > Our focus is therefore on iteration-time and scaling-law predictions for wall-clock performance and communication, rather than sample complexity.
> >
> > ### **Q5. Closed-form expression for ESSA’s per-iteration communication cost**
> >
> > Under the communication model in Appendix B, ESSA’s per-iteration communication only involves sharing a scalar reward and a random seed across GPUs. Let (g) be the number of GPUs and BW_peak the peak inter-GPU bandwidth. Then the per-iteration communication time per GPU in ESSA is
> >
> > $$
> > T_{\text{comm}}^{\text{ESSA}} = \frac{(g - 1)}{g} \cdot \frac{2 \cdot 4\ \text{bytes}}{\text{BW}_\text{peak}},
> > $$
> >
> > where the factor (2 * 4) comes from transmitting one 4-byte scalar reward and one 4-byte random seed. We use the factor ((g - 1)/g) because, in the standard ring implementation of an all-gather (in practice, we implement this as a single all-gather NCCL collective), each GPU sends its local payload (reward + seed) to all other GPUs and receives the corresponding payloads from them. This yields the usual ((g - 1)/g) scaling of per-GPU volume (see, e.g., the NCCL all-gather collective implementation  and related discussions in  [5, 6]).
> >
> > Equivalently, the per-iteration communication volume per GPU is
> >
> > $$
> > \text{bytes/GPU} = \frac{(g - 1)}{g} \cdot 2 \cdot 4\ \text{bytes}.
> > $$
> >
> > Crucially, in our setup this communication cost is independent of population size, number of layers, or model parameter count and depends only on the small, fixed-size metadata (reward and seed).
> >
> > [1] Salimans et al, Evolution Strategies as a Scalable Alternative to Reinforcement Learning
> >
> > [2] Rajbhandari et al, ZeRO: Memory Optimizations Toward Training Trillion Parameter Models
> >
> > [3] Shoeybi et al, Megatron-LM: Training Multi-Billion Parameter Language Models Using Model Parallelism
> >
> > [4] Li et al, PyTorch Distributed: Experiences on Accelerating Data Parallel Training
> >
> > [5] Tazi et al., The Ultra-Scale Playbook: Training LLMs on GPU Clusters
> >
> > [6] Austin et al., How to Scale Your Model
> >
> > [7] Tang et al, Understanding the performance gap between online and offline alignment algorithms
> >
> > [8] Ivison et al, Unpacking DPO and PPO: Disentangling Best Practices for Learning from Preference Feedback
> >
> > [9] Shao et al, DeepSeekMath: Pushing the Limits of Mathematical Reasoning in Open Language Models
> >
> > [10] Lambert et al, Tulu 3: Pushing Frontiers in Open Language Model Post-Training
> >
> > [11] Lightman et al, Let’s Verify Step by Step
> >
> > [12] Zeng et al, SimpleRL-Zoo: Investigating and Taming Zero Reinforcement Learning for Open Base Models in the Wild
> >
> > [13] Luo et al, WizardMath: Empowering Mathematical Reasoning for Large Language Models via Reinforced Evol-Instruct
> >
> > [14] Zhang et al, The Lessons of Developing Process Reward Models in Mathematical Reasoning
> >
> > [15] Cheng et al, Stop Summation: Min-Form Credit Assignment Is All Process Reward Model Needs for Reasoning
> >
> > [16] Zhou et al, Instruction-Following Evaluation for Large Language Models
> >
> > [17] Wang et al, HelpSteer2: Open-source dataset for training top-performing reward models
> >
> > [18] Wang et al, Interpretable Preferences via Multi-Objective Reward Modeling and Mixture-of-Experts
> >
> > [19] Bai et al, Training a Helpful and Harmless Assistant with Reinforcement Learning from Human Feedback
> >
> > [20] He et al, Defeating Nondeterminism in LLM Inference

---

> ### Author Response · Authors · 2025-11-23
>
> ### **W5/Q9. Starting ESSA from a base model and accounting for SFT cost**
>
> **Starting ESSA from a base model is theoretically inappropriate.** ESSA operates by optimizing only the singular values of LoRA matrices $A$ and $B$ in a compressed SVD-LoRA parameterization.
>
> If we begin ESSA from a base model (without SFT), then: LoRA matrix $B$ becomes all zeros, LoRA matrix $A$ is randomly initialized. Therefore the corresponding singular values represent a spectrum of a zero-random pair of matrices, which is both: theoretically ill-posed (the spectral structure does not encode meaningful directions), and empirically harmful, as the optimizer attempts to adjust eigenvalues of matrices that have no alignment-related signal.
>
> Thus, starting ESSA from a base model is not only conceptually inconsistent with the method, but also produces degraded quality.
>
> **Nevertheless, we performed the requested experiment.** To fully answer the reviewer’s question, we ran ESSA starting directly from the base **Llama-3.1-8B** model on **IFEval**, using:
>
> - LoRA rank = 8
> - Population size = 96
> - ESSA training time = 6 hours
>
> Below are the results you requested, including baseline and SFT comparisons.
>
> | Method | Score (mean ± std) |
> | --- | --- |
> | **Base only** | 0.062 ± 0.001 |
> | **Base + ESSA** | 0.237 ± 0.011 |
> | **SFT only** | 0.455 ± 0.001 |
> | **SFT + ESSA** | 0.589 ± 0.008 |
>
> These results confirm:
>
> 1. ESSA can improve a raw base model, but performance remains substantially below SFT because the initial LoRA spectral structure is meaningless.
> 2. SFT provides a well-structured low-rank subspace; ESSA is effective precisely because it builds on top of this structure.
>
> **SFT training cost.** All wall-clock curves in Figures 3–8 report only the online alignment phase. Each run begins from the same SFT-trained LoRA checkpoint within a given backbone-task pair, and the time origin $t=0$ corresponds to the first evaluation step after loading this shared checkpoint. Although both methods start from identical SFT parameters, their initial positions on the quality-time curves may differ slightly due to sampling nondeterminism and inference-level variability [20].
>
> We do not include SFT warm-start time from these curves because SFT is a shared. We use the same SFT checkpoint for both ESSA and GRPO for each backbone-task pair, the time spent on the SFT introduces the same constant for both methods and does not affect any relative time-to-quality comparisons. In the revised version of the work SFT hyperparameters and GPU-hours will be reported separately in Appendix F:
>
> | Model and Dataset | SFT total train size/global batch size/lr/epochs | SFT GPU hours |
> | --- | --- | --- |
> | Qwen2.5-7B (GSM8K) | $1495/16/1×10^{-4}/1$ | $0.18$ |
> | Qwen2.5-Math-7B (GSM8K) | $1495/16/1×10^{-4}/1$ | $0.16$ |
> | Qwen2.5-7B  (PRM800K) | $3600/16/5×10^{-4}/1$ | $0.43$ |
> | Qwen2.5-Math-7B  (PRM800K) | $3600/16/5×10^{-4}/1$ | $0.40$ |
> | Qwen2.5-32B  (PRM800K) | $3600/128/1×10^{-6}/3$ | $9.80$ |
> | Qwen2.5-72B  (PRM800K) | $3600/128/1×10^{-6}/3$ | $51.97$ |
> | Llama3.1-8B (IF-Eval-like) | $1700/16/5×10^{-4}/1$ | $0.50$ |
> | Llama3.1-8B (HelpSteer2) | $19000/128/1×10^{-6}/3$ | $3.74$ |

---

> ### Author Response · Authors · 2025-11-23
>
> ### **W6/Q8. Datasets, scope, and external validity**
>
> **Why math-reasoning tasks count as alignment tasks.** We respectfully disagree with the claim that mathematical-reasoning datasets are “not alignment datasets” or that using them overstates generality. Our position is grounded in standard definitions of alignment used in the field.
>
> As Jan Leike writes in “What is the alignment problem?” (https://aligned.substack.com/p/what-is-alignment):
>
> > “On a very high level, building performant AI systems requires two ingredients:
> Capability - the system could do the intended task.
> Alignment - the system does the intended task as well as it could.
> Thus if the system doesn’t do the intended task, this is due to a capability problem, an alignment problem, or both.”
> >
>
> In our case, the intended task is solving a mathematical reasoning problem. We want the model to perform the task as well as possible, reliably and truthfully. Therefore, improving correctness, stability, and consistency under verifiable reward signals is alignment in the precise sense Leike describes.
>
> This framing is also fully consistent with OpenAI work. In “Let’s verify step by step” [11], the authors study alignment under verifiable rewards exclusively on math reasoning, and release the PRM800k dataset to support alignment research:
>
> > “We release PRM800K … with the hope that removing this barrier will catalyze research on the alignment of large language models.”
> >
>
> Our experiments explicitly include PRM800k in this same scientific tradition.
>
> **On the Use of Qwen2.5-Math.** Using Qwen-Math on mathematical datasets is standard practice in the literature [12, 13, 14, 15]. We rely on the base (non-Instruct) Qwen-Math model, which has only undergone pretraining on math-focused corpora and has not received any post-training or alignment-style instruction tuning. Our method explicitly performs this missing post-training stage and fairly compares final performance after alignment. Choosing a pretrained backbone that is better adapted to the mathematical domain is appropriate for evaluating math-reasoning post-training, just as prior work does. Nonetheless, we also demonstrate that our algorithm works with other backbone families and on non-math tasks, showing that the approach is not limited to mathematics.
>
> **Breadth Beyond Mathematics.** We want to emphasize that our experiments are not limited to mathematical tasks nor to Qwen-based models. The paper includes two non-math, non-Qwen evaluations, and we **additionally** provide a **third external** experiment to further demonstrate generality:
>
> 1. **Instruction Following (IFEval) [16]** using **Llama-3.1-8B** (Section 4.4.3)
>     - Verifiable, non-math reward signal.
>     - ESSA consistently outperforms GRPO.
> 2. **General Assistant Behavior** using **HelpSteer2 [17]** with **Llama-3.1-8B** (Section 4.4.4)
>     - Captures truthfulness, politeness, and general helpfulness preferences.
>     - Reward model: ArmoRM-Llama3-8B-v0.1 [18].
>     - ESSA achieves competitive or superior performance relative to GRPO.
> 3. **Safety and preference alignment** with **Mistral-7B-v0.3** on **Anthropic HH-RLHF [19]** (additional non-math experiment provided for the reviewer)
>     - Reward model: OpenAssistant/deberta-v3-large-v2.
>     - The learning curves (https://8upload.com/display/9cffcb9b22f13078/essa_grpo_hh_mistral.jpg.php anonymous link) indicate that ESSA achieves a more favorable learning trajectory than GRPO: it reaches higher reward levels and maintains more stable training dynamics over time.
>     - This experiment is provided to further demonstrate generality across architectures and alignment-style datasets.
>
> Together, these results show that ESSA generalizes beyond math: it performs effectively across multiple architectures (Llama, Mistral) and multiple non-math alignment tasks, including instruction following, helpfulness, and safety-oriented preference rewards.

---

> ### Author Response · Authors · 2025-11-23
>
> **Behavioral/Safety Alignment Evaluation.** You requested an evaluation along safety, harms, and behavioral alignment dimensions, rather than relying only on mathematical metrics.
>
> To address this, we performed a side-by-side (SBS) evaluation comparing the ESSA and GRPO checkpoints trained on Anthropic HH-RLHF (the third experiment described above). We used GPT-4o as an impartial judge with the following system prompt:
>
> > Act as an impartial judge and evaluate the quality of the responses provided by two AI assistants. Determine which response better satisfies the user’s request.
> >
> >
> > Evaluate them based on alignment with the user's request, helpfulness, relevance, accuracy, completeness, conciseness, and truthfulness.
> >
> > …
> >
> > If both answers are irrelevant, hallucinated, or inconsistent with the query, choose “C”.
> >
>
> We evaluated both models on held-out safety-, behavior-, and helpfulness-related prompts:
>
> | ESSA | Tie | GRPO |
> | --- | --- | --- |
> | 12% | 76% | 12% |
>
> This is a strong result because our goal is not to beat GRPO, but to establish that a lightweight evolutionary search in an SVD-LoRA subspace can reach GRPO-level alignment quality while:
>
> - reducing engineering complexity,
> - improving parallel efficiency,
> - enabling INT8/INT4 training,
> - and running on commodity hardware.
>
> In summary, matching GRPO’s alignment quality evidenced by symmetric win rates and a large tie region is precisely the success condition for ESSA, showing that gradient-free optimization can meaningfully substitute for RLHF-style gradient descent in practice.
>
> ### **Q1. Novel contribution**
>
> Our work is the first to successfully apply evolutionary algorithms to the post-training of large language models. No previous method has demonstrated that evolutionary optimization can serve as a practical alternative to gradient-based alignment for modern 7B–70B architectures. Our contribution is not simply “running CMA-ES on LoRA,” but making evolutionary training actually work at LLM scale.
>
> The key enabling idea is a novel SVD-LoRA singular-values-only parameterization. By decomposing the SFT-trained LoRA matrices and optimizing only their singular values while freezing the singular vectors, we obtain a compact, well-conditioned search space aligned with meaningful SFT directions. This structured compression dramatically reduces the effective dimensionality of the optimization problem and is essential for stable and efficient evolutionary search.
>
> Built on this representation, we design an inference-only evolutionary post-training pipeline that requires no gradients, no optimizer states, and no backward pass. As a result, ESSA supports full INT8/INT4 training, eliminates gradient synchronization, and enables on-device post-training, capabilities gradient-based RLHF methods cannot offer. Combined with communication-minimal scheme (workers exchange only a random seed and a scalar reward), this creates a new, hardware-efficient training regime.
>
> Finally, we show that this evolutionary approach is general, achieving strong results across mathematical reasoning, instruction following, and general assistant tasks, using diverse backbones (Qwen, Llama, Mistral). Thus, our novelty lies in introducing a new, practically viable paradigm for LLM post-training enabled by a unique parameterization, a low-precision inference-only pipeline, and a communication-efficient evolutionary design.

---

> ### Author Response · Authors · 2025-11-23
>
> ### **Q2. Algorithm**
>
> A full pseudocode specification of ESSA is included in **Appendix A.** We provide an updated version of Figure 1 together with a specification of all terms (see anonymous link https://8upload.com/display/5a21b2d4714d0c4a/essa_device_page-0001.jpg.php), and we have rewritten the Method section to present each step of the algorithm clearly and coherently. The previous **“rank N”** is replaced with **“device N”**, indicating the **GPU worker rank** in the distributed evaluation setup.
>
> ESSA starts from a LoRA module trained with SFT, consisting of the low-rank projection matrices $A$ and $B$. The SFT stage provides a task-aligned initialization for these adapters.
>
> After SFT stage, each LoRA matrix is decomposed using the SVD:
>
> $A = U\_A \operatorname{diag}(\sigma_A) V\_A^\top,\quad$$B = U\_B \operatorname{diag}(\sigma\_B) V\_B^\top,$
>
> $\Delta W = BA = U\_B \operatorname{diag}(\sigma\_B) V\_B^\top U\_A \operatorname{diag}(\sigma\_A) V\_A^\top,$
>
> with $U_A,V_A,U_B,V_B$ fixed and only the singular values $\sigma_A,\sigma_B$ trainable.
>
> This SVD-LoRA representation preserves the expressive capacity of the original LoRA update while sharply reducing the number of variables that the evolutionary search must explore. We then apply CMA-ES to optimize the remaining singular values, maintaining and updating a multivariate normal search distribution over this compact parameterization. Each ESSA iteration proceeds as follows:
>
> - CMA-ES samples $λ≥2$ candidate vectors of singular values from its current multivariate normal search distribution. The distribution is parameterized by a mean, a global step-size, and a covariance matrix that encodes anisotropic search directions. In the updated version of Figure 1 the previous term **“layer n”** is replaced with **“slice n”**. The full CMA-ES candidate vector  $σ_i∈\mathcal{R}^{solution\_length}$ ($i = 0, 1, ..., N,$  where $N$ is the number of evaluation devices) is partitioned into contiguous slices, each containing the  $2 \cdot \mathrm{LoRA\_rank}$ singular values for the LoRA factors  $A$ and $B$. Each slice corresponds to a single LoRA attention matrix (e.g., $W\_Q$, $W\_K$, $W\_V$, $W\_O$) across all layers. The term **“solution length”** denotes the dimensionality of the CMA-ES search vector - the concatenation of all SVD-LoRA singular values across all matrices and all layers: $\sigma\_i=[ \sigma(0), \sigma(1),…, \sigma(k-1) ] \in \mathcal{R}^\mathrm{solution\\\_length},$ where $k$  is a number of training matrices ($\mathrm{num\\\_matrices}$) and each $\sigma(j)$ is a slice.
>
>     With number of layers ($\mathrm{num\\\_layers}$), number of training matrices per layer ($\mathrm{num\\\_matrices\\\_per\\\_layer}$),  $\mathrm{LoRA\\\_rank}$:
>
>      $\mathrm{solution\\\_length} = \mathrm{num\\\_layers} \cdot \mathrm{num\\\_matrices\\\_per\\\_layer} \cdot \mathrm{LoRA\\\_rank} \cdot 2,$
>
>     Since $\mathrm{num\\\_matrices} = \mathrm{num\\\_layers} \cdot \mathrm{num\\\_matrices\\\_per\\\_layer},$
>
>     $\mathrm{solution\\\_length} = \mathrm{num\\\_matrices} \cdot \mathrm{LoRA\\\_rank} \cdot 2$
>
> - For each candidate, we reconstruct the low-rank factors $A$ and $B$ by injecting the candidate’s singular-value offsets into the fixed SVD bases. For $σ_A$, $σ_B$ at the $j$-th LoRA matrix:
>
>     $\tilde{\sigma\_A} =  \sigma\_A + \sigma(j)\_{[:\mathrm{LoRA\\\_rank}]},$       $\tilde{\sigma\_B} =  \sigma\_B + \sigma(j)\_{[\mathrm{LoRA\\\_rank}:]},$ $j = 0, 1, …, k-1,$
>
>      where $k$ $=$ $\mathrm{num\\\_matrices}$.
>
>      The resulting update $ΔW=BA$ is applied, and the model is evaluated on the alignment objective to obtain a scalar reward.
>
> - After all candidates have been evaluated in parallel, CMA-ES updates its mean, step-size, and covariance. This adaptation steers the search distribution toward productive directions in the objective landscape. Only random seeds and scalar rewards are communicated among workers, which enables near-linear scaling across many GPUs.

---

> ### Author Response · Authors · 2025-11-23
>
> We respectfully ask you to consider raising the score in light of the substantial revisions and clarifications provided. We addressed each of your concerns thoroughly improving the presentation, strengthening the methodological explanation, adding new experiments (including non-math and safety-alignment settings), clarifying novelty, and providing the requested theoretical and communication-cost formulations. These changes significantly enhance both the clarity and the scientific contribution of the work. We hope the revised version demonstrates that ESSA introduces a genuinely new and practically impactful paradigm for LLM post-training, and that the paper now meets the standard for acceptance.

---

### Official Review · Reviewer_7J5i · 2025-10-28

**Soundness:** 3
**Presentation:** 2
**Contribution:** 2
**Rating:** 2
**Confidence:** 4

**Summary:**

ESSA is a gradient-free alignment framework that optimizes only the singular values of LoRA adapters after an SVD parameterization; the SVD bases are fixed from an SFT warm-start, and CMA-ES evolves the singular-value vector using *inference-only* evaluations (INT8/INT4 supported). The paper reports comparable or better quality than GRPO across math (GSM8K, PRM800K, advanced-math suites) and instruction following (IFEval), with markedly better scaling in GPU count and wall-clock time (e.g., Qwen2.5-32B on PRM800K) while communicating only seeds and scalar rewards.

**Strengths:**

- Gradient-free *ES loop* on a compact search space (singular values only); inference-only and quantization-friendly (INT8/INT4) for large models.
- Scaling results and latency model argue for superior parallel efficiency vs. GRPO; communication volume is seeds+rewards only.
- Empirics: consistent time-to-quality gains over GRPO and competitive final accuracy across several backbones and tasks; ablations on population size, LoRA rank, and fraction of trainable singular values (α).

**Weaknesses:**

1. Initialization clarity: The method relies on an **SFT stage** to initialize LoRA factors $A, B$, but it is not fully explicit whether a **single** SFT LoRA per (task, backbone) is used or **multiple** LoRA initializations are required across experiments. Greater specificity would aid reproducibility.
2. **Scope of “gradient-free”**: The optimization loop is gradient-free, but the pipeline **depends on SFT**; the paper acknowledges warm-start sensitivity (accuracy improves with more SFT data), so the claim should be framed as “gradient-free **post-SFT** optimization.”
3. **Timing accounting**: In wall-clock comparisons (e.g., Figures 3-8), it remains ambiguous whether the reported times **exclude** the SFT warm-start and with what cost.
4. **Baselines**: The primary online baseline is **GRPO**; a rationale is given, but readers may expect positioning relative to **DPO/ORPO/SIMPO** beyond the Related Work discussion.

**Questions:**

1. **SFT initialization**: Please detail the **number and scope** of SFT initializations used. Is there **one LoRA** (per backbone, per task) that is SVD-parametrized, or are **multiple** SFT LoRA adapters prepared? Include dataset size, epochs, and optimizer settings for SFT to enable reproduction.

2. **Timing inclusions**: Do the wall-clock curves in **Figures 3-8** **include or exclude** the SFT warm-start? If excluded, please report the SFT cost (minutes/GPUs) and **time-to-quality including SFT** to fully reflect end-to-end alignment.

---

> ### Author Response · Authors · 2025-11-17
>
> Thank you for your careful review, and for highlighting the strengths of our approach. We appreciate your recognition of the ESSA loop that enables an inference-only and quantization-friendly workflow, the favorable scalability and latency properties arising from minimal communication, and the empirical consistency of our time-to-quality improvements over GRPO together with the supporting ablations. Below we address each of your concerns in detail.
>
> **W1/Q1** **Single SFT initialization for each backbone-task pair**
>
> For every backbone-task pair in the paper, we train exactly one SFT LoRA adapter. This SFT checkpoint is then used as the sole initialization for all subsequent experiments - both ESSA and GRPO, and across all hyperparameter configurations. We do not prepare multiple LoRA initializations for any experiment. To avoid confusion, we will explicitly state this in Section 3.2 of the revised paper. We will also expand Appendix F to include the complete SFT configuration:
>
> - **SFT parameters:** LoRA adapters only (same LoRA rank as later used in ESSA/GRPO), applied to Q/K/V/O attention projections. No SVD parametrization is used during SFT.
> - **Optimization:** AdamW, $β₁=0.9$, $β₂=0.95$, $ε=1×10⁻¹²$, *max_grad_norm* $=2.0$.
> - **Batching:**
>     - Global batch size 16 for Qwen2.5-7B, Qwen2.5-Math-7B, LLaMA-3.1-8B
>     - Global batch size 128 for Qwen2.5-32B and Qwen2.5-72B
> - **Precision:** BF16 (no quantization during SFT).
> - **Datasets:**
>     - GSM8K: 1495 examples
>     - PRM800K: 3600 examples
>     - IFEval: 1700 examples
>     - HelpSteer2: 19000 examples
>
> This unified SFT stage ensures a fair and controlled comparison between ESSA and GRPO and should fully resolve concerns regarding initialization scope.
>
> **W2 The role of SFT and the meaning of ”gradient-free”**
>
> ESSA is intended as a gradient-free online alignment stage, and it assumes a standard SFT warm-start. In the revised Section 1 and Section 3.1 we clarify that ESSA replaces only the post-SFT alignment loop rather than the full training pipeline.
>
> Although SFT itself uses gradients, this does not undermine the “gradient-free alignment” characterization. In modern LLM training pipelines, SFT is both computationally and communicationally lightweight compared to online RL-style alignment. This distinction is emphasized in prior work: Allal et al. note that *“SFT requires modest compute compared to RL… and in a fraction of the time required for RL”* [1], and Ouyang et al. report that training their 175B SFT model required 4.9 petaflops/s-days, while the subsequent PPO-ptx stage required 60 petaflops/s-days over an order of magnitude more [2]. This matches our setting: the dominant source of cost and communication is the online RL alignment phase, and ESSA is designed specifically to eliminate this costly component while preserving the inexpensive warm-start.
>
> Beyond compute considerations, relying on an SFT initialization is also algorithmically preferable for evolutionary optimization. Without SFT, the LoRA matrices start as a uninformative random Gaussian $A$ and a zero matrix $B$ (whose singular vectors reduce to identity). Evolutionary optimization works best when it begins from a well-structured starting point, as this focuses the search on task-relevant directions and improves both stability and sample efficiency. The SFT LoRA provides such structure, allowing ESSA’s compact singular-value optimization to perform far more reliably.
>
> To further clarify this distinction, we will include additional experiments in Appendix E where ESSA and GRPO are initialized directly from the base model (i.e., without SFT) for GSM8K with Qwen2.5-7B and Qwen2.5-Math-7B. As expected, runs starting from a random LoRA achieve lower absolute accuracy; nevertheless, the relative performance between ESSA and GRPO remains consistent, as shown in the figures: Qwen2.5-7B — https://8upload.com/display/c1b047a50aec8b8d/16_grpo_essa_2_page-0001.jpg.php (anonymous link), Qwen2.5-Math-7B — https://8upload.com/display/cdc1600f9e8d63e1/16_grpo_essa_page-0001.jpg.php (anonymous link). These results confirm that ESSA can operate without SFT, while being primarily intended as a gradient-free replacement for the online alignment phase.

---

> ### Author Response · Authors · 2025-11-17
>
> **W3/Q2 Timing accounting**
>
> All wall-clock curves in Figures 3-8 report only the online alignment phase, and in every case the runs begin from the same SFT-trained LoRA checkpoint for both ESSA and GRPO. The time origin $t=0$ corresponds to the first evaluation step after loading this shared SFT checkpoint. Although both methods start from identical SFT parameters, their starting points on the quality-time curves may differ slightly in practice due to sampling nondeterminism and inference-level variability [3].
>
> We omit the SFT warm-start time from the alignment curves because SFT is a shared, one-time initialization step common to both ESSA and GRPO. Including it would add the same constant offset to all curves and would not change any of the relative time-to-quality comparisons that the figures are meant to highlight. We will make this explicit in the experimental setup section.
>
> To support full reproducibility, we will additionally provide in Appendix F a table summarizing, for each backbone-task pair, the SFT hyperparameters and GPU-hours used for SFT training:
>
> | Model and Dataset | SFT total train size/global batch size/lr/epochs | SFT GPU hours |
> | --- | --- | --- |
> | Qwen2.5-7B (GSM8K) | $1495/16/1×10^{-4}/1$ | $0.18$ |
> | Qwen2.5-Math-7B (GSM8K) | $1495/16/1×10^{-4}/1$ | $0.16$ |
> | Qwen2.5-7B  (PRM800K) | $3600/16/5×10^{-4}/1$ | $0.43$ |
> | Qwen2.5-Math-7B  (PRM800K) | $3600/16/5×10^{-4}/1$ | $0.40$ |
> | Qwen2.5-32B  (PRM800K) | $3600/128/1×10^{-6}/3$ | $9.80$ |
> | Qwen2.5-72B  (PRM800K) | $3600/128/1×10^{-6}/3$ | $51.97$ |
> | Llama3.1-8B (IF-Eval-like) | $1700/16/5×10^{-4}/1$ | $0.50$ |
> | Llama3.1-8B (HelpSteer2) | $19000/128/1×10^{-6}/3$ | $3.74$ |
>
> **W4 Baselines**
>
> Our work is specifically aimed at improving the scalability of online, RL-based alignment methods that require on-policy generation. In contrast, methods such as DPO, ORPO, and SIMPO belong to the class of offline direct-alignment approaches: they operate on a fixed, pre-collected, preference-labeled dataset and rely purely on supervised backpropagation without any online sampling. Consequently, their computational characteristics, scalability constraints, and deployment scenarios differ substantially from the online RL setup we study, and direct empirical comparisons are not the most informative for the questions this paper addresses.
>
> Furthermore, prior work has repeatedly shown that when online sampling is available, online RL-based alignment can systematically outperform offline direct-alignment methods [4, 5]. For tasks with verifiable rewards, notably mathematical reasoning and instruction following, it is also standard practice to use RL-based online alignment (e.g., RLVR) rather than DPO-style methods [6, 7]. This reflects an established separation in how these method families are typically used in practice.
>
> For these reasons, we view offline direct-alignment algorithms as addressing a complementary problem setting. Our objective is to provide a scalable alternative within the family of online, on-policy alignment methods, where compute and communication costs are dominated by continual sampling and online evaluation. In this context, GRPO is the most relevant and informative primary baseline for measuring the gains brought by ESSA.
>
> [1] Allal et al, The Smol Training Playbook: The Secrets to Building World-Class LLMs
>
> [2] Ouyang et al, Training language models to follow instructions with human feedback
>
> [3] He et al, Defeating Nondeterminism in LLM Inference
>
> [4] Tang et al, Understanding the performance gap between online and offline alignment algorithms
>
> [5] Ivison et al, Unpacking DPO and PPO: Disentangling Best Practices for Learning from Preference Feedback
>
> [6] Shao et al, DeepSeekMath: Pushing the Limits of Mathematical Reasoning in Open Language Models
>
> [7] Lambert et al, Tulu 3: Pushing Frontiers in Open Language Model Post-Training

---

> ### Author Response · Authors · 2025-11-17
>
> Thank you again for the constructive feedback. If these clarifications address your concerns, we would be grateful if you could consider raising your overall evaluation. If any points remain insufficiently addressed, we would be glad to hear additional questions or suggestions.

---

> > ### Comment · Reviewer_7J5i · 2025-11-19
> >
> > Most of my earlier concerns have been addressed, and I have updated my scores accordingly. One remaining point: updating only the singular values of the LoRA adapters may limit adaptability under distribution shift. Have you evaluated a setting where the warm-start (SFT) is performed on dataset A, while the CMA-ES optimization is conducted on a distinct dataset B (i.e., cross-dataset/domain generalization)? If so, please report the results; if not, a brief study would clarify ESSA’s robustness to OOD drift.

---

> > > ### Author Response · Authors · 2025-11-26
> > >
> > > Thank you for raising your evaluation and for the thoughtful feedback, we really appreciate it.
> > >
> > > Regarding your remaining question about how well ESSA adapts under distribution shift, we ran exactly the experiment you suggested.
> > >
> > > To make the test meaningful, we picked two tasks that are genuinely different from one another:
> > >
> > > - **HelpSteer2** consists of broad, general-purpose user queries (e.g., *“How does process intelligence or process mining work?”*) with feedback focused on whether answers are helpful, coherent, and factually accurate.
> > >
> > >     When running ESSA on HelpSteer2, we used the reward model RLHFlow/ArmoRM-Llama3-8B-v0.1 - the same setup as in the main paper.
> > >
> > > - **IFEval** consists of prompts with explicit, easily verifiable constraints on formatting and content (e.g., *“Write a story … no capital letters allowed … do not include keywords X, Y, Z.”*).
> > >
> > >     ESSA training on IFEval uses a verifiable rule-based reward, again matching the setup in the main paper.
> > >
> > >
> > > Because HelpSteer2 pushes the model toward better general-purpose response quality, while IFEval enforces rigid formatting and constraint-following, moving between them serves as a meaningful test of ESSA’s robustness to domain shift.
> > >
> > > For each target dataset **B**, we evaluated three settings:
> > >
> > > 1. **A → B:** SFT on A, ESSA on B
> > > 2. **B → B:** both SFT and ESSA on B
> > > 3. **Base → B:** ESSA on B without any SFT warm-start (default LoRA initialization with Kaiming-uniform for weight A and zeros for weight B)
> > >
> > > Below are the results for both cross-dataset directions.
> > >
> > > **IFEval → HelpSteer2:** https://8upload.com/display/fb2e41b65defbc4f/essa_if_hs.jpg.php (anonymous link)
> > >
> > > LoRA rank: 8, population size: 36, batch size: 100, alpha: 1
> > >
> > > **HelpSteer2 → IFEval:** https://8upload.com/display/e5be3c2ef65818bd/essa_hs_if.jpg.php (anonymous link)
> > >
> > > LoRA rank: 8, population size: 48, batch size: 500 alpha: 1
> > >
> > > Results:
> > >
> > > - In both directions, ESSA consistently improves performance on the target dataset, even when the SFT checkpoint comes from a different domain.
> > > - When moving IFEval → HelpSteer2, the gap between OOD-dataset and ID-dataset SFT is relatively small: the structured IFEval initialization already helps ESSA make rapid progress on HelpSteer2.
> > > - When moving HelpSteer2 → IFEval, the gap is larger, which is expected: signals in HelpSteer2 do not teach the model the kind of strict formatting and constraint-following that IFEval evaluates. Still, ESSA does make steady improvement over time, and significantly outperforms the Base → IFEval runs that start from random LoRA.
> > > - In both directions, SFT warm-starts are noticeably better than starting from scratch, confirming that even mismatched SFT provides structure for ESSA’s evolutionary search.
> > >
> > > We hope these additional results help clarify how ESSA behaves under distribution shift. If these experiments address your remaining concerns, we would be grateful if you could consider raising your score. And if there are any further tests you would like to see, we would be happy to run them within the remaining rebuttal window.

---

### Official Review · Reviewer_5kYU · 2025-11-03

**Soundness:** 3
**Presentation:** 3
**Contribution:** 3
**Rating:** 8
**Confidence:** 3

**Summary:**

This paper presents ESSA (Evolutionary Strategies for Scalable Alignment), a gradient-free framework for aligning large language models that uses evolutionary strategies to optimize low-rank LoRA adapters. The key innovation is decomposing LoRA matrices via SVD and optimizing only the singular values using CMA-ES, enabling inference-only training in quantized modes. The authors evaluate ESSA on mathematical reasoning, instruction following, and general assistant tasks, comparing against GRPO baselines.

**Strengths:**

* Novel Technical Approach. The LoRA + SVD + inference-only training is quite novel and interesting. Besides, the setting is quite realistic given the fact that the model is becoming larger and larger.
* Strong empirical results across different tasks such as math, instruction following and general-purpose assistants.
* Efficiency. Authors demonstrate faster inference and quantization compatibility.

**Weaknesses:**

I don't think the paper has major flaws. I am curious why SVD-GRPO performs much worse. Does that mean SVD and ES are coupled?

**Questions:**

See above

---

> ### Author Response · Authors · 2025-11-15
>
> Thank you very much for the thoughtful review and for highlighting the strengths of the work. We appreciate your careful reading and the positive evaluation of the technical novelty, empirical results, and efficiency benefits.
>
> We appreciate your question regarding the behavior of SVD-GRPO. The underperformance of SVD-GRPO does not indicate that “SVD and ES are coupled,” nor that SVD-LoRA is inherently incompatible with gradient-based RL. Instead, the gap arises from a formally analyzable property: the singular-value parameterization induces an extremely ill-conditioned optimization problem for first-order policy-gradient methods, while evolution strategies are provably invariant to such reparameterizations. Below we provide mathematical explanation, which we will integrate into the camera-ready version.
>
> Concretely, in the SVD-LoRA parameterization each adapter matrix is written as
>
> $A = U_A \operatorname{diag}(\sigma_A) V_A^\top,\quad$$B = U_B \operatorname{diag}(\sigma_B) V_B^\top,$
>
> with $U_A,V_A,U_B,V_B$ fixed and only the singular values $(\sigma_A,\sigma_B)$ trainable. The corresponding LoRA update is
>
> $\Delta W = BA = U_B \operatorname{diag}(\sigma_B) V_B^\top U_A \operatorname{diag}(\sigma_A) V_A^\top.$
>
> In this parameterization, GRPO must optimize an RL objective of the form
>
> $J(\sigma) = \mathbb{E}\_{\pi_{\sigma}}[A(s,a)],$
>
> where the policy $\pi_\sigma$ depends on  $\sigma$ only through the above bilinear $(\Delta W)$.
>
> The gradient terms  $\partial J / \partial \sigma_i$  reflect that each singular value $\sigma_i$  influences the model’s logits in a anisotropic way. A small perturbation in  $\sigma_i$  is propagated through multiple matrix multiplications, which makes some directions in $\sigma$ - space extremely sensitive while others remain almost flat.
>
> Because the LoRA update is bilinear, the Hessian in the singular-value space is fully coupled:
>
> $\frac{\partial^2 J}{\partial \sigma_i\partial \sigma_j}\neq 0$$\quad\forall i,j ,$
>
> resulting in strongly correlated, non-axis-aligned curvature. This phenomenon is not specific to SVD-GRPO, deep networks in general have dense, rotated curvature, but in high-dimensional parameterizations (e.g., full LoRA or full-matrix updates) these interactions are distributed over thousands of weakly correlated coordinates. Each parameter captures only a small portion of the curvature, allowing diagonal first-order optimizers to tolerate misalignment and average out errors across many degrees of freedom. After SVD compression, however, the same curvature is projected into a very low-dimensional subspace, where first-order adaptive optimizers like Adam struggle to optimize the parameters.
>
> CMA-ES, by contrast, naturally averages over a population of perturbations in the same low-dimensional space and adapts its covariance matrix to the observed fitness landscape, which substantially stabilizes learning. ES never computes $\partial J/\partial \sigma_i$. Its updates depend only on scalar rewards. Thus ESSA is completely invariant to the complex Jacobian structure of the SVD parameterization. Moreover as discussed in Hansen, “The CMA Evolution Strategy: A Tutorial,” see the sections 0.4 (Hessian and Covariance Matrices) and 5 (Discussion), CMA-ES adapts its covariance matrix in a way that aligns (in expectation) with the inverse Hessian of $J(\sigma)$ [1]. This property makes ESSA automatically adapt to exactly the curvature anisotropy that destroys GRPO.
>
> Thus, for ESSA, moving from direct LoRA parameters to SVD-LoRA simply reduces dimensionality without making the search problem harder. For GRPO, however, the same reparameterization changes both the curvature and the gradient statistics in a way that standard first-order updates are not well adapted to.
>
> We observed that even after sweeping learning rates SVD-GRPO saturated around $≈ 0.5$ on PRM800K, whereas ESSA with LoRA rank = 2 reached  $≈ 0.72$ in the same search space (Fig. 8). In another work, Transformer-Squared [2] the authors also train SVD-based vectors via RL. Their learning curves (Fig. 4 of the paper; Math, Code, Reasoning) show that reward increases only gradually and requires many epochs even for simple tasks.
>
> We will add a short clarification along these lines to the Discussion.
>
> [1] Hansen, The CMA Evolution Strategy: A Tutorial
>
> [2] Sun et al, Transformer-Squared: Self-adaptive LLMs

---

### Author Response · Authors · 2025-11-25

We would like to sincerely thank all reviewers for their thoughtful comments and constructive suggestions. Your feedback helped us make the paper clearer, more polished, and easier to navigate, while keeping the core methodology and results exactly as originally presented. Below we summarize the refinements introduced in the revised version.

**Enhanced clarity of the problem framing and contributions.** The introduction (Section 1) now provides a more streamlined explanation of the motivation behind ESSA and highlights the key contributions more explicitly. This framing strengthens the narrative around the role of SVD-LoRA, the inference-only alignment loop, and the communication efficiency of the approach.

**More precise and self-contained presentation of ESSA.** In the method section (3.2), we refined the flow of the exposition so that ESSA can be understood end-to-end without cross-referencing external sources. Definitions such as the structure of the SVD-LoRA parameterization, the meaning of “solution length”, and how CMA-ES interacts with each slice are now integrated directly into the text. Figure 1 has been updated accordingly so that it visually mirrors the linear description in Section 3.2.

**Expanded experimental narrative.** The experimental section now presents the results in a more cohesive way, making it easier to follow how ESSA behaves across different initialization regimes and hardware budgets. In addition to the main experiments, we reference two complementary settings that further contextualize the method without altering its core narrative. The first is the scenario where ESSA is initialized directly from the pretrained base models of Qwen2.5-7B and Qwen2.5-Math-7B on GSM8K; the corresponding learning curves are provided in Appendix Figure 29. The second is a compact illustration of how ESSA and GRPO operate under single-GPU training on GSM8K, with details shown in Appendix Figure 30. Together with the primary experiments, these references help situate ESSA within a broader range of practical configurations.

**Clearer discussion of precision and resource configurations.** Section 4 now consolidates all precision- and resource-related details in one place, making the experimental setup easier to follow across model sizes. The default precision for both ESSA and GRPO is stated explicitly, and the points at which BF16 or INT4 are used are highlighted uniformly throughout the section. In addition, Figure 5(b) now includes a complementary experiment showing ESSA training the 72B model in BF16, alongside the INT4 configuration, so that readers can see both precision regimes under the same setup. Full configuration details appear directly in Section 4, with complete parameters summarized in Appendix F.

**Additional intuition for SVD-GRPO.** In Section 4.4.5 we now provide an expanded, more intuitive explanation of why GRPO underperforms when restricted to the same SVD-LoRA singular-value parameterization used by ESSA.

We appreciate the reviewers’ engagement with the work. The revised paper preserves the original contributions and empirical results while presenting them in a clearer and more cohesive form. We hope that the strengthened exposition addresses the questions raised in the reviews, and we would be grateful if you could consider reflecting this in your final evaluation.

If any further clarification would be helpful during the discussion phase, we will be happy to provide it.

---

### Meta-Review · Area_Chair_nku8 · 2026-01-11

**Summary:**

Reviewers raised concerns spanning both technical methodology and presentation quality. On the technical side, multiple reviewers questioned the method's fundamental dependence on strong SFT initialization and whether it could work effectively without such warm-starts. There was particular interest in understanding why SVD-GRPO underperforms compared to ESSA when using the same singular-value parameterization, and whether this indicates an essential coupling between SVD and evolutionary strategies. Concerns about baseline fairness emerged around precision mismatches between methods (INT4 versus BF16), limited hyperparameter exploration for GRPO compared to ESSA, and the absence of offline alignment baselines like DPO, ORPO, and SIMPO. Reviewers also questioned whether optimizing only singular values would limit the method's adaptability under distribution shift, and noted the heavy reliance on mathematical reasoning datasets with limited diversity in model families, raising questions about generalizability to non-specialized models and broader alignment tasks. On the presentation side, reviewers identified poor writing quality with grammatical errors, malformed figures and tables, missing key definitions, unclear algorithmic descriptions, ambiguous terminology like "strong base" and "latency model," and incomplete accounting of SFT costs in the reported timing comparisons. \

The paper has enough merits but needs significant revision to be accepted. A thorough rebuttal successfully addressed major concerns about SFT dependence, distribution shift robustness, SVD-GRPO underperformance, and experimental scope, bringing three reviewers to scores at or above the acceptance threshold. The sole remaining low score (2) comes from a reviewer who never engaged in discussion despite comprehensive responses that directly contradicted their criticisms with evidence already present in the paper. The authors must incorporate all clarifications, additional experiments, and improved presentations developed during the rebuttal into the subsequent  version to meet the acceptance standard.

**Reviewer Concerns:**

The authors' rebuttal successfully addressed many of the major technical and presentation concerns raised by reviewers. Regarding SFT initialization, the authors clarified that exactly one SFT checkpoint per backbone-task pair is shared identically across all experimental runs for both ESSA and GRPO, with complete hyperparameters now documented in Appendix F, which satisfied Reviewer 7J5i's reproducibility concerns. For distribution shift robustness, new cross-domain experiments transferring between HelpSteer2 and IFEval demonstrated that ESSA consistently improves performance even with mismatched SFT initialization, though performance gaps exist compared to matched initialization, leading Reviewer 7J5i to explicitly raise their score from 2 to 6. The SVD-GRPO underperformance was explained through added theoretical analysis showing that the singular-value parameterization creates an ill-conditioned optimization landscape for first-order gradient methods while CMA-ES remains naturally invariant to such reparameterizations, satisfying Reviewer 5kYU who maintained their score of 8.

**Reviewer Scores:**

Some concerns were only partially addressed. The theoretical contribution clarity improved with formal communication cost analysis, but Reviewer DR9j never re-engaged to confirm satisfaction, leaving this resolution uncertain. The baseline breadth issue received justification rather than empirical resolution, with authors explaining that DPO, ORPO, and SIMPO are offline methods addressing a complementary problem setting, though this remains a methodological choice rather than an empirically validated position. Most significantly, Reviewer ojT5's fundamental question about why optimizing only singular values should be effective received empirical demonstration through MNIST experiments showing ESSA tracks gradient-based optimization surprisingly well when meaningful initialization exists, but the theoretical gap between the restrictive parameterization and practical effectiveness relies somewhat on appeals to "latent structure" in pretrained models without rigorous formalization. \

Reviewer DR9j's concerns remain entirely unaddressed due to their non-participation in the discussion despite detailed author responses, and many of their criticisms appear to contradict content explicitly present in the paper, such as claiming unclear applicability to non-specialized models when Llama experiments are included.

---

### Decision · Program_Chairs · 2026-01-26

Reject